# Principal-Agent Bandit Games with Self-Interested and Exploratory Learning Agents

**Junyan Liu** [1]   **Lillian J. Ratliff** [2]

## Abstract

This paper studies the repeated principal-agent bandit game, where the principal indirectly explores an unknown environment by incentivizing an agent to play arms. Unlike prior work that assumes a greedy agent with full knowledge of reward means, we consider a self-interested learning agent who iteratively updates reward estimates and may explore arbitrarily with some probability. As a warm-up, we first consider a self-interested learning agent without exploration. We propose algorithms for both i.i.d. and linear reward settings with bandit feedback in a finite horizon $T$, achieving regret bounds of $\widetilde{\mathcal{O}}(\sqrt{T})$ and $\widetilde{\mathcal{O}}(T^{2/3})$, respectively. Specifically, these algorithms rely on a novel elimination framework coupled with new search algorithms which accommodate the uncertainty from the agent's learning behavior. We then extend the framework to handle an exploratory learning agent and develop an algorithm to achieve a $\widetilde{\mathcal{O}}(T^{2/3})$ regret bound in i.i.d. reward setup by enhancing the robustness of our elimination framework to the potential agent exploration. Finally, when our agent model reduces to that in Dogan et al. (2023a), we propose an algorithm based on our robust framework, which achieves a $\widetilde{\mathcal{O}}(\sqrt{T})$ regret bound, significantly improving upon their $\widetilde{\mathcal{O}}(T^{11/12})$ bound.

## 1. Introduction

Bandits learning is a powerful framework for solving a broad spectrum of sequential decision-making problems, such as recommendation systems (Li et al., 2010), clinical trials (Villar et al., 2015), and resource allocation (Latti-more et al., 2015). In most bandit frameworks, a player *directly* interacts with an unknown environment by repeatedly playing arms and observing the corresponding rewards. However, those frameworks often fail to capture the unique challenges faced in many online marketplaces, where the player can only *indirectly* interact with the environment. For instance, in online shopping, the website (player) learns users' preference (unknown environment) by observing purchase behaviors. To acquire a comprehensive knowledge of users' preferences, the website typically needs to provide external incentives, such as discounts or coupons, for agents to encourage exploration.

Motivated by such scenarios, recent work (e.g., Dogan et al. (2023a;b); Scheid et al. (2024b)) frames this problem as a *principal-agent bandit game*, modifying the classic principal-agent problem in economics (see, e.g., Bolton & Dewatripont (2004); Laffont & Martimort (2009)) with repeated engagements and stochastic rewards where the aforementioned player refers to a principal, and the agent refers to end-users. With arms representing a vector of incentives, in each round of the game, the principal first selects an arm, and after observing these incentives, the agent *greedily* selects amongst their own arms to maximize their expected reward plus the offered incentive (henceforth we call this arm *true-maximizer*). The chosen arm yields two stochastic rewards drawn from different distributions: one for the principal and one for the agent. Notably, the principal observes their own reward and the selected arm, yet remains agnostic of the reward on the agent side.

Despite the merger of classic models of incentives with bandits, most work relies on an *oracle-agent assumption*, where the agent has *complete knowledge* of the true reward means (expected rewards) and always selects the true-maximizer. This assumption, in general, will not hold in real-world scenarios. For instance, in the case of online shopping, users rarely fully understand their own preferences over many options without a prolonged learning process. Recently, a notable step towards a more realistic model was taken by Dogan et al. (2023a) who consider an an agent that employs an exploratory learning behavior: the agent is allowed to explore an arm, different from the true-maximizer, with a small probability decreasing over time. However, even in

[1]Paul G. Allen School of Computer Science & Engineering, University of Washington, Seattle, WA USA [2]Electrical & Computer Engineering, University of Washington, Seattle, WA USA. Correspondence to: Junyan Liu <junyanl1@cs.washington.edu>.

*Proceedings of the 42$^{nd}$ International Conference on Machine Learning*, Vancouver, Canada. PMLR 267, 2025. Copyright 2025 by the author(s).

this more practical model, the assumption remains that the agent selects the true-maximizer when not exploring[1], and their result only achieves an $\widetilde{\mathcal{O}}(T^{11/12})$ regret bound where $T$ is the horizon.

A more recent work by Scheid et al. (2024a) allows the agent to adopt no-regret learning algorithms that adhere to the *key assumption* that the no-regret property holds *universally* for any time period and any principal's algorithm offering constant incentives during this period. However, this universal no-regret preservation assumption limits their framework's applicability to certain simple, yet realistic, learning behaviors. This is due to the fact that their assumption implicitly requires the agent's no-regret algorithm to *strategically* explore the environment, but in many real-world applications such as online shopping, the agent (buyer) is unlikely to design such sophisticated strategies. Consequently, their agent model does not encompass simple but arguably realistic agent models such as the self-interested learning agent with and without arbitrary exploration (see Appendix B for details). Given results , it still remains unclear what regret bounds can be achieved for these realistic agent models and whether $\widetilde{\mathcal{O}}(T^{11/12})$ regret bound in Dogan et al. (2023a) can be improved. This motivates two natural questions:

1. *Can we generalize the agent's behavior considered in Dogan et al. (2023a)?*

2. *Given the aforementioned generalization, can we design algorithms that outperform the $\widetilde{\mathcal{O}}(T^{11/12})$ regret bound?*

In this paper, we provide affirmative answers to both questions. To address the first question, we consider a self-interested learning agent with exploration behavior which allows the agent to explore non-maximizer arms with a decreasing probability similar to to the setup in (Dogan et al., 2023a). The key distinction is that when the agent decides not to explore, she plays an arm that maximizes the *empirical mean* plus the incentive (henceforth we call this arm empirical maximizer), instead of the true-maximizer as assumed by Dogan et al. (2023a). As discussed in Section 2, this learning behavior subsumes that of (Dogan et al., 2023a) as a special case. Under this more realistic learning behavior, we examine the i.i.d. reward setting, where the rewards of the principle and the agent are independently sampled from distinct yet fixed distributions. Further, we study the linear reward setting, in which the rewards of arms can be linearly approximated.

The contributions are summarized as follows; refer to Table 1 for a summary table of regret bounds.

---

[1]One caveat here is that with a special coordination between the principal and the agent, an agent can select the true-maximizer without the knowledge of the expected rewards. We refer readers to Appendix C for a discussion.

- **I.I.D. rewards:** We first propose Algorithm 1 for the i.i.d. reward and greedy agent setting, where the agent always greedily chooses the empirical maximizer without exploration. The proposed algorithm enjoys $\mathcal{O}\big(\sqrt{KT\log(KT)}\big)$ expected regret bound, where $K$ is the number of arms. The proposed algorithm is established upon a novel elimination framework coupled with an efficient search algorithm (Algorithm 3). Different from existing elimination approaches that permanently eliminate badly-behaved arms, our algorithm plays bad arms in some rounds so as to benefit Algorithm 3 in future searches for optimal incentives. The regret upper bound of our algorithm matches the lower bound up to a logarithmic factor. Notably, when reducing to the oracle-agent setup, our worst-case bound matches that of (Scheid et al., 2024b, Algorithm 1) as long as $T \geq K$.

  We further propose Algorithm 5 for a self-interested learning agent with exploration and the i.i.d. reward setting, achieving a $\widetilde{\mathcal{O}}(K^{1/3}T^{2/3})$ regret bound. In particular, Algorithm 5 builds upon Algorithm 1 and leverages the probability amplification idea to enhance its robustness to agent exploration. Abstractly, Algorithm 5 repeats the incentive search and elimination processes logarithmically many times. It uses an *incentive testing procedure* to identify the most reliable estimated incentives and employs a *median selection strategy* to refine the active arm set. Since we consider a more general learning behavior than Dogan et al. (2023a), our algorithm improves their regret bounds from $\widetilde{\mathcal{O}}(T^{11/12})$ to $\widetilde{\mathcal{O}}(T^{2/3})$. Furthermore, by reducing our agent's learning behavior to the one studied by Dogan et al. (2023a), we can achieve an even better regret bound $\widetilde{\mathcal{O}}(\sqrt{KT})$.

- **Linear rewards:** We introduce Algorithm 7 for the linear reward setting, achieving an expected regret bound of $\widetilde{\mathcal{O}}(d^{4/3}T^{2/3})$, where $d$ is the dimension. Our algorithm is grounded in the new elimination framework analogous to the i.i.d. case, but more importantly it is equipped with an efficient and robust search method(Algorithm 8) for the high-dimensional case. Specifically, Algorithm 8 is built upon (Liu et al., 2021, Algorithm 1), but it cuts the space conservatively to account for the uncertainty resultant from the learning agent.

**Related Works.** The principal-agent model is a fundamental problem in contract theory (Bolton & Dewatripont, 2004; Grossman & Hart, 1992; Laffont & Martimort, 2009). Motivated by online marketplaces, a substantial body of research has explored repeated principal-agent models to, for example, estimate the agent model or minimize the regret for the principal (Ben-Porat et al., 2024; Fiez et al., 2018a;b; Ho et al., 2014; Ivanov et al., 2024; Ratliff & Fiez, 2020; Zhu et al., 2022; 2023). Categorized by different types of information asymmetry, principal-agent problems are typ-

Table 1: Overview of related work for principal-agent bandit games with different agent's behaviors.

| Algorithms | Agent behavior involves | | Reward | Regret bound[b] |
|---|---|---|---|---|
| | Empirical mean?[a] | Exploration? | | |
| Dogan et al. (2023b) | ✗ | ✗ | | $\mathcal{O}\big(\max\{K, \sqrt{\log T}\}\sqrt{T}\big)$ |
| IPA Scheid et al. (2024b)[c] | ✗ | ✗ | | $\mathcal{O}\big(\sqrt{KT\log(T)}\big)$ |
| Algorithm 1 (**This paper**) | ✓ | ✗ | i.i.d. | $\mathcal{O}\big(\sqrt{KT\log(T)}\big)$ |
| Dogan et al. (2023a) | ✗ | ✓ | | $\mathcal{O}\big(\text{poly}(K)T^{\frac{11}{12}}\sqrt{\log T}\big)$ |
| Algorithm 6 (**This paper**) | ✗ | ✓ | | $\mathcal{O}\big(\log^2(T)\sqrt{KT}\big)$ |
| Algorithm 5 (**This paper**) | ✓ | ✓ | | $\mathcal{O}\big(K^{\frac{1}{3}}T^{\frac{2}{3}}\log^{\frac{2}{3}}(T)\big)$ |
| C-IPA Scheid et al. (2024b)[d] | ✗ | ✗ | linear | $\mathcal{O}(d\sqrt{T}\log T)$ |
| Algorithm 7 (**This paper**) | ✓ | ✗ | | $\mathcal{O}\big(d^{\frac{4}{3}}T^{\frac{2}{3}}\log^{\frac{2}{3}}(T)\big)$ |

[a] We use ✗ to indicate that the agent behavior is defined by true means.
[b] For fair comparison, we use the same regret metric to compare all regret bounds. We refer readers to Appendix A for details. All our algorithms can work in the oracle-agent setup without modification and maintain the same regret bounds. We assume a large $T$ to drop lower order terms.
[c] We here present the worst-case bound of IPA, but it also enjoys a gap-dependent bound in the oracle-agent setup.
[d] Contextual IPA (C-IPA) works for time-varying arm sets and thus also works in our fixed arm linear reward setting.

ically studied in moral hazard setting (Abreu et al., 1990; Radner, 1981; Rogerson, 1985; Sannikov, 2013; Wu et al., 2024) where the agents' actions are invisible to the principal, and adverse selection setting (Dogan et al., 2023a;b; Eső & Szentes, 2017; Fiez et al., 2018a;b; Gottlieb & Moreira, 2022; Halac et al., 2016; Ratliff & Fiez, 2020; Scheid et al., 2024b), in which the principal does not known the agent types/preferences. While theoretical results are well established in these models, they typically assume that the agent has a full knowledge of the underlying parameters, and the reward functions remain fixed for both principal and agent.

Recently, Lin & Chen (2025) considers a no-regret learning agent problem, but assume that the principal has full knowledge of its utility function. Fiez et al. (2018a;b) consider incentive design as a bandit problem in a dynamic environment where the users preferences *may change in time* according to a Markov process thereby resulting in dynamic, stochastic rewards that are correlated over time from the perspective of the principal. Different from our approach, this work employs an epoch-based algorithm to take advantage of the mixing of the Markov process. Dogan et al. (2023b); Scheid et al. (2024b); Wu et al. (2024) consider reward uncertainty on the principal's side, though they still rely on the assumption of an oracle-agent who always selects the true maximizer. Dogan et al. (2023a) took a step further by incorporating a small degree of uncertainty on the agent's side, yet the agent still acts as a true maximizer with increasingly high probability. In contrast, we study the scenario where both the agent and principal are always faced with uncertainty and the agent can only make decisions based on her empirical estimations and incentives. This problem raises new challenges and requires novel approaches to solve.

A recent work by Scheid et al. (2024a) studies the principal-agent interactions under the assumption that for any time period and any principal's algorithm offering constant incentives during this period, the agent's learning algorithm always preserves the no-regret property. While this enables a regret bound of $\widetilde{\mathcal{O}}(T^{3/4})$ if the agent adopt some algorithms with $\sqrt{t}$-type regret, this framework excludes simple yet realistic agent behaviors, such as self-interested learning agents with or without arbitrary exploration (see Appendix B for details). Moreover, their approach involves binary search per arm, which is impractical in linear reward settings due to its linear dependence on the number of arms. In this paper, we address these questions by focusing on these simple yet realistic learning behaviors.

**Notations.** Let $\text{Vol}(\cdot)$ be the standard volume in $\mathbb{R}^d$ and let $\|x\|$ be $\ell_2$-norm of vector $x$. For a positive definite matrix $A \in \mathbb{R}^{d \times d}$, the weighted 2-norm of vector $x \in \mathbb{R}^d$ is given by $\|x\|_A = \sqrt{x^\top A x}$. For matrices $A, B$, we use $A \succ B$ ($A \succeq B$) to indicate that $A - B$ is positive (semi-)definite. For two sets $\mathcal{A}, \mathcal{B}$, we use $\mathcal{A} - \mathcal{B}$ to indicate the exclusion. We use $\widetilde{\mathcal{O}}(\cdot)$ to suppress (poly)-log terms.

## 2. Preliminaries

We consider a repeated principal-agent game over a finite horizon $T$, where the agent is given an arm set $\mathcal{A}$ with $|\mathcal{A}| = K$, and $\mathcal{A}$ is also known to the principal. At each round $t \in [T]$, the principal first proposes an incentive $\pi(t) = (\pi_1(t), \ldots, \pi_K(t)) \in \mathbb{R}^K_{\geq 0}$. Then, the agent observes incentive $\pi(t)$ and plays an arm $A_t \in \mathcal{A}$ according to a certain strategy. After playing arm $A_t$, the agent observes reward $R_{A_t}(t)$ while the principal observes only the arm $A_t$

played by the agent and herself reward $X_{A_t}(t)$. The utility of the principal at round $t$ is given by $X_{A_t}(t) - \pi_{A_t}(t)$. Throughout the interaction, the agent updates the empirical means $\{\widehat{\mu}_a(t)\}_{a \in \mathcal{A}}$ whenever playing an arm. In this paper, we focus on the following two learning behaviors.

**Self-interested learning agent:** At each round $t$, the agent selects an arm $A_t$ that maximizes the empirical mean plus an incentive proposed by the principal:

$$A_t \in \underset{a \in \mathcal{A}}{\operatorname{argmax}} \left\{ \widehat{\mu}_a(t) + \pi_a(t) \right\}, \tag{1}$$

where ties are broken arbitrarily. Similar to Dogan et al. (2023b); Scheid et al. (2024b), we assume the principal knows the agent's selection strategy in Eq. (1), but has no knowledge of the empirical means, $\{\widehat{\mu}_a(t)\}_{t,a}$.

**Exploratory learning agent:** This behavior generalizes the above by allowing the agent to play arms other than the empirical maximizer. Formally:

**Definition 2.1.** Let the probability that agent explores at round $t$ be $p_t = \mathbb{P}(A_t \notin \operatorname{argmax}_{a \in \mathcal{A}} \{\widehat{\mu}_a(t) + \pi_a(t)\})$. There exists a absolute constant $c_0 \geq 0$ such that $p_t \leq c_0 \sqrt{t^{-1} \log(2t)}$ at any time step $t \in [\tau, T]$ where $\tau \geq 2$ is minimum value satisfying $c_0 \sqrt{\log(2\tau)} < \sqrt{\tau}$.

In fact, $p_t$ in Definition 2.1 captures the probability that the agent deviates from playing the empirical-maximizer. This exploration behavior is not only analytically convenient but also motivated by behavioral and cognitive science where this type of model has been widely used to characterize how humans make sequential decisions under uncertainty (Barron & Erev, 2003; Daw et al., 2006; Gershman, 2018; Kalidindi & Bowman, 2007; Lee et al., 2011).

Similar to Dogan et al. (2023a), we do not restrict how the agent explores arms and assume that the principal knows: (i) the agent either chooses an empirical-maximizer or arbitrarily explores an arm; (ii) $p_t$ is upper bounded by a known function $c_0 \sqrt{t^{-1} \log(2t)}$. We further assume that $p_t$ is as a function of history. In fact, our algorithm also works when $p_t$ only depends on $t$ (e.g., agent adopts $\epsilon$-greedy algorithm).

Then, we specify the reward models and how agent updates the empirical means for the i.i.d. and linear reward settings.

**I.I.D. reward setting.** In this setting, we let $\mathcal{A} = [K] := \{1, \ldots, K\}$. For each arm $a \in [K]$, the reward of each arm $a$ for the principal is drawn from a $[0,1]$-bounded distribution $\mathcal{D}_a^P$ with mean $\theta_a$, and the reward of each arm $a$ for the agent is drawn from another $[0,1]$-bounded distribution $\mathcal{D}_a^A$ with mean $\mu_a$. The agent is unaware of $\mu_a$, but keeps track of $N_a(t)$, the number of plays of arm $a$ before round $t$ and computes $\widehat{\mu}_a(t) = \frac{\widehat{\mu}_a^0 + \sum_{s=1}^{t-1} R_{A_s}(t) \mathbb{I}\{A_s = a\}}{\max\{1, N_a(t)\}}$, where $\widehat{\mu}_a^0$ is the initial empirical mean for arm $a$[2], which is allowed to

be chosen arbitrarily in the range of $[0,1]$.

**Linear reward setting.** We assume $\mathcal{A} \subseteq B(0,1)$ where $B(0,1) = \{x \in \mathbb{R}^d : \|x\| \leq 1\}$, and without loss of generality, $\mathcal{A}$ spans $\mathbb{R}^d$. The reward observed by the agent is $R_{A_t}(t) = \langle s^\star, A_t \rangle + \eta^A(t)$, and the reward on the principal side is $X_{A_t}(t) = \langle \nu^\star, A_t \rangle + \eta^P(t)$ where $s^\star, \nu^\star \in B(0,1)$ are unknown to agent and principal, and $\eta^A(t)$ and $\eta^P(t)$ are assumed to be conditionally 1-subgaussian. Assume that at each round $t$, the agent uses the ordinary least square to estimate $\widehat{s}_t = \left(\sum_{s=1}^t A_s A_s^\top\right)^\dagger \sum_{s=1}^t R_{A_s}(s) A_s$ for $t \geq 2$ where for any matrix $M$, $M^\dagger$ is Moore–Penrose inverse. Moreover, $\widehat{s}_1$ can be arbitrarily selected from $B(0,1)$. Then, $\widehat{\mu}_a(t) = \langle \widehat{s}_t, a \rangle$ is the empirical mean of arm $a$ at round $t$.

**Optimal incentive and regret.** For any arbitrary $\epsilon > 0$, the principal ensures to incentivize the agent to play arm $a$ by proposing incentive $\pi_a^\epsilon(t)$ on arm $a$ as:

$$\pi_a^\epsilon(t) = \max_{b \in \mathcal{A}} \widehat{\mu}_b(t) - \widehat{\mu}_a(t) + \epsilon,$$

and $\pi_{a'}(t) = 0$ for all $a' \neq a$. This incentive ensures that $\pi_a^\epsilon(t) + \widehat{\mu}_a(t) > \widehat{\mu}_b(t)$ for all $b \neq a$, which implies that $A_t = a$. The minimum incentive to force the agent to play a target arm $a$ at round $t$ is

$$\pi_a^\star(t) := \inf_{\epsilon > 0} \pi_a^\epsilon(t) = \max_{b \in \mathcal{A}} \widehat{\mu}_b(t) - \widehat{\mu}_a(t). \tag{2}$$

The principal aims to minimize regret $\mathbb{E}[R_T]$ where

$$R_T = \sum_{t=1}^T \left( \max_{a \in \mathcal{A}} \{\theta_a - \pi_a^\star(t)\} - (\theta_{A_t} - \pi_{A_t}(t)) \right). \tag{3}$$

Here, $\mathbb{E}[R_T]$ measures the cumulative per-round gap between the expected utility of an oracle with knowledge of the optimal incentives and that of our algorithm. In other words, we hope to find the *best per-iterate* incentive adapting to agent's empirical means at each round, rather than the *best fixed* incentive in hindsight. For the exploratory learning agent, since $A_t$ may be an arbitrary arm when agent explores, independent of the offered incentives, achieving maximal utility is infeasible in certain rounds. However, the total regret incurred during exploration is at most $\mathcal{O}(\sqrt{T})$ regardless of the benchmark. To maintain consistency, we still compare against the best achievable utility, as in (Dogan et al., 2023a). Moreover, if the principal is restricted to incentivize only a single arm, our regret aligns with that of Scheid et al. (2024b). We refer readers to Appendix A for a more detailed discussion on the implications of different regret definitions.

---

[2]For simplicity, we assume that the initial empirical means are

all equal to zero, but our algorithms work for the general unknown $\{\widehat{\mu}_a^0\}_a$ case with only constant modification. We refer readers to Appendix D.1 for more details.

*Remark* 2.2 (**Generalization**). Since the principal is agnostic to the agent's empirical means, by assuming that the agent always receives the constant reward equal to the true mean and the initial value is $\widehat{\mu}_a^0 = \mu_a$ for every $a \in \mathcal{A}$, one can see that Eq. (1) generalizes the agent model of (Scheid et al., 2024b), and Definition 2.1 generalizes the agent's behavior of Dogan et al. (2023a).

*Remark* 2.3 (**Lower bound**). As our setting subsumes the oracle-agent setting as a special case when $R_a(t) = \widehat{\mu}_a^0 = \mu_a$, the lower bound for the oracle-agent problem serves as a lower bound for our problem. This implies that it remains to lower-bound $\mathbb{E}[T(\theta_{a^\star} + \mu_{a^\star}) - \sum_{t=1}^{T}(\theta_{A_t} + \mu_{A_t})]$. Further, even if the principal has access to optimal incentives, the principle still needs to solve a stochastic bandits problem with mean $\theta_a + \mu_a$. Thus, the lower bound of stochastic MAB (e.g., Lattimore & Szepesvári (2020)) $\mathbb{E}[R_T] = \Omega(\sqrt{KT})$ also serves as a lower bound for our i.i.d. reward problem and the lower bound of stochastic linear bandits $\mathbb{E}[R_T] = \Omega(\sqrt{dT})$ serves as a lower bound for our linear reward problem setting (Dani et al., 2008).

# 3. Self-Interested Learning Agent & IID Rewards

In this section, we propose Algorithm 1[3] for the self-interested learning agent and i.i.d. reward case, which achieves $\mathcal{O}(\sqrt{KT \log(KT)})$ regret bound. The omitted proof can be found in Appendix D.

## 3.1. Novel Elimination Framework

We build our algorithm upon a new phased-elimination scheme, which consists of three components, each with an underscore in Algorithm 1. The algorithm proceeds in phases indexed by $m$, and the phase length increases exponentially. In each phase $m$, the algorithm maintains an active *good* arm set $\mathcal{A}_m$ and a *bad* arm set $\mathcal{B}_m = [K] - \mathcal{A}_m$.

One distinction between our elimination framework and that of Even-Dar et al. (2006) is that in each phase, the algorithm plays all bad arms (if any) for a *moderate* number of times. This stabilizes their estimators while incurring only moderate regret. Once the estimators are stable, the algorithm searches for optimal incentives, with the error decreasing inversely with the phase length. As the algorithm proceeds in phases, estimation accuracy improves, while the number of searches is only logarithmic. The algorithm then uses accurately estimated incentives to entice the agent to uniformly explore the active arms. Finally, the algorithm identifies bad-behaved arms and steps into the next phase.

---

[3]For all algorithms proposed in this paper, the principal tracks and updates $\{N_a(t), \widehat{\theta}_a(t)\}_{a \in [K]}$ whenever an arm is played. We omit these updates for the ease of presentation.

---

**Algorithm 1** Proposed algorithm for i.i.d. reward

**Input**: confidence $\delta \in (0,1)$, horizon $T$.
**Initialize**: $\mathcal{A}_1 = [K]$, $\mathcal{B}_1 = \emptyset$, $T_0 = 1$.

1 **for** $m = 1, 2, \ldots$ **do**
2    Set $T_m$ as

$$T_m = \max\left\{2^{2m+5}\log(4TK\delta^{-1}), |\mathcal{A}_m|\log T\right\}. \quad (4)$$

3    **for** $a \in \mathcal{B}_m$ **do**    ▷ Stabilize estimators for bad arms
4      Propose incentives $\pi^0(a; 1 + 1/T)$ for $Z_m$ rounds where $Z_m$ is given in Eq. (7).
5    **for** $a \in \mathcal{A}_m$ **do**
6      Invoke Algorithm 3 with input $(a, T)$ to get $b_{m,a}$.   ▷ Search near-optimal incentive
7      Set $\bar{b}_{m,a}$ based on Eq. (6).
8      Propose incentives $\pi^0(a; \bar{b}_{m,a})$ for $T_m$ rounds.
9    **for** $a \in \mathcal{A}_m$ **do**    ▷ Online Elimination
10      Let $t$ be the round current, and $\{\widehat{\theta}_a(t)\}_{a \in [K]}$ are empirical means of all arms at round $t$.
11      Propose incentives $\pi(t)$ with $\pi_a(t) = 1 + \widehat{\theta}_a(t) + \frac{3}{2} \cdot 2^{-m}$, $\pi_b(t) = 1 + \widehat{\theta}_b(t)$, $\forall b \in \mathcal{A}_m - \{a\}$, and $\pi_i(t) = 0$, $\forall i \in \mathcal{B}_m$.
12      If $A_t \neq a$, then update $\mathcal{A}_{m+1} = \mathcal{A}_m - \{a\}$ and $\mathcal{B}_{m+1} = \mathcal{B}_m \cup \{a\}$.

---

**Incentivize which arm?** Now, we figure out which arm should be incentivized. To minimize regret, we aim to incentivize agent to play an arm with small regret. From Eq. (3), the regret of a single round $t$ is $\max_{a \in \mathcal{A}}\{\theta_a - \pi_a^\star(t)\} - (\theta_{A_t} - \pi_{A_t}(t))$. Assume that we have accurate estimates of optimal incentives ($\pi_{A_t}(t) \approx \pi_{A_t}^\star(t)$), and the regret at round $t$ is $\max_{a \in \mathcal{A}}\{\theta_a + \widehat{\mu}_a(t)\} - (\theta_{A_t} + \widehat{\mu}_{A_t}(t))$. As the agent observes more samples, the empirical mean $\widehat{\mu}_a(t)$ will be closer to the true mean $\mu_a$. Therefore, the regret at a large round $t$ will be roughly $\max_{a \in \mathcal{A}}\{\theta_a + \mu_a\} - (\theta_{A_t} + \mu_{A_t})$. Hence, our algorithm will incentivize an arm that maximizes the joint true means of the principal and the agent.

Before explaining each component, let $\pi^0(a; c) \in \mathbb{R}_{\geq 0}^K$ be a single-arm incentive, $\forall a \in [K], \forall c \in \mathbb{R}_{>0}$:

$$[\pi^0(a; c)]_i = \begin{cases} c, & \text{if } i = a, \\ 0, & \text{otherwise.} \end{cases} \quad (5)$$

## 3.2. Efficient Incentive Search

We first describe our key subroutine (Algorithm 3 deferred to Appendix D.2) for the near-optimal incentive search. The search algorithm aims to provide an estimation for the target arm $a$. The following lemma characterizes the estimation error and efficiency. Let $b_{m,a}$ be the estimated incentive for arm $a$ in phase $m$, outputted by Algorithm 3.

**Lemma 3.1.** *For each phase* $m$*, if Algorithm 3 ends the search for arm* $a$ *at round* $t$ *in phase* $m$*, then*

$$b_{m,a} - \pi_a^\star(t) \in \left(0, \frac{4}{T} + \frac{\lceil \log_2 T \rceil}{N_a(t)} + \frac{2}{\min_{i \in [K]} N_i(t)}\right].$$

*Moreover, Algorithm 3 lasts at most* $2\lceil \log_2 T \rceil$ *rounds.*

The estimation error suffers $(\min_{i \in [K]} N_i(t))^{-1}$ because the search algorithm compares the empirical means between the target arm $a$ and the agent's best empirical arm $\operatorname{argmax}_{b \in [K]} \widehat{\mu}_b(t)$, which could be arbitrarily suboptimal on the principal's side. To control this error, we lower-bound it by $\min_{i \in [K]} N_i(t)$. Algorithm 3 inherits a basic binary search structure (line 4-line 11), while incorporating an asymmetric check for better accuracy.

**Asymmetric check.** The goal of this design is to ensure the output estimation $b_{m,a}$ is *strictly larger than* and close to the optimal incentive for arm $a$. The algorithm refines the search range $[\underline{y}_a(t), \overline{y}_a(t)]$ by testing whether the offered incentive entice the agent to play the target arm. In oracle-agent setup, running standard binary search $\mathcal{O}(\log T)$ times ensures $\pi_a^\star(t) \in [\underline{y}_a(t), \overline{y}_a(t)]$ with an error at most $\overline{y}_a(t) - \underline{y}_a(t) = \mathcal{O}(T^{-1})$. However, in our setup where the agent keeps update empirical means, simply running binary search does not ensure $\pi_a^\star(t) \notin [\underline{y}_a(t), \overline{y}_a(t)]$. Consider a round $t$ where binary search is used for target arm $a$, and arm $z$ achieves $\widehat{\mu}_z(t) = \max_{b \in [K]} \widehat{\mu}_b(t)$ and $\overline{y}_a(t) - \widehat{\mu}_z(t) = \mathcal{O}(T^{-1})$. If arm $z$ is played at round $t$ generating reward 1, the agent updates $\widehat{\mu}_z(t+1)$, potentially causing $\pi_a^\star(t+1) = \widehat{\mu}_z(t+1) - \widehat{\mu}_a(t+1) > \overline{y}_a(t)$, which pushes $\pi_a^\star(t+1)$ outside the search range. Since the search lasts $\mathcal{O}(\log T)$ rounds, $\widehat{\mu}_z(t)$ increases at most $\mathcal{O}(\log T)$ times. To ensure the estimated incentive is strictly larger than the optimal when search ends, we enlarge $\overline{y}_a(t)$ to account for possible increases, which makes $\frac{2}{\min_{i \in [K]} N_i(t)}$ in Lemma 3.1 becomes $\frac{\log T}{\min_{i \in [K]} N_i(t)}$. As a result, an extra logarithmic multiplication appears in the regret bound.

To address the above issue, we introduce an asymmetric check procedure in binary search. The algorithm tracks $y^{\text{upper}}$, recording the *latest* incentive that successfully enticed agent to play the target arm. When agent does not play the target arm, the algorithm immediately re-offers $y^{\text{upper}}$. If $y^{\text{upper}}$ also fails, the search terminates, indicating that the optimal incentive exceeds the search range. This design quickly detects the out-of-range issue, reducing the error to $\frac{2}{\min_{i \in [K]} N_i(t)}$, which saves a logarithmic factor. Since the check step is triggered only when the incentive fails, we thus call it asymmetric check.

**Enlarged incentive.** Lemma 3.1 ensures $b_{m,a}$ is close to $\pi_a^\star(t)$ when search ends at round $t$. However, there is no guarantee that $b_{m,a}$ can incentivize the agent to play arm

$a$ in subsequent rounds when empirical means are updated. To this end, we enlarge $b_{m,a}$ to $\overline{b}_{m,a}$ given as

$$\overline{b}_{m,a} = \min\left\{1 + \frac{1}{T}, b_{m,a} + 4C_m + Z_m^{-1}\right\}. \quad (6)$$

where $C_m = \sqrt{\frac{\log(4KT/\delta)}{2T_{m-1}}}$. The idea behind this enlargement is to leverage the fact that the rewards are i.i.d. with Hoeffding's inequality which provides an interval for the possible fluctuation of empirical means. Such an interval shrinks as more observations are collected. We use $4C_m$ as an upper bound, accounting for all future fluctuations. Let $\mathcal{T}_{m,a}^E$ be a set of all rounds when Algorithm 1 explores active arm $a$ at line 8 in phase $m$. Then, we have the following.

**Lemma 3.2.** *With probability at least* $1 - \delta$*, for each phase* $m$ *and arm* $a \in \mathcal{A}_m$*, agent plays arm* $a$ *for all* $t \in \mathcal{T}_{m,a}^E$*.*

### 3.3. Necessity of Playing Bad Arms

Since the algorithm only searches for incentives for active arms and Lemma 3.2 ensures each active arm is played for $T_m$ times in phase $m$, $N_a(t)^{-1}$ will be sufficiently small. Thus, it remains to control $(\min_{i \in [K]} N_i(t))^{-1}$ for the estimation error in Lemma 3.1. In fact, this can be achieved by playing bad arms for $Z_m$ times for each phase $m$ where

$$Z_m = \sqrt{|\mathcal{A}_m| \left(\max\{1, |\mathcal{B}_m|\}\right)^{-1} T_{m-1}}, \quad (7)$$

where $T_m$ is defined in Eq. (4).

If one follows the classical phased-elimination framework (Even-Dar et al., 2006), then $\min_{i \in [K]} N_i(t)$ could be exponentially smaller than $T_m$ because the arm that realizes the minimum might have been permanently eliminated in an early phase. As a result, this term may cause linear regret in their framework. In our algorithm, playing all bad arms $Z_m$ times ensures the regret on cumulative estimation errors between $\overline{b}_{m,a}$ and $\pi_a^\star(t)$ is bounded by $\widetilde{\mathcal{O}}(\sqrt{T})$.

### 3.4. Online Elimination

Recall that a bad arm is with small value of $\theta_a + \mu_a$, and thus the algorithm aims to find arms with small value of the corresponding estimate $\widehat{\theta}_a(t) + \widehat{\mu}_a(t)$. Since the empirical means maintained by the agent are unknown, the algorithm cannot eliminate arms in an offline fashion by simply comparing estimators. To this end, the algorithm adopts an online approach to compare, by proposing proper incentives. Specifically, the algorithm tests each active arm $a \in \mathcal{A}_m$ by adding its own estimators $\{\widehat{\theta}_a(t)\}_{a \in [K]}$ (line 11) into the incentive. By doing this, the algorithm can compare their joint estimation $\widehat{\theta}_a(t) + \widehat{\mu}_a(t)$ by observing the played arm. Note that the incentive is increased by one for all active arms so that the comparisons are limited within active arms. If $A_t \neq a$, then $\exists b \in \mathcal{A}_m$ such that $\widehat{\mu}_a(t) + \pi_a(t) \leq \widehat{\mu}_b(t) + \pi_b(t)$ which

leads to $(\widehat{\mu}_b(t) + \widehat{\theta}_b(t)) - (\widehat{\mu}_a(t) + \widehat{\theta}_a(t)) \geq 3 \times 2^{-m}$. Therefore, the algorithm is able to realize the elimination procedure in an online manner.

*Remark* 3.3 (**Offline elimination**). Our algorithm can also do the elimination in an offline manner, and the modified algorithm maintains the same regret bound. We refer readers to Appendix D.8 for more details.

### 3.5. Main Result

The main result is given as follows.

**Theorem 3.4.** *By choosing $\delta = T^{-1}$ for Algorithm 1, we have $\mathbb{E}[R_T] = \mathcal{O}\big(K \log^2 T + \sqrt{KT \log(KT)}\big)$.*

According to the $\Omega(\sqrt{KT})$ lower bound in Remark 2.3, our upper bound matches the lower bound up to a logarithmic factor. Moreover, if one reduces the learning agent to the oracle-agent, our worst-case bound matches that of Scheid et al. (2024b) as long as $T \geq K$. However, our algorithm cannot achieve a gap-dependent bound since it keeps playing bad arms. While Scheid et al. (2024b) show that a gap-dependent bound is attainable in the oracle-agent problem, it remains open whether it is achievable in our setting.

## 4. Exploratory Learning Agent & IID Reward

In this section, we study the problem of an exploratory learning agent in the i.i.d. reward setting, where the agent may arbitrarily select arms other than the empirical maximizer. This problem is initiated by Dogan et al. (2023a), but our setup *generalizes theirs*.

### 4.1. Algorithm Description

At a high-level, Algorithm 5 (see Appendix E.1) is a robust version of Algorithm 1, ensuring the agent exploration does not disrupt the learning process. We maintain main building blocks in Algorithm 1 and introduce new adjustments to enhance the robustness to the agent's exploration behavior.

**Incentive testing.** Recall from Algorithm 1 that to incentivize the agent to play the target active arms, the algorithm needs to accurately estimate optimal incentives by calling Algorithm 3. However, obtaining accurate estimations in this setup becomes challenging because the agent occasionally ignores the incentive to explore an arbitrary arm. To circumvent this issue, we repeat the search for logarithmic times, which leads to the following result.

**Lemma 4.1.** *With probability at least $1 - \delta/4$, $\forall m \geq 2$, among total $2 \log(4 \log_2 T/\delta)$ calls, there is at least one call of Algorithm 3 such that the agent makes no exploration.*

This design leverages the probability amplification (see Lemma E.3): if a random trial succeeds at round $t$ with

---

**Algorithm 2** Informal Version of Algorithm 5 for phase $m$

Play bad arms $\widetilde{\mathcal{O}}\big(T_{m-1}^{2/3}\big)$ times where $T_m$ is given in Eq. (8).
Repeat Algorithm 3 $\mathcal{O}(\log T)$ times for each $a \in \mathcal{A}_m$ to get $\{b_{m,a}^{(i)}\}_{i,a}$ which is sorted in ascending order.
Enlarge $\overline{b}_{m,a}^{(i)} = b_{m,a}^{(i)} + \mathcal{O}(T_m^{-1/3})$ for each $i, a$.

**for** $a \in \mathcal{A}_m$ **do**
    **for** $i = 1, \ldots, \mathcal{O}(\log T)$ **do**   ▷ Incentive testing
        Set a counter $c_{m,a}^{(i)} = 0$ and $Y_{m,a}^{(i)} = 0$.
        **repeat**
            Offer $\pi^0(a; \overline{b}_{m,a}^{(i)})$ and update $Y_{m,a}^{(i)} = Y_{m,a}^{(i)} + 1$
            If agent does not play arm $a$, $c_{m,a}^{(i)} = c_{m,a}^{(i)} + 1$.
        **until** $c_{m,a}^{(i)}$ *exceeds a threshold or* $Y_{m,a}^{(i)} = 2T_m$;
        If $\sum_{j \leq i}(Y_{m,a}^{(j)} - c_{m,a}^{(j)}) \geq T_m$, then break loop $i$.

**for** $a \in \mathcal{A}_m$ **do**   ▷ Trustworthy online elimination
    Set $\mathcal{L}_{m,a} = \emptyset$ and repeat incentive strategy in Algorithm 1 for $\mathcal{O}(\log T)$ times.
    If agent plays target arm $a$, then $\mathcal{L}_{m,a} = \mathcal{L}_{m,a} \cup \{1\}$ and $\mathcal{L}_{m,a} = \mathcal{L}_{m,a} \cup \{0\}$ otherwise. Then, sort $\mathcal{L}_{m,a}$ in ascending order.
    **if** $Median(\mathcal{L}_{m,a}) = 0$ **then**
        Update $\mathcal{A}_{m+1} = \mathcal{A}_m - \{a\}$ and $\mathcal{B}_{m+1} = \mathcal{B}_m \cup \{a\}$.

---

probability $p_t$ (potentially depending on history), repeating it logarithmic times ensures at least one success with high probability. We refer to the call of Algorithm 3 where the agent does not explore, as a "*successful call*". Indeed, the incentive generated by the successful call is close to the optimal one only at that specific round. We further enlarge those incentives by $\mathcal{O}(T_m^{-1/3})$ to force the agent to play the target arm unless exploration occurs.

As the algorithm is unaware of the occurrence of exploration, a successful call cannot be directly identified by the algorithm, even if we know the existence. To address this, the algorithm sorts all estimated incentives $\{\overline{b}_{m,a}^{(i)}\}_i$ in ascending order and tests each of them *iteratively*. We refer to the incentive from a successful call as "*successful incentive*". Such a sorting ensures $(i)$ the error of incentives before the successful incentive is bounded by the error of successful incentive $(ii)$ the iteration for the incentive testing does not exceed that of the successful incentive. It is obvious that $(i)$ holds due to ascending order sorting, and then we will explain the underlying reason that $(ii)$ holds. During the test, the algorithm tracks $c_{m,a}^{(i)}$, the number of times that agent does not play target arm $a$, and $Y_{m,a}^{(i)}$, the total number of rounds in the $i$-th iteration. If $c_{m,a}^{(i)}$ exceeds a threshold, indicating excessive non-target plays, the algorithm believes $\overline{b}_{m,a}^{(i)}$ is inaccurate and tests the next one $\overline{b}_{m,a}^{(i+1)}$. Since a successful incentive guarantees the agent

plays the target arm unless exploring (similar to that of Algorithm 1), and the exploration probability is bounded, we can set a threshold (smaller than $T_m$) ensuring $c_{m,a}^{(i)}$ never exceeds it when testing successful incentive. Consequently, the repeat-loop breaks only when $Y_{m,a}^{(i)} = 2T_m$. If $j$ is the iteration that corresponds to the successful incentive, then $\sum_{j \leq i}(Y_{m,a}^{(j)} - c_{m,a}^{(j)}) \geq T_m$, which implies $(ii)$ holds and the algorithm can acquire at least $T_m$ samples per target arm $a$ (refer to Lemma E.11), where

$$T_m = 32c_0^3 \cdot 2^{2m} K \log(16TK/\delta) \log^2(\iota), \qquad (8)$$

and $\iota = 16KT^2 \log_2 T \log(4 \log_2 T/\delta)\delta^{-1}$.

**Trustworthy online elimination.** The online and offline elimination approaches previously shown in Section 3.4 cannot be directly applied. For the online elimination, the agent's exploration could cause the algorithm to incorrectly eliminate good arms and retain bad ones. The offline elimination (recall Algorithm 4) requires exact knowledge of successful incentive, which is hard to be identified. To tackle this issue, we propose *trustworthy online elimination*. For each active arm $a$, the algorithm offers the same incentive logarithmically many times, observes the agent's action $A_t$, and updates the set $\mathcal{L}_{m,a} = \mathcal{L}_{m,a} \cup \{1\}$ if $A_t = a$ and $\mathcal{L}_{m,a} = \mathcal{L}_{m,a} \cup \{0\}$, otherwise. After gathering enough data, the algorithm sorts $\mathcal{L}_{m,a}$ and uses the median of $\mathcal{L}_{m,a}$ to determine whether to eliminate arm $a$. This median-based approach ensures that the elimination process is robust against the agent's exploration, with high probability.

### 4.2. Main Result

We present the regret bound of Algorithm 5 as follows.

**Theorem 4.2.** *Suppose $T = \Omega(K)$. Choosing $\delta = 1/T$ for Algorithm 5 ensures $\mathbb{E}[R_T] = \mathcal{O}\big((K \log^2 T)^{1/3}T^{2/3}\big)$.*

Recall that Algorithm 1 enlarges the $b_{m,a}$ to $\bar{b}_{m,a}$ by roughly $\mathcal{O}(1/\sqrt{T})$ to ensure the algorithm always successfully incentivizes the agent to play the target active arm. However, due to the exploration behavior, we need to enlarge $b_{m,a}$ further by $\mathcal{O}(T^{-1/3})$ to achieve the goal. This modification requires the algorithm plays each bad arm for $\mathcal{O}(T^{2/3})$ times, compared to $\mathcal{O}(\sqrt{T})$ plays in Algorithm 1, which incurs $\widetilde{\mathcal{O}}(T^{2/3})$ regret bound. Closing the gap between the upper and lower bounds is left for the future work.

### 4.3. Refinement for Exploratory Oracle-Agent

When we reduce the learning behavior in Definition 2.1 to the exploratory oracle-agent considered by (Dogan et al., 2023a), Algorithm 6 (see Appendix F) achieves a $\sqrt{T}$-type regret bound. As the agent selects the true maximizer when not exploring in the reduced setting, we simplify certain algorithmic designs used to handle the agent's learning un-

certainty. For a fair comparison, we adopt $\mathbb{E}[\overline{R}_T]$ (see Appendix A) consistent with (Dogan et al., 2023a).

**Theorem 4.3.** *By choosing $\delta = 1/T$, Algorithm 6 ensures that $\mathbb{E}[\overline{R}_T] = \mathcal{O}\big(\log^2(KT)\sqrt{KT} + K^2 \log^3(KT)\big)$.*

Theorem 4.3 gives a $\widetilde{\mathcal{O}}(\sqrt{KT})$ regret bound, which matches the lower bound up to some logarithmic factors. More importantly, it significantly improves upon $\widetilde{\mathcal{O}}(T^{11/12})$ regret bound in (Dogan et al., 2023a).

## 5. Self-Interested Learning Agent & Linear Reward

In this section, we propose an algorithm (Algorithm 7) for the linear reward setting with the self-interested learning agent. The omitted details in this section can be found in Appendix G. Algorithm 7 is built upon the framework of Algorithm 1. However, to bypass the linear dependence on $K$, the framework needs to be carefully tailored for the linear reward structure. We will show how to modify each component of Algorithm 1 for the linear reward model.

Recall from Algorithm 1 that all arms will played for a fixed number of times in each phase. However, this results in a linear dependence on $K$. To address this, we use $G$-optimal design (Kiefer & Wolfowitz, 1960) a standard technique in the linear bandit literature (Lattimore et al., 2020). It is noteworthy that our algorithm computes a design not only for active arms, but also for bad arms. The reason is similar to that of i.i.d. reward setting, i.e., to acquire accurate estimations of optimal incentives, one needs to stabilize the estimators of bad arms.

Moreover, the algorithm cannot directly apply the elimination approach used in Algorithm 1 to check each arm. This is because the online elimination procedure requires the principal to test each active arm once, but the total number of test scales with $K$. Fortunately, our search algorithm (will be clear in the following subsection) can accurately predict the underlying parameter $s^\star$, which allows to have good estimations of $\{\hat{\mu}_a(t)\}_{a \in \mathcal{A}}$. Consequently, the algorithm conducts the elimination in an offline manner, similar to classical elimination methods.

### 5.1. Robust High Dimensional Search

In the linear reward setting, adopting a similar approach in Algorithm 1 to search for the incentive for each active arm is undesirable, as it incurs a linear dependence on $K$. Therefore, we instead search a space in each phase $m$ in which $s^\star$ resides and all points in this space are close to $s^\star$. Specifically, let $c_m$ be an arbitrary point in this space. It follows that $\langle c_m, a \rangle \approx \langle s^\star, a \rangle$. Furthermore, when the agent collects more samples, her own estimates $\hat{s}_t$ also gets closer to the true parameter $s^\star$, which implies that

$\langle \widehat{s}_t, a \rangle \approx \langle s^\star, a \rangle \approx \langle c_m, a \rangle$ for all arms $a \in \mathcal{A}$. Hence, the optimal incentive $\pi_a^\star(t) = \max_{b \in \mathcal{A}} \langle \widehat{s}_t, b - a \rangle$ can be well-approximated by $\max_{b \in \mathcal{A}} \langle c_m, b - a \rangle$. This idea previously appears in (Scheid et al., 2024b) where the feasible space is refined based on $s^\star$ in the oracle-agent setup, but in our *learning agent* setting, their approach would fail to learn $s^\star$. This is because the space is refined based on the estimate $\widehat{s}_t$ in the learning agent setting, and thus the fluctuation of $\widehat{s}_t$ could mislead the algorithm to wrongly exclude $s^\star$. To this end, we adjust Multiscale Steiner Potential (MSP) (Liu et al., 2021) to cut the space *conservatively*, accounting for the uncertainty caused by the learning agent. Before showing our algorithm, we first define the width for set $S \subseteq \mathbb{R}^d$ and vector $u \in \mathbb{R}^d$ as follows:

$$\texttt{width}(S, u) = \max_{x, y \in S} \langle u, x - y \rangle. \tag{9}$$

Algorithm 8 keeps track of a sequence of confidence sets $\{S_t\}_t$ iteratively. The algorithm initializes $S_1 = B(0, 1)$, ensuring that the unknown parameter $s^\star \in S_1$. At each iteration $t$, Algorithm 8 first picks $a_t^1 - a_t^2$ that maximizes $\texttt{width}(S_t, a_t^1 - a_t^2)$ where $a_t^1, a_t^2 \in \mathcal{A}$, and sets direction $x_t = (a_t^1 - a_t^2) \left\| a_t^1 - a_t^2 \right\|^{-1}$. The idea behind the choices of $a_t^1, a_t^2$ is to enable the algorithm to explore a direction with the maximum uncertainty.

Then, it finds an integer $i$ such that $\texttt{width}(S_t, x_t) \in (2^{-i-1}, 2^{-i}]$, and the length of the interval reflects the uncertainty of $S_t$ along the direction $x_t$. When the if-condition (line 4) is satisfied, the algorithm will return an arbitrary point in $S_t$ because the estimation error for the optimal incentive is up to $\mathcal{O}(\epsilon)$. If the condition is not satisfied, then the algorithm finds a $y_t \in \mathbb{R}$ which reduces $\text{Vol}(S_t + z_i B(0, 1))$ (known as Steiner potential) by half, where the sum is the Minkowski sum

$$S_t + z_i B(0, 1) = \{s + z_i \cdot b : s \in S_t, b \in B(0, 1)\}.$$

Such a cut decreases the volume of $S_t + z_i B(0, 1)$ by a constant multiplicative factor, and thus the total number of iterations is at most $\mathcal{O}(d \log^2(d\epsilon))$ by Lemma G.5.

Since $\langle \widehat{s}_t, a \rangle \leq \|\widehat{s}_t\| \|a\| \leq d$ for all $t \in [T]$, adding $d + \xi$ for $\pi_{a_t^1}(t), \pi_{a_t^2}(t)$ ensures that the agent plays only either $a_t^1$ or $a_t^2$. Here, we use $A_t = a_t^2$ as an example (the other one is analogous) to show why we update $S_{t+1}$ with $\epsilon$ shift. From Eq. (1), the condition $A_t = a_t^2$ implies $\langle \widehat{s}_t, a_t^1 \rangle + \pi_{a_t^1}(t) \leq \langle \widehat{s}_t, a_t^2 \rangle + \pi_{a_t^2}(t)$. Rearranging it and dividing $\left\| a_t^1 - a_t^2 \right\|$ on both sides gives $\langle \widehat{s}_t, x_t \rangle \leq \left\| a_t^1 - a_t^2 \right\|^{-1} y_t$. An natural idea is to update $S_{t+1} = \{v \in S_t : \langle v, x_t \rangle \leq \left\| a_t^1 - a_t^2 \right\|^{-1} y_t\}$ since it ensures $s^\star \in S_{t+1}$ in the oracle-agent setting. However, the estimate $\widehat{s}_t$ varies across time in our learning agent setting, and thus if the algorithm cuts the space based on $\widehat{s}_t$, it is entirely possible that $S_{t+1} = \emptyset$ which invalidates the algorithm. To avoid this issue, we enlarge

the threshold to make the cutting more conservative than before. As a result, the algorithm can guarantee $s^\star \in S_t$ for all $t$. The analysis is deferred to Appendix G.4.

## 5.2. Main Result

We present the regret bound in the following theorem.

**Theorem 5.1.** *Suppose $T = \Omega(d)$. By choosing $\delta = T^{-1}$ in Algorithm 7, we have that $\mathbb{E}[R_T] = \mathcal{O}\left(d^{\frac{4}{3}} T^{\frac{2}{3}} \log^{\frac{2}{3}}(KT)\right)$.*

Theorem 5.1 gives the first sublinear regret bound for linear rewards in the learning agent setting. One may notice that for some $T$, we have $\mathbb{E}[R_T] = \widetilde{\mathcal{O}}(d^{4/3} T^{2/3})$. Unfortunately, the regret bound does not match the lower bound given in Remark 2.3, so there is room for further improvement. Moreover, compared this bound with other $T^{2/3}$-type bound in linear bandits problem (typically $\widetilde{\mathcal{O}}(d^{1/3} T^{2/3})$), our bound suffers an extra multiplicative dependence on $d$. This extra $d$ serves as a cost for the conservative cut. It remains unclear whether this extra $d$ can be removed by more careful analysis or additional adjustment for the search algorithm. We left this problem for future work.

## 6. Conclusion & Future Work

In this paper, we study principal-agent bandit games with self-interested learning and exploratory agents. We first consider the learning agent who greedily chooses the empirical-maximizer and propose Algorithm 1, which achieves a regret bound matching the lower bound up to logarithmic factors. Then, we extend Algorithm 1 to a more general setting in which the agent is allowed to explore an arm different from the empirical maximizer with a small probability, and introduce Algorithm 5 to achieve $\widetilde{\mathcal{O}}(T^{2/3})$ regret bounds. Building on this, we reduce our setting to that of Dogan et al. (2023a) and propose Algorithm 6 with $\widetilde{\mathcal{O}}(\sqrt{T})$ regret, which significantly improves upon $\widetilde{\mathcal{O}}(T^{11/12})$ (Dogan et al., 2023a). Finally, we present Algorithm 7 for the linear reward setting and propose an algorithm with $\widetilde{\mathcal{O}}(d^{4/3} T^{2/3})$ regret bound.

Our results leave several intriguing questions open for further investigation. First, closing the gap between the upper and lower regret bounds in the linear setting, even without agent exploration, remains open. Additionally, it remains unclear whether the $\widetilde{\mathcal{O}}(T^{2/3})$ regret bound in Theorem 4.2 could be further improved to $\mathcal{O}(\sqrt{T})$. Finally, extending our results to handle more general exploration behaviors (e.g., $p_t = \widetilde{\mathcal{O}}(t^{-\alpha})$ for any $\alpha \in (0, 1)$) is a promising direction.

## Acknowledgements

This work was supported in part by an ONR YIP Award (#N00014-20-1-2571) and an NSF CAREER Award (#1844729).

## Impact Statement

This paper presents work whose goal is to advance the field of Machine Learning. There are many potential societal consequences of our work, none which we feel must be specifically highlighted here.

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

## A. Comparisons of Regret Notions and Additional Notations

As the optimal incentive is time-varying in our setting, $R_T$ compares the cumulative utility difference between the per-round optimal arm and the arm selected by the algorithm.

$$R_T = \sum_{t=1}^{T} \left( \max_{a \in \mathcal{A}} \{\theta_a - \pi_a^\star(t)\} - (\theta_{A_t} - \pi_{A_t}(t)) \right). \tag{10}$$

Notice that if one reduces our setting to the *oracle-agent setting* by letting $\widehat{\mu}_a^0 = \mu_a$ and $R_a(t) = \mu_a$ for all $t$, then $\mathbb{E}[R_T] = \mathbb{E}[R_T^{\mathrm{oracle}}]$ where

$$R_T^{\mathrm{oracle}} = \sum_{t=1}^{T} \left( \max_{a \in \mathcal{A}} \left\{ \theta_a - \left( \max_{b \in \mathcal{A}} \mu_b - \mu_a \right) \right\} - (\theta_{A_t} - \pi_{A_t}(t)) \right). \tag{11}$$

In (Scheid et al., 2024b), the principal only proposes one-hot incentives i.e., only one coordinate of $\pi(t)$ is positive at each round, and they define regret as

$$\mathbb{E} \left[ \sum_{t=1}^{T} \left( \max_{a \in \mathcal{A}} \left\{ \theta_a - \left( \max_{b \in \mathcal{A}} \mu_b - \mu_a \right) \right\} - (\theta_{A_t} - \mathbb{I}\{A_t = z_t\} \cdot \pi_{A_t}(t)) \right) \right],$$

where $z_t \in \mathcal{A}$ is an arm that the principal gives positive incentive at round $t$. If we also restrict the principal to always provide one-hot incentives, $\mathbb{E}[R_T^{\mathrm{oracle}}]$ is exactly the same as that of Scheid et al. (2024b). However, if the principal needs to provide incentives for multiple arms, then their notion will be ill-defined.

Dogan et al. (2023a) uses the regret $\mathbb{E}[\overline{R}_T]$ to evaluate the algorithm performance where

$$\overline{R}_T = \sum_{t=1}^{T} \left( \max_{b \in [K]} \{\theta_b + \mu_b\} - \max_{z \in [K]} \mu_z - \left( \theta_{A_t} - \sum_{a \in \mathcal{A}} \pi_a(t) \right) \right). \tag{12}$$

One can observe that $\overline{R}_T \geq R_T^{\mathrm{oracle}}$. Therefore, the regret bound under $\mathbb{E}[\overline{R}_T]$ in (Dogan et al., 2023a) can be directly translated to the regret bound measured by $\mathbb{E}[R_T^{\mathrm{oracle}}]$. In fact, changing the regret measure from $R_T^{\mathrm{oracle}}$ to $\overline{R}_T$ only slightly impacts the low-order terms in the regret bounds of some of our algorithms because our algorithms use one-hot incentives in most of the rounds. For fair comparison, all results in Table 1 are compared via $\mathbb{E}[R_T^{\mathrm{oracle}}]$.

Notice that all our algorithms can get the same regret bound under $\mathbb{E}[R_T^{\mathrm{oracle}}]$ and $\mathbb{E}[R_T]$ without any modification. This is because our algorithm can handle the unknown empirical means, and thus one can assume the agent receives constant reward $R_a(t) = \mu_a$ for all $t, a$.

**Additional Notations.** Based on the definition of $\pi_a^\star(t)$ in Eq. (2), $R_T$ can be rewritten as

$$\begin{aligned}
R_T &= \sum_{t=1}^{T} \left( \max_{a \in \mathcal{A}} \left\{ \theta_a - \left( \max_{b \in \mathcal{A}} \widehat{\mu}_b(t) - \widehat{\mu}_a(t) \right) \right\} - (\theta_{A_t} - \pi_{A_t}(t)) \right) \\
&= \sum_{t=1}^{T} \left( \theta_{a_t^\star} + \widehat{\mu}_{a_t^\star}(t) - \max_{b \in \mathcal{A}} \widehat{\mu}_b(t) - (\theta_{A_t} - \pi_{A_t}(t)) \right),
\end{aligned} \tag{13}$$

where $a_t^\star$ is defined as:

(i.i.d. reward) $\quad a_t^\star \in \underset{a \in \mathcal{A}}{\operatorname{argmax}} \{\theta_a + \widehat{\mu}_a(t)\} \quad$ and $\quad$ (linear reward) $\quad a_t^\star \in \underset{a \in \mathcal{A}}{\operatorname{argmax}} \{\langle \nu^\star, a \rangle + \widehat{\mu}_a(t)\}.$

We further define the gap $\Delta_a := \theta_{a^\star} + \mu_{a^\star} - (\theta_a + \mu_a)$ where

(i.i.d. reward) $\quad a^\star \in \underset{a \in \mathcal{A}}{\operatorname{argmax}} \{\theta_a + \mu_a\} \quad$ and $\quad$ (linear reward) $\quad a^\star \in \underset{a \in \mathcal{A}}{\operatorname{argmax}} \langle \nu^\star + s^\star, a \rangle.$

## B. Discussion on Universal No-Regret Preservation Assumption

We translate the assumption (Scheid et al., 2024a, H2) with our notations in the following.

**Assumption B.1.** There exist $C, \zeta > 0, \kappa \in [0, 1)$ such that for any $s, t \in [T]$ with $s + t \leq T$, any $\{\tau_a\}_{a \in [K]} \in \mathbb{R}_+^K$ and any policy of principal that offers almost surely an incentive $\pi_l = \pi^0(\tilde{a}_l, \tau_{\tilde{a}_l})$ (see Eq. (5)) for any $l \in \{s+1, \ldots, s+t\}$, the batched regret of the agent following policy $\Pi$ satisfies, with probability at least $1 - t^{-\zeta}$,

$$\sum_{l=s+1}^{s+t} \max_{a \in \mathcal{A}} \{\mu_a + \mathbb{I}\{\tilde{a}_l = a\} \cdot \pi_{\tilde{a}_l}\} - (\mu_{A_l} + \mathbb{I}\{A_l = \tilde{a}_l\} \cdot \pi_{\tilde{a}_l}) \leq Ct^\kappa.$$

Since Assumption B.1 requires the no-regret property preservers for any time period from $s + 1$ to $s + t$, and also for any principal's policy, their model does not cover the self-interested learning agent in our paper. Specifically, let us consider two-armed setup with $\mu_1 = 0.1$ and $\mu_2 = 0.9$, and Bernoulli distributions with mean $\mu_1, \mu_2$, respectively. As our model allows the initial values of empirical means are chosen arbitrarily, we assume them to be zero. Suppose that in the first round, the principal incentivizes the agent to play arm 1 and the agent receives reward 1, which occurs with a probability 0.1. Then, we have $\widehat{\mu}_1(t) = 1$ and $\widehat{\mu}_1(t) = 0$. Then, the principal always proposes incentive $\pi^0(2, T^{-10})$ for all rounds $l$ such that $2 \leq l \leq T$. According to the self-interested learning agent behavior $A_t \in \text{argmax}_{a \in \mathcal{A}} \{\widehat{\mu}_a(t) + \pi_a(t)\}$, we have with probability 0.1 (i.e., the probability that agent gets reward 1 at the first round),

$$\sum_{l=s+1}^{s+t} \max_{a \in \mathcal{A}} \{\mu_a + \mathbb{I}\{\tilde{a}_l = a\} \cdot \pi_{\tilde{a}_l}\} - (\mu_{A_l} + \mathbb{I}\{A_l = \tilde{a}_l\} \cdot \pi_{\tilde{a}_l}) = \Omega(t).$$

Then, we show that Assumption B.1 also does not cover the self-interested learning agent with exploration (refer to Definition 2.1). The reason is that in our model (also in (Dogan et al., 2023a)), we consider a general exploration behavior of the agent, in the sense that we do not add any assumption on how the agent explores a non-empirical-maximizer. However, Assumption B.1 implicitly requires the algorithm to strategically explore the environment.

## C. Discussion on Agent's Behavior in (Dogan et al., 2023a)

It is noteworthy that even if an agent's behavior is characterized by the true means (i.e., expected rewards), this does not necessarily require the agent to have the knowledge of those underlying parameters. For example, in Assumption 2 of Dogan et al. (2023a), the authors assume that if the agent does not explore, then the agent must choose a true-maximizer (i.e., the maximizer of the expected reward plus incentive). Dogan et al. (2023a) show that the agent can adopt a learning algorithm without the knowledge of expected rewards to *simulate* that behavior. However, this simulation requires a special coordination between the principal and the agent, in the sense that the agent and principal must *adopt specific algorithms with specific parameters*. In real-world applications, however, it is often infeasible for the principal or the agent to enforce the other party's use of particular algorithms or parameters.

## D. Omitted Proof for I.I.D. Reward in Section 3

### D.1. Discussion on Modifications for General Initial Empirical Means

To avoid clutter in our analysis, we assume for i.i.d. case that the initial empirical means $\{\widehat{\mu}_a^0\}_{a \in [K]}$ are all zeros. For initial empirical means which are selected arbitrarily in $[0, 1]$ and unknown to principal, our algorithms still work with two simple modifications. Note that these modifications work for Algorithm 1, Algorithm 4, Algorithm 5, and Algorithm 6. These two modifications rely on the fact that the mismatch of empirical mean of arm $a$ at round $t$ between assuming it to be zero and an arbitrary one, is at most $\frac{1}{N_a(t)}$. Such a mismatch only causes issue for active arms since every bad arm is played deterministically. Moreover, the number of plays of each active arm $a$ is proportional to phase length, and thus $\frac{1}{N_a(t)} = \Theta(2^{-2m})$ (if $t$ is in phase $m$). For the first modification, when the algorithm aims to incentive the agent to play active arm $a$ at round $t$, one can enlarge the offered incentive by $\frac{1}{N_a(t)}$. Such an enlargement accounts for all possible initial values since the deviation of each arm $a$ is bounded as $\frac{\widehat{\mu}_a^0}{N_a(t)} \leq \frac{1}{N_a(t)}$. As this change is small enough, it does not impact the order of our regret bounds.

---

**Algorithm 3** Noisy binary search with asymmetric check

---

**Input**: target arm $a$, horizon $T$.

**Initialize**: Check $=$ False, $y_a^{\text{upper}} = 1$, $C_1 = C_2 = 0$.

1 Let $t_0$ be the current round, and we set $\overline{y}_a(t_0) = 1$ and $\underline{y}_a(t_0) = 0$.

2 **for** $t = t_0, t_0 + 1 \ldots$ **do**

3     **if** $Check = False$ **then**

4         Set $y_a^{\text{mid}}(t) = \frac{\overline{y}_a(t) + \underline{y}_a(t)}{2}$ and $C_1 = C_1 + 1$.

5         Propose incentives $\pi^0(a; y_a^{\text{mid}}(t))$ and observe $A_t$.

6         **if** $A_t = a$ **then**

7             **if** $C_1 \geq \lceil \log_2 T \rceil$ **then**

8                 Return $y_a^{\text{mid}}(t) + \frac{1}{T}$.

9             Set $y_a^{\text{upper}} = y_a^{\text{mid}}(t)$, $\overline{y}_a(t+1) = y_a^{\text{mid}}(t)$ and $\underline{y}_a(t+1) = \underline{y}_a(t)$.

10         **else**

11             Set Check $=$ True, $\overline{y}_a(t+1) = \overline{y}_a(t)$ and $\underline{y}_a(t+1) = y_a^{\text{mid}}(t)$.

12     **else**

13         Propose incentives $\pi^0(a; y_a^{\text{upper}})$ and observe $A_t$.

14         **if** $A_t = a$ **then**

15             $C_2 = C_2 + 1$.

16             **if** $C_2 = \lceil \log_2 T \rceil$ **then**

17                 Return $y_a^{\text{upper}} + \frac{2}{T}$

18             Set Check $=$ False.

19             Set $\overline{y}_a(t+1) = \overline{y}_a(t)$, and $\underline{y}_a(t+1) = \underline{y}_a(t)$.

20         **else**

21             Return $y_a^{\text{upper}} + \frac{1}{T} + \frac{1}{N_a(t)} + \frac{2}{\min_{i \in [K]} N_i(t)}$.

---

The second change is in elimination period. We take online elimination period as an example. Let $\beta > 0$ be a absolute constant. The algorithm now proposes incentives $\pi(t)$ with $\pi_a(t) = 2 + \widehat{\theta}_a(t) + \beta \cdot 2^{-m} + \frac{1}{N_a(t)}$, $\pi_b(t) = 2 + \widehat{\theta}_b(t) - \frac{1}{N_b(t)}$, $\forall b \in \mathcal{A}_m - \{a\}$, and $\pi_i(t) = 0$, $\forall i \in \mathcal{B}_m$. Notice that the newly added (subtracted) $\frac{1}{N_a(t)}$ ($\frac{1}{N_b(t)}$) is used to account for possible mismatch. Since the elimination process only focuses on active arms, this change may cause $\Theta(2^{-2m})$ inaccuracy in elimination. Then one can adjust the constant $\beta$ to accommodate this change such that our analysis again holds (can be equivalently seen as enlarging the threshold from $\frac{3}{2} \cdot 2^{-m}$ to $\beta 2^{-m}$) and consequently the order of regret bound retains the same. For offline elimination Algorithm 4, one can directly enlarge the threshold to accommodate possible mismatch.

### D.2. Omitted Pseudocode of Algorithm 3

The omitted pseudo-code of our search algorithm can be found in Algorithm 3.

### D.3. Notations

The following are common notations that we adopt in this section. Refer to Appendix A for some general notations.

- Let $\mathcal{T}_m$ be the set of rounds, *excluding the elimination period* that in phase $m$.

- Let $\mathcal{T}_{m,a} = \{t \in \mathcal{T}_m : A_t = a\}$ denote the set of rounds in $\mathcal{T}_m$ such that $A_t = a$.

- Let $\mathcal{T}_{m,a}^E$ be the set of rounds that Algorithm 1 runs in line 8 for arm $a$ in phase $m$. Notice that $\mathcal{T}_{m,a}^E$ excludes those rounds from $\mathcal{T}_{m,a}$ such that Algorithm 3 plays arm $a$.

- Define event $\mathcal{E}$ as follows:

$$\mathcal{E} := \left\{ \forall (t, a) \in [T] \times [K] : \ |\widehat{\mu}_a(t) - \mu_a| \le \sqrt{\frac{\log(4TK/\delta)}{2N_a(t)}}, \ \left|\widehat{\theta}_a(t) - \theta_a\right| \le \sqrt{\frac{\log(4TK/\delta)}{2N_a(t)}} \right\}. \tag{14}$$

### D.4. Proof of Theorem 3.4

The following analyzes conditions on $\mathcal{E}$. In Lemma D.3 below we show that $\mathbb{P}(\mathcal{E}) \ge 1 - \delta$. As the algorithm runs in phases, we bound the regret in each phase $m$. Since the elimination period will last at most $K$ rounds in each phase, the number of phases is at most $\mathcal{O}(\log T)$, and the per-round regret is bounded by an absolute constant, we have

$$R_T \le \mathcal{O}\left(K \log T\right) + \sum_m R_m,$$

where $R_m$ is defined as follows:

$$R_m = \sum_{t \in \mathcal{T}_m} \left( \max_{a \in [K]} \{\theta_a - \pi_a^\star(t)\} - (\theta_{A_t} - \pi_{A_t}(t)) \right).$$

In which follows, we focus on phase $m$ with $m \ge 2$ and $T_m \ne |\mathcal{A}_m| \log T$ since $R_m = \mathcal{O}(K \log T)$ for either phase $m = 1$ or the phase $m$ that satisfies $T_m = |\mathcal{A}_m| \log T$. One can show that

$$
\begin{aligned}
R_m &= \sum_{t \in \mathcal{T}_m} \left( \max_{a \in [K]} \{\theta_a - \pi_a^\star(t)\} - (\theta_{A_t} - \pi_{A_t}(t)) \right) \\
&= \sum_{a \in \mathcal{A}_m} \sum_{t \in \mathcal{T}_{m,a}} \left( \max_{a \in [K]} \{\theta_a - \pi_a^\star(t)\} - (\theta_{A_t} - \pi_{A_t}(t)) \right) \\
&\quad + \sum_{a \in \mathcal{B}_m} \sum_{t \in \mathcal{T}_{m,a}} \left( \max_{a \in [K]} \{\theta_a - \pi_a^\star(t)\} - (\theta_{A_t} - \pi_{A_t}(t)) \right) \\
&\le \sum_{a \in \mathcal{A}_m} \sum_{t \in \mathcal{T}_{m,a}} \left( \max_{a \in [K]} \{\theta_a - \pi_a^\star(t)\} - (\theta_{A_t} - \pi_{A_t}(t)) \right) + \mathcal{O}\left(\sqrt{|\mathcal{A}_m||\mathcal{B}_m|T_m}\right).
\end{aligned}
$$

As Algorithm 3 will last $\mathcal{O}(K \log T)$ rounds and the elimination period will last at most $K$ rounds, and the per-round regret is bounded by a absolute constant, we have that

$$
\begin{aligned}
&\sum_{a \in \mathcal{A}_m} \sum_{t \in \mathcal{T}_{m,a}} \left( \max_{a \in [K]} \{\theta_a - \pi_a^\star(t)\} - (\theta_{A_t} - \pi_{A_t}(t)) \right) \\
&\le \sum_{a \in \mathcal{A}_m} \sum_{t \in \mathcal{T}_{m,a}^E} \left( \max_{a \in [K]} \{\theta_a - \pi_a^\star(t)\} - (\theta_{A_t} - \pi_{A_t}(t)) \right) + \mathcal{O}\left(K \log T\right) \\
&= \sum_{a \in \mathcal{A}_m} \sum_{t \in \mathcal{T}_{m,a}^E} \left( \theta_{a_t^\star} + \widehat{\mu}_{a_t^\star}(t) - \max_{b \in [K]} \widehat{\mu}_b(t) - (\theta_a - \pi_a(t)) \right) + \mathcal{O}\left(K \log T\right),
\end{aligned}
$$

where the last step uses the definition of $a_t^\star$ and $\pi_a^\star(t)$.

Then, for each $a \in \mathcal{A}_m$ and each $t \in \mathcal{T}_{m,a}^E$, we turn to bound

$$
\begin{aligned}
&\theta_{a_t^\star} + \widehat{\mu}_{a_t^\star}(t) - \max_{b \in [K]} \widehat{\mu}_b(t) - (\theta_a - \pi_a(t)) \\
&\le \theta_{a_t^\star} + \mu_{a_t^\star} + \sqrt{\frac{\log(4TK/\delta)}{2N_{a_t^\star}(t)}} - \max_{b \in [K]} \widehat{\mu}_b(t) - (\theta_a - \pi_a(t)) \\
&\le \theta_{a_t^\star} + \mu_{a_t^\star} + \sqrt{\frac{\log(4TK/\delta)}{2T_{m-1}}} - \max_{b \in [K]} \widehat{\mu}_b(t) - (\theta_a - \pi_a(t))
\end{aligned}
$$

$$= \theta_{a_t^*} + \mu_{a_t^*} + \sqrt{\frac{\log(4TK/\delta)}{2T_{m-1}}} - \max_{b \in [K]} \widehat{\mu}_b(t) - (\theta_a - \bar{b}_{m,a})$$

$$\leq \theta_{a^*} + \mu_{a^*} + \sqrt{\frac{\log(4TK/\delta)}{2T_{m-1}}} - \max_{b \in [K]} \widehat{\mu}_b(t) - (\theta_a - \bar{b}_{m,a})$$

$$\leq \mathcal{O}\left(\Delta_a + \sqrt{\frac{\log(KT/\delta)}{T_{m-1}}} + \frac{1}{T} + \frac{\lceil \log T \rceil}{T_{m-1}} + \sqrt{\frac{\max\{1, |\mathcal{B}_m|\}}{T_{m-1}|\mathcal{A}_m|}}\right)$$

$$\leq \mathcal{O}\left(\Delta_a + \sqrt{\frac{\log(KT/\delta)}{1 + N_{A_t}(t)}} + \frac{1}{T} + \frac{\lceil \log T \rceil}{T_{m-1}} + \sqrt{\frac{\max\{1, |\mathcal{B}_m|\}}{T_{m-1}|\mathcal{A}_m|}}\right), \tag{15}$$

where the first inequality holds due to $\mathcal{E}$, the second inequality uses $N_{a_t^*}(t) \geq T_{m-1}$ by Lemma D.13, the equality follows from the fact that for each $m$ and $a \in \mathcal{A}_m$, $\pi_a(t) = \bar{b}_{m,a}$ for all $t \in \mathcal{T}_{m,a}^E$, and the second-to-last inequality uses Lemma D.9 to upper-bound $\bar{b}_{m,a} - \max_{b \in [K]} \widehat{\mu}_b(t)$, and the last inequality holds because $T_m = \Theta(T_{m-1})$ by Lemma D.2 and all active arms in a phase are played for $T_m$ times by Lemma D.8.

By again Lemma D.8, each active arm $a$ in period $\mathcal{T}_{m,a}^E$ will be played for $T_m$ times. Hence, using $|\mathcal{T}_{m,a}^E| = T_m$ and $T_{m+1} = \Theta(T_m)$ for all $m$ from Lemma D.2, we have

$$\sum_{a \in \mathcal{A}_m} \sum_{t \in \mathcal{T}_{m,a}^E} \left(\max_{a \in [K]} \left\{\theta_a + \widehat{\mu}_a(t) - \max_{b \in [K]} \widehat{\mu}_b(t)\right\} - (\theta_a - \pi_a(t))\right)$$

$$\leq \mathcal{O}\left(\sum_{a \in \mathcal{A}_m} \sum_{t \in \mathcal{T}_{m,a}^E} \Delta_a + \sum_{t \in \mathcal{T}_m} \sqrt{\frac{\log(KT/\delta)}{1 + N_{A_t}(t)}} + |\mathcal{A}_m| \left(\frac{T_m}{T} + \log T + \sqrt{\frac{T_m \max\{1, |\mathcal{B}_m|\}}{|\mathcal{A}_m|}}\right)\right) \tag{16}$$

From Lemma D.12, if a suboptimal arm $a$ is active in phase $m$, then $m \leq m_a$ where $m_a$ is the smallest phase such that $\frac{\Delta_a}{2} > 2^{-m_a}$. This implies that

$$\forall a \in \mathcal{A}_m \text{ with } \Delta_a > 0: \quad \frac{\Delta_a}{2} \leq 2^{-(m_a-1)} \leq 2^{-m+1} \leq \mathcal{O}\left(\sqrt{\frac{\log(KT/\delta)}{T_m}}\right), \tag{17}$$

where the reader may recall that our focus is on phase $m \geq 2$ with $T_m \neq |\mathcal{A}_m| \log T$. Hence, we can bound

$$\sum_{t \in \mathcal{T}_{m,a}^E} \sum_{a \in \mathcal{A}_m} \Delta_a \leq \mathcal{O}\left(\sum_{a \in \mathcal{A}_m} \sum_{t \in \mathcal{T}_{m,a}^E} \sqrt{\frac{\log(KT/\delta)}{T_m}}\right)$$

$$\leq \mathcal{O}\left(\sum_{a \in \mathcal{A}_m} \sum_{t \in \mathcal{T}_{m,a}^E} \sqrt{\frac{\log(KT/\delta)}{1 + N_{A_t}(t)}}\right)$$

$$\leq \mathcal{O}\left(\sum_{t \in \mathcal{T}_m} \sqrt{\frac{\log(KT/\delta)}{1 + N_{A_t}(t)}}\right),$$

where the second inequality follows the same reason of Eq. (15). Then, we have

$$\sum_m \sum_{t \in \mathcal{T}_m} \sqrt{\frac{\log(KT/\delta)}{1 + N_{A_t}(t)}} \leq \sum_{t=1}^{T} \sqrt{\frac{\log(KT/\delta)}{1 + N_{A_t}(t)}}$$

$$= \sqrt{\log(KT/\delta)} \sum_{t=1}^{T} \sum_{a \in [K]} \frac{\mathbb{I}\{A_t = a\}}{\sqrt{1 + N_a(t)}}$$

$$= \sqrt{\log(KT/\delta)} \sum_{a \in [K]} \sum_{t=1}^{T} \frac{\mathbb{I}\{A_t = a\}}{\sqrt{1 + \sum_{s=1}^{t-1} \mathbb{I}\{A_s = a\}}}$$

$$\leq \mathcal{O}\left(\sqrt{\log(KT/\delta)} \sum_{a \in [K]} \sqrt{1 + \sum_{t=1}^{T} \mathbb{I}\{A_t = a\}}\right)$$

$$\leq \mathcal{O}\left(\sqrt{KT \log(KT/\delta)}\right),$$

where the first inequality uses Lemma D.1, and the last inequality uses the Cauchy–Schwarz inequality together with the fact $\sum_{a \in [K]} \sum_{t=1}^{T} \mathbb{I}\{A_t = a\} = T$.

By the fact $|\mathcal{A}_m| T_m = \mathcal{O}(|\mathcal{T}_m|)$ and the above results, we have

$$R_T \leq \mathcal{O}\left(K \log^2 T + \sqrt{KT \log(KT/\delta)} + \sum_m \sqrt{|\mathcal{B}_m||\mathcal{A}_m|T_m}\right)$$

$$\leq \mathcal{O}\left(K \log^2 T + \sqrt{KT \log(KT/\delta)} + \sum_m \sqrt{|\mathcal{B}_m||\mathcal{T}_m|}\right)$$

$$\leq \mathcal{O}\left(\sqrt{KT \log(KT/\delta)}\right),$$

where the last inequality first bounds $|\mathcal{B}_m| \leq K$ and then uses Hölder's inequality with the fact that the number of phases is at most $\mathcal{O}(\log T)$. This completes the proof.

### D.5. Auxiliary Lemmas

In this subsection, we present several technical lemmas that are used in the proof of Theorem 3.4.

**Lemma D.1.** *(Pogodin & Lattimore, 2020, Lemma 4.8) Let $\{x_t\}_{t=1}^{T}$ be a sequence with $x_t \in [0, B]$ for all $t \in [T]$. Then, the following estimate holds:*

$$\sum_{t=1}^{T} \frac{x_t}{\sqrt{1 + \sum_{s=1}^{t-1} x_s}} \leq 4\sqrt{1 + \frac{1}{2} \sum_{t=1}^{T} x_t} + B.$$

**Lemma D.2.** *For all $m \in \mathbb{N}$, it is the case that $T_{m+1} = \Theta(T_m)$.*

*Proof.* We consider the following cases for any phase $m$.

1. If $T_m = T_{m+1} = |\mathcal{A}_m| \log T$, then the claim holds.

2. If $T_m = |\mathcal{A}_m| \log T$ and $T_{m+1} \neq |\mathcal{A}_m| \log T$, then $T_m \leq T_{m+1} \leq 4T_m$.

3. If $T_m \neq |\mathcal{A}_m| \log T$ and $T_{m+1} \neq |\mathcal{A}_m| \log T$, the claim again holds as $T_{m+1} = 4T_m$.

Notice that it is impossible for $T_{m+1} = |\mathcal{A}_m| \log T$ and $T_m \neq |\mathcal{A}_m| \log T$. Thus, the proof is complete. $\square$

**Lemma D.3.** *The estimate $\mathbb{P}(\mathcal{E}) \geq 1 - \delta$ holds.*

*Proof.* By Hoeffding's inequality and invoking union bound, one can obtain the desired claim. $\square$

**Lemma D.4.** *For all $t \in [T]$ and $a \in [K]$, we have that*

$$\widehat{\mu}_a(t+1) - \widehat{\mu}_a(t) \geq -\frac{1}{N_a(t+1)}, \tag{18}$$

$$\widehat{\mu}_a(t+1) - \widehat{\mu}_a(t) \leq \frac{1}{N_a(t)}. \tag{19}$$

*Proof.* To show (18), we observe, for all $a \in [K]$, that we have

$$\widehat{\mu}_a(t+1) - \widehat{\mu}_a(t) = \frac{N_a(t) \cdot \widehat{\mu}_a(t) + R_a(t)\mathbb{I}\{A_t = a\}}{N_a(t+1)} - \widehat{\mu}_a(t)$$

$$\geq \frac{N_a(t) \cdot \widehat{\mu}_a(t) + R_a(t)\mathbb{I}\{A_t = a\}}{N_a(t) + 1} - \widehat{\mu}_a(t)$$

$$= \frac{R_a(t)\mathbb{I}\{A_t = a\} - \widehat{\mu}_a(t)}{N_a(t) + 1} \geq -\frac{1}{N_a(t) + 1} \geq -\frac{1}{N_a(t + 1)}.$$

On the other hand, to show (19), observe that

$$\widehat{\mu}_a(t+1) - \widehat{\mu}_a(t) = \frac{N_a(t) \cdot \widehat{\mu}_a(t) + R_a(t)\mathbb{I}\{A_t = a\}}{N_a(t+1)} - \widehat{\mu}_a(t)$$

$$\leq \frac{N_a(t) \cdot \widehat{\mu}_a(t) + R_a(t)\mathbb{I}\{A_t = a\}}{N_a(t)} - \widehat{\mu}_a(t)$$

$$= \frac{R_a(t)\mathbb{I}\{A_t = a\} - \widehat{\mu}_a(t)}{N_a(t)} \leq \frac{1}{N_a(t)}.$$

Hence, the claims hold. $\square$

## D.6. Lemmas for Algorithm 3

In this subsection, we include technical lemmas for Algorithm 3.

**Lemma D.5.** *If Algorithm 3 runs for target arm $a$ and breaks at line 21 at round $t$, then $A_{t-2} = a$.*

*Proof.* We prove the claim by contradiction. Suppose that $A_{t-2} \neq a$. The only possible situation for $A_{t-2} \neq a$ is that Algorithm 3 enters line 10-11, in which the algorithm sets Check = True. Once Check = True is set, we must have $A_{t-1} = a$ and Check = False at round $t - 1$, and otherwise the algorithm will terminate. Notice that Algorithm 3 cannot break at line 21 at round $t$ whether or not the algorithm plays $A_t = a$. This presents a contradiction, and the proof is thus complete. $\square$

**Lemma D.6** (Restatement of Lemma 3.1). *For each phase $m$, if Algorithm 3 ends the search for arm $a$ at round $t$ in phase $m$, then*

$$b_{m,a} - \pi_a^\star(t) \in \left(0, \frac{4}{T} + \frac{\lceil \log_2 T \rceil}{N_a(t)} + \frac{2}{\min_{i \in [K]} N_i(t)}\right], \tag{20}$$

*where $\pi_a^\star(t) = \max_{b \in [K]} \widehat{\mu}_b(t) - \widehat{\mu}_a(t)$. Moreover, Algorithm 3 lasts at most $2\lceil \log_2 T \rceil$ rounds.*

*Proof.* Let us first prove the duration bound. It suffices to consider the worst case that the agent always plays non-target arm (i.e., $A_t \neq a$). Since a single play for non-target arm during line 4-line 11 will cause a check which takes one round and $C_2$ is at most $\lceil \log_2 T \rceil$. Therefore the total number of rounds in Algorithm 3 is $2\lceil \log_2 T \rceil$. Then, we prove the estimation error by considering the following cases.

**Case 1.** Algorithm 3 breaks at line 8 **and** there exists $s \in [t_0, t - 1] \cap \mathbb{N}$ such that $A_s \neq a$. In this case, we have $A_t = a$ and

$$y_a^{\mathrm{mid}}(t) \geq \max_{b \in [K]} \widehat{\mu}_b(t) - \widehat{\mu}_a(t).$$

Let $\tau \in [t_0, t - 1] \cap \mathbb{N}$ be the last round such that $A_\tau \neq a$. Then, $N_{A_\tau}(t) = N_{A_\tau}(\tau) + 1$ holds and

$$\widehat{\mu}_{A_\tau}(t) - \widehat{\mu}_{A_\tau}(\tau) = \frac{N_{A_\tau}(\tau) \cdot \widehat{\mu}_{A_\tau}(\tau) + R_{A_\tau}(\tau)}{N_{A_\tau}(\tau) + 1} - \widehat{\mu}_{A_\tau}(\tau) = \frac{R_{A_\tau}(\tau) - \widehat{\mu}_{A_\tau}(\tau)}{N_{A_\tau}(\tau) + 1} \geq -\frac{1}{N_{A_\tau}(t)}. \tag{21}$$

By definition of $\tau$ and $A_\tau$, we also have

$$\underline{y}_a(t) = y_a^{\mathrm{mid}}(\tau) \leq \max_{b \in [K]} \widehat{\mu}_b(\tau) - \widehat{\mu}_a(\tau) = \widehat{\mu}_{A_\tau}(\tau) - \widehat{\mu}_a(\tau) \tag{22}$$

Then, one can show that

$$0 \leq y_a^{\mathrm{mid}}(t) - \left(\max_{b \in [K]} \widehat{\mu}_b(t) - \widehat{\mu}_a(t)\right)$$

$$\leq y_a^{\mathrm{mid}}(t) - \widehat{\mu}_{A_\tau}(t) + \widehat{\mu}_a(t)$$

$$\leq \frac{\overline{y}_a(t) + \underline{y}_a(t)}{2} - \widehat{\mu}_{A_\tau}(\tau) + \frac{1}{N_{A_\tau}(t)} + \widehat{\mu}_a(\tau) + \frac{\lceil \log_2(T) \rceil}{N_a(\tau)}$$

$$\leq \frac{\overline{y}_a(t) - \underline{y}_a(t)}{2} + \frac{1}{N_{A_\tau}(t)} + \frac{\lceil \log_2(T) \rceil}{N_a(\tau)}$$

$$\leq \frac{1}{T} + \frac{1}{N_{A_\tau}(t)} + \frac{\lceil \log_2(T) \rceil}{N_a(\tau)}$$

$$\leq \frac{1}{T} + \frac{1}{\min_{i \in [K]} N_i(t)} + \frac{\lceil \log_2(T) \rceil}{N_a(\tau)},$$

where the third inequality holds due to Eq. (21) and repeatedly arguing $\widehat{\mu}_a(t) \leq \widehat{\mu}_a(t-1) + \frac{1}{N_a(t-1)} \leq \widehat{\mu}_a(t-2) + \frac{1}{N_a(t-1)} + \frac{1}{N_a(t-2)} \leq \cdots$ from $t$ to $\tau$ (refer to Eq. (19)), the fourth inequality follows from Eq. (22), and the fifth inequality holds by the standard binary search property, i.e., $\overline{y}_a(t) - \underline{y}_a(t)$ exponentially shrinking and the number of shrinks is at least $\lceil \log_2 T \rceil$.

Recall that if Algorithm 3 breaks at line 8, then $b_{m,a} = y_a^{\mathrm{mid}}(t) + \frac{1}{T}$, and Eq. (20) holds.

**Case 2.** Algorithm 3 breaks at line 8 **and** for all $s \in [t_0, t-1] \cap \mathbb{N}$ such that $A_s = a$. In this case, $\underline{y}_a(t) = 0$ and $\overline{y}_a(t) = 2^{1-\lceil \log_2(T) \rceil} \in [\frac{1}{T}, \frac{2}{T}]$.

$$0 \leq y_a^{\mathrm{mid}}(t) - \left( \max_{b \in [K]} \widehat{\mu}_b(t) - \widehat{\mu}_a(t) \right) \leq y_a^{\mathrm{mid}}(t) \leq \frac{1}{T}.$$

Recall that if Algorithm 3 breaks at line 8, then $b_{m,a} = y_a^{\mathrm{mid}}(t) + \frac{1}{T}$, and Eq. (20) holds.

**Case 3.** Algorithm 3 breaks at line 17 **and** $\exists z \in [t_0, t-1] \cap \mathbb{N}$ such that $y_a^{\mathrm{upper}}$ gets updated at round $z$. Let $s \in [t_0, t-1] \cap \mathbb{N}$ be the last round such that $y_a^{\mathrm{upper}}$ gets updated at this round. In this case, we have $A_s = a$ and

$$y_a^{\mathrm{upper}} = y_a^{\mathrm{mid}}(s) = \overline{y}_a(s+1) = \overline{y}_a(t-1) \leq \underline{y}_a(t-1) + \frac{2}{T}, \tag{23}$$

where the last inequality holds since there will be at least $\lceil \log_2 T \rceil$ updates among $\{\underline{y}_a(\tau)\}_\tau$ until the break.

Let $b_{t-1}$ be the arm that such that $\widehat{\mu}_{b_{t-1}}(t-1) = \max_{b \in [K]} \widehat{\mu}_b(t-1)$ and $A_{t-1} = b_{t-1}$. Then,

$$0 \leq y_a^{\mathrm{upper}} - \left( \max_{b \in [K]} \widehat{\mu}_b(t) - \widehat{\mu}_a(t) \right)$$

$$\leq y_a^{\mathrm{mid}}(t-1) + \frac{2}{T} - \max_{b \in [K]} \widehat{\mu}_b(t) + \widehat{\mu}_a(t)$$

$$\leq y_a^{\mathrm{mid}}(t-1) + \frac{2}{T} - \widehat{\mu}_{b_{t-1}}(t) + \widehat{\mu}_a(t)$$

$$\leq y_a^{\mathrm{mid}}(t-1) + \frac{2}{T} - \widehat{\mu}_{b_{t-1}}(t-1) + \frac{1}{N_{b_{t-1}}(t)} + \widehat{\mu}_a(t-1) + \frac{1}{N_a(t-1)}$$

$$\leq \frac{2}{T} + \frac{1}{N_{b_{t-1}}(t)} + \frac{1}{N_a(t-1)},$$

where the first inequality follows from Eq. (23) together with $\underline{y}_a(t-1) \leq y_a^{\mathrm{mid}}(t-1)$, the fourth inequality uses Eq. (18) and Eq. (19), and the last inequality holds since $A_{t-1} = b_{t-1}$.

If Algorithm 3 breaks at line 17, then $b_{m,a} = y_a^{\mathrm{upper}} + \frac{2}{T}$, and thus we have Eq. (20).

**Case 4.** Algorithm 3 breaks at line 17 **and** $\forall z \in [t_0, t-1] \cap \mathbb{N}$, $y_a^{\mathrm{upper}}$ never gets updated. In this case, $y_a^{\mathrm{upper}} = 1$ and $\underline{y}_a(t) = 1 - 2^{1-\lceil \log_2(T) \rceil} \in [1 - \frac{2}{T}, 1 - \frac{1}{T}]$. One can show

$$0 \geq y_a^{\mathrm{upper}} - \left( \max_{b \in [K]} \widehat{\mu}_b(t) - \widehat{\mu}_a(t) \right)$$

$$\geq y_a^{\mathrm{mid}}(t) - \left( \max_{b \in [K]} \widehat{\mu}_b(t) - \widehat{\mu}_a(t) \right)$$

$$\geq 1 - \frac{1}{T} - \left( \max_{b \in [K]} \widehat{\mu}_b(t) - \widehat{\mu}_a(t) \right)$$

$$\geq -\frac{1}{T}.$$

If Algorithm 3 breaks at line 17, then $b_{m,a} = y_a^{\mathrm{upper}} + \frac{2}{T}$, and thus we have Eq. (20).

**Case 5.** Algorithm 3 breaks at line 21. In this case, we have $A_{t-1} \neq a$, $A_t \neq a$. By Lemma D.5, we further have $A_{t-2} = a$. Therefore, one can show

$$0 \geq y_a^{\mathrm{upper}} - \left( \max_{b \in [K]} \widehat{\mu}_b(t) - \widehat{\mu}_a(t) \right)$$

$$= y_a^{\mathrm{upper}} - \widehat{\mu}_{A_t}(t) + \widehat{\mu}_a(t-1)$$

$$\geq y_a^{\mathrm{upper}} - \widehat{\mu}_{A_t}(t-2) - \frac{2}{N_{A_t}(t-2)} + \widehat{\mu}_a(t-2) - \frac{1}{N_a(t-1)}$$

$$\geq y_a^{\mathrm{upper}} - \max_{b \in [K]} \widehat{\mu}_b(t-2) - \frac{2}{N_{A_t}(t-2)} + \widehat{\mu}_a(t-2) - \frac{1}{N_a(t-1)}$$

$$\geq -\frac{2}{N_{A_t}(t-2)} - \frac{1}{N_a(t-1)},$$

where the second inequality uses Eq. (18) and Eq. (19), and the last inequality holds as $A_{t-2} = a$.

If Algorithm 3 breaks at line 21, then $b_{m,a} = y_a^{\mathrm{upper}} + \frac{1}{T} + \frac{1}{N_a(t)} + \frac{2}{\min_{i \in [K]} N_i(t)}$. By noticing $N_a(t-1) \geq N_a(t)$ and $N_{A_t}(t-2) \geq N_{A_t}(t) \geq \min_{i \in [K]} N_i(t)$, we have Eq. (20).

Once Eq. (20) has been proved for phase $m$, Lemma D.8 immediately gives that arm $a$ will be played for the following $T_m$ consecutive rounds. Therefore, the proof is complete. $\square$

**Lemma D.7.** *Suppose that $\mathcal{E}$ occurs. For all $m \geq 2$ and all $a \in \mathcal{A}_m$, if every active arm is played for $T_{m-1}$ times, then we have $A_t = a$ for all $t \in \mathcal{T}_{m,a}^E$.*

*Proof.* Let us consider fixed phase $m \geq 2$ and arm $a \in \mathcal{A}_m \subseteq \mathcal{A}_{m-1}$ and let $t_{m,a}$ be the round that Algorithm 3 ends the search for target arm $a$ in phase $m$. Recall that $\mathcal{T}_{m,a}^E$ is the set of rounds that Algorithm 1 runs in line 8 for arm $a$ in phase $m$. We use strong induction on $t$ in $\mathcal{T}_{m,a}^E$ to prove the claim. For the base case (the first round in $\mathcal{T}_{m,a}^E$)

$$\overline{b}_{m,a} - \pi_a^\star(t_{m,a} + 1)$$

$$= \overline{b}_{m,a} - \min\{1, \pi_a^\star(t_{m,a} + 1)\}$$

$$\geq \overline{b}_{m,a} - \min\left\{ 1, \pi_a^\star(t_{m,a}) + \frac{1}{\min_{i \in [K]} N_i(t)} \right\}$$

$$\geq \overline{b}_{m,a} - \min\left\{ 1, \pi_a^\star(t_{m,a}) + \sqrt{\frac{\max\{1, |\mathcal{B}_m|\}}{T_{m-1} |\mathcal{A}_m|}} \right\}$$

$$= \min\left\{ 1 + \frac{1}{T}, b_{m,a} + 4\sqrt{\frac{\log(4KT/\delta)}{2T_{m-1}}} + \sqrt{\frac{\max\{1, |\mathcal{B}_m|\}}{T_{m-1} |\mathcal{A}_m|}} \right\} - \min\left\{ 1, \pi_a^\star(t_{m,a}) + \sqrt{\frac{\max\{1, |\mathcal{B}_m|\}}{T_{m-1} |\mathcal{A}_m|}} \right\}$$

$$> 0,$$

where the first inequality holds since the one-update shifting is at most $\frac{1}{\min_{i \in [K]} N_i(t)}$, the second inequality holds because the active arm is played for at least $T_{m-1}$ by the assumption and each bad arm is played for at least $\sqrt{\frac{T_{m-1} |\mathcal{A}_m|}{\max\{1, |\mathcal{B}_m|\}}}$ times (by $T_{m-1} \geq |\mathcal{A}_{m-1}| \geq |\mathcal{A}_m|$, we have $\sqrt{\frac{T_{m-1} |\mathcal{A}_m|}{\max\{1, |\mathcal{B}_m|\}}} \leq T_{m-1}$), and the last inequality holds due to Lemma 3.1.

Suppose that arm $a$ gets played for all rounds $\leq t$ in $\mathcal{T}_{m,a}^E$, and then we consider the round $t+1$. To this end, we first show

$$\max_{b\in[K]} \widehat{\mu}_b(t+1) - \max_{b\in[K]} \widehat{\mu}_b(t_{m,a}+1) \leq 2\sqrt{\frac{\log(4KT/\delta)}{2T_{m-1}}}. \tag{24}$$

The induction hypothesis gives that if arm $a$ gets played for all rounds $\leq t$ in $\mathcal{T}_{m,a}^E$, i.e., no bad arms will be played, then for all bad arms $z \in \mathcal{B}_m$, $\widehat{\mu}_z(t+1) = \widehat{\mu}_z(t_{m,a}+1)$. By Hoeffding' inequality, for all active arm $a \in \mathcal{A}_m$ and $\forall \tau \in \mathcal{T}_{m,a}^E : N_a(\tau) \geq T_{m-1}$ and

$$\forall \tau \in \mathcal{T}_{m,a}^E : \quad \widehat{\mu}_a(\tau) \in \left[ \mu_a - \sqrt{\frac{\log(4KT/\delta)}{2T_{m-1}}}, \mu_a + \sqrt{\frac{\log(4KT/\delta)}{2T_{m-1}}} \right]. \tag{25}$$

Therefore, to verify Eq. (24), we only need to consider the two maximums are achieved by two active arms, and their difference is at most $2\sqrt{\frac{\log(4KT/\delta)}{2T_{m-1}}}$.

Hence, one can show

$$\begin{aligned}
&\bar{b}_{m,a} - \left( \max_{b\in[K]} \widehat{\mu}_b(t+1) - \widehat{\mu}_a(t+1) \right) \\
&= \bar{b}_{m,a} - \min\left\{ 1, \max_{b\in[K]} \widehat{\mu}_b(t+1) - \widehat{\mu}_a(t+1) \right\} \\
&\geq \bar{b}_{m,a} - \min\left\{ 1, \max_{b\in[K]} \widehat{\mu}_b(t_{m,a}+1) - \widehat{\mu}_a(t_{m,a}+1) + 4\sqrt{\frac{\log(4KT/\delta)}{2T_{m-1}}} \right\} \\
&\geq \bar{b}_{m,a} - \min\left\{ 1, \max_{b\in[K]} \widehat{\mu}_b(t_{m,a}) - \widehat{\mu}_a(t_{m,a}) + \sqrt{\frac{\max\{1,|\mathcal{B}_m|\}}{T_{m-1}|\mathcal{A}_m|}} + 4\sqrt{\frac{\log(4KT/\delta)}{2T_{m-1}}} \right\} \\
&= \bar{b}_{m,a} - \min\left\{ 1, \pi_a^\star(t_{m,a}) + \sqrt{\frac{\max\{1,|\mathcal{B}_m|\}}{T_{m-1}|\mathcal{A}_m|}} + 4\sqrt{\frac{\log(4KT/\delta)}{2T_{m-1}}} \right\} \\
&> 0,
\end{aligned}$$

where the first inequality uses Eq. (24) and Eq. (25), the second inequality follows from the fact that there will be only one arm played at $t_{m,a}$, and the the per-update error is bounded by $\sqrt{\frac{\max\{1,|\mathcal{B}_m|\}}{T_{m-1}|\mathcal{A}_m|}}$, and the last inequality uses Lemma 3.1 together with the assumption.

This inequality implies that arm $a$ gets played at round $t+1$. Once the induction is done, we get the desired claim for fixed $m, a$. Conditioning on $\mathcal{E}$, the claim holds for all $m, a$, which thus completes the proof. $\qquad \square$

We are now ready to prove Lemma 3.2. Since $\mathbb{P}(\mathcal{E}) \geq 1 - \delta$, we make an equivalent statement as follows.

**Lemma D.8** (Restatement of Lemma 3.2). *Suppose that $\mathcal{E}$ occurs. For each phase $m$ and each active arm $a \in \mathcal{A}_m$, the agent plays arm $a$ for all $t \in \mathcal{T}_{m,a}^E$.*

*Proof.* Let $t_{m,a}$ be the round that Algorithm 3 ends the search for target arm $a$ in phase $m$. We prove the claim by induction on $m$. For $m = 1$, the incentive on each arm $a \in \mathcal{A}$ is $1 + \xi$, then all arms will be played for $T_1$ times. Suppose that the claim holds for $m$ and then we consider for $m + 1$. By induction hypothesis, every active arm $a \in \mathcal{A}_m$ is played for $T_m$ times, then we directly invoking Lemma D.7 to get that $a \in \mathcal{A}_{m+1}$ are played for $T_{m+1}$ times, which completes the induction. Once the induction is done, we get the claim for the fixed $m, a$. Conditioning on $\mathcal{E}$, this argument holds for all $m, a$, thereby completing the proof. $\qquad \square$

**Lemma D.9.** *Suppose that $\mathcal{E}$ occurs. For each phase $m \geq 2$ and each arm $a \in \mathcal{A}_m$, we have*

$$\forall t \in \mathcal{T}_{m,a}^E : \quad \bar{b}_{m,a} \leq \max_{b\in[K]} \widehat{\mu}_b(t) - \mu_a + 7\sqrt{\frac{\log(4KT/\delta)}{2T_{m-1}}} + \frac{4}{T} + \frac{\lceil \log_2 T \rceil}{T_{m-1}} + 5\sqrt{\frac{\max\{1,|\mathcal{B}_m|\}}{T_{m-1}|\mathcal{A}_m|}}.$$

*Proof.* Consider fixed $m, a$. Let $t_{m,a}$ be the round that Algorithm 3 ends the search for arm $a$ at round $m$. For all $t \in \mathcal{T}_{m,a}^E$, one can show

$$
\begin{aligned}
\bar{b}_{m,a} &\leq b_{m,a} + 4\sqrt{\frac{\log(4KT/\delta)}{2T_{m-1}}} + \sqrt{\frac{\max\{1, |\mathcal{B}_m|\}}{T_{m-1}|\mathcal{A}_m|}} \\
&\leq \left( \pi_a^\star(t_{m,a}) + \frac{4}{T} + \frac{\lceil \log_2 T \rceil}{N_a(t_{m,a})} + \frac{2}{\min_{i \in [K]} N_i(t_{m,a})} \right) + 4\sqrt{\frac{\log(4KT/\delta)}{2T_{m-1}}} + \sqrt{\frac{\max\{1, |\mathcal{B}_m|\}}{T_{m-1}|\mathcal{A}_m|}} \\
&\leq \pi_a^\star(t_{m,a}) + \frac{4}{T} + \frac{\lceil \log_2 T \rceil}{T_{m-1}} + 4\sqrt{\frac{\log(4KT/\delta)}{2T_{m-1}}} + 3\sqrt{\frac{\max\{1, |\mathcal{B}_m|\}}{T_{m-1}|\mathcal{A}_m|}} \\
&= \max_{b \in [K]} \widehat{\mu}_b(t_{m,a}) - \widehat{\mu}_a(t_{m,a}) + \frac{4}{T} + \frac{\lceil \log_2 T \rceil}{T_{m-1}} + 4\sqrt{\frac{\log(4KT/\delta)}{2T_{m-1}}} + 3\sqrt{\frac{\max\{1, |\mathcal{B}_m|\}}{T_{m-1}|\mathcal{A}_m|}} \\
&\leq \max_{b \in [K]} \widehat{\mu}_b(t_{m,a}+1) - \widehat{\mu}_a(t_{m,a}+1) + 4\sqrt{\frac{\log(4KT/\delta)}{2T_{m-1}}} + \frac{4}{T} + \frac{\lceil \log_2 T \rceil}{T_{m-1}} + 5\sqrt{\frac{\max\{1, |\mathcal{B}_m|\}}{T_{m-1}|\mathcal{A}_m|}} \\
&\leq \max_{b \in [K]} \widehat{\mu}_b(t) - \mu_a + 7\sqrt{\frac{\log(4KT/\delta)}{2T_{m-1}}} + \frac{4}{T} + \frac{\lceil \log_2 T \rceil}{T_{m-1}} + 5\sqrt{\frac{\max\{1, |\mathcal{B}_m|\}}{T_{m-1}|\mathcal{A}_m|}},
\end{aligned}
$$

where the first inequality holds as the definition of $\bar{b}_{m,a}$ takes the minimum, the second inequality follows from Lemma 3.1, the third inequality bounds $N_a(t_{m,a}) \geq T_{m-1}$ and $\min_{i \in [K]} N_i(tm, a) \geq (\frac{T_{m-1}|\mathcal{A}_m|}{\max\{1, |\mathcal{B}_m|\}})^{1/2}$, the fourth inequality uses Lemma D.4 to bound the per-update error, and the last inequality uses the same reasoning to prove Eq. (24). Conditioning on $\mathcal{E}$, the argument holds for each $m \geq 2$ and $a \in \mathcal{A}_m$, and thus the proof is complete. □

### D.7. Lemmas for Online Elimination

**Lemma D.10.** *Suppose event $\mathcal{E}$ occurs. For all $m \in \mathbb{N}$, $a^\star \in \mathcal{A}_m$ holds.*

*Proof.* We prove the claim by the induction. For $m = 1$, the claim trivially holds. Suppose the claim holds for $m$ and consider for $m + 1$. Let $t_m$ be the round that the algorithm tries to entice arm $a^\star$ in phase $m$ (in line 11). As $\forall a \in \mathcal{A}_m : N_a(t_m) \geq T_m$, we have for each $a \in \mathcal{A}_m$:

$$
0 \leq \theta_{a^\star} + \mu_{a^\star} - (\theta_a + \mu_a) \leq \widehat{\theta}_{a^\star}(t_m) + \widehat{\mu}_{a^\star}(t_m) - (\widehat{\theta}_a(t_m) + \widehat{\mu}_a(t_m)) + 2^{-m},
$$

where the last inequality holds by event $\mathcal{E}$, the induction hypothesis $a^\star \in \mathcal{A}_m$, and Lemma 3.2.

From Algorithm 1, when the algorithm tries to entice arm $a^\star$, $\pi_{a^\star}(t_m) = 1 + \widehat{\theta}_{a^\star}(t_m) + \frac{3}{2} \cdot 2^{-m}$ and $1 + \widehat{\theta}_a(t_m)$ for all $a \in \mathcal{A}_m - \{a^\star\}$. Plugging these into the above gives

$$
0 \leq \pi_{a^\star}(t_m) + \widehat{\mu}_{a^\star}(t_m) - (\pi_a(t_m) + \widehat{\mu}_a(t_m)) - 2^{-m-1},
$$

which gives $\pi_{a^\star}(t_m) + \widehat{\mu}_{a^\star}(t_m) > \pi_a(t_m) + \widehat{\mu}_a(t_m)$ for all $a \in \mathcal{A}_m - \{a^\star\}$, thereby $a^\star \in \mathcal{A}_{m+1}$. Once the induction done, the proof is complete. □

**Lemma D.11.** *Suppose event $\mathcal{E}$ occurs. For each arm $a \in [K]$, if arm $a \in \mathcal{A}_m$ and $a \notin \mathcal{A}_{m+1}$, then $\Delta_a \geq 2^{-m}$.*

*Proof.* Notice that $a \in \mathcal{A}_m$ and $a \notin \mathcal{A}_{m+1}$ imply that the proposed incentive does not successfully entice arm $a$ at the end of phase $m$, thereby being eliminated. Suppose that such an elimination occurs at round $t$. When testing arm $a$ at round $t$, we have $\pi_a(t) = 1 + \widehat{\theta}_a(t) + \frac{3}{2} \cdot 2^{-m}$ and $1 + \widehat{\theta}_b(t)$ for all $b \in \mathcal{A}_m - \{a\}$. Since arm $a$ gets eliminated at round $t$, we have

$$
\begin{aligned}
0 &\leq \max_{j \in \mathcal{A}_m - \{a\}} \{\widehat{\mu}_j(t) + \pi_j(t)\} - (\widehat{\mu}_a(t) + \pi_a(t)) \\
&= \max_{j \in \mathcal{A}_m - \{a\}} \left\{ \widehat{\mu}_j(t) + \widehat{\theta}_j(t) \right\} - \left( \widehat{\mu}_a(t) + \widehat{\theta}_a(t) \right) - \frac{3}{2} \cdot 2^{-m}
\end{aligned}
$$

$$\leq \max_{j \in \mathcal{A}_m} \left\{ \widehat{\mu}_j(t) + \widehat{\theta}_j(t) \right\} - \left( \widehat{\mu}_a(t) + \widehat{\theta}_a(t) \right) - \frac{3}{2} \cdot 2^{-m}.$$

Then, we can further show

$$\begin{aligned}
\frac{3}{2} \cdot 2^{-m} &\leq \max_{j \in \mathcal{A}_m} \left\{ \widehat{\mu}_j(t) + \widehat{\theta}_j(t) \right\} - \left( \widehat{\mu}_a(t) + \widehat{\theta}_a(t) \right) \\
&\leq \max_{j \in \mathcal{A}_m} \left\{ \mu_j + \theta_j + 2^{-m-2} \right\} - \left( \mu_a + \theta_a - 2^{-m-2} \right) \\
&\leq \Delta_a + 2^{-m-1},
\end{aligned}$$

where the second inequality holds due to $\mathcal{E}$ together with Lemma 3.2 and the last inequality holds follows from Lemma D.10 that $a^\star \in \mathcal{A}_m$ for all $m$. Rearranging the above, we obtain the desired claim. □

**Lemma D.12.** *Let $m_a$ be the smallest phase such that $\frac{\Delta_a}{2} > 2^{-m_a}$. Suppose that $\mathcal{E}$ occurs. For each arm $a$ with $\Delta_a > 0$, it will not be in $\mathcal{A}_m$ for all phases $m \geq m_a + 1$.*

*Proof.* Consider any arm $a$ with $\Delta_a > 0$. We only need to consider $a \in \mathcal{A}_{m_a}$ and otherwise, the claim naturally holds. Let $t$ be the round in phase $m$ when the algorithm aims to test if arm $a$ should be active in the next phase. When testing arm $a$ at round $t$, we have $\pi_a(t) = 1 + \widehat{\theta}_a(t) + \frac{3}{2} \cdot 2^{-m}$ and $1 + \widehat{\theta}_b(t)$ for all $b \in \mathcal{A}_m - \{a\}$. Then,

$$\begin{aligned}
&\max_{b \in \mathcal{A}_{m_a} - \{a\}} \left\{ \widehat{\mu}_b(t) + \pi_b(t) \right\} - \left( \widehat{\mu}_a(t) + \pi_a(t) \right) \\
&= \max_{b \in \mathcal{A}_{m_a} - \{a\}} \left\{ \widehat{\mu}_b(t) + \widehat{\theta}_b(t) \right\} - \left( \widehat{\mu}_a(t) + \widehat{\theta}_a(t) \right) - \frac{3}{2} \cdot 2^{-m_a} \\
&\geq \widehat{\mu}_{a^\star}(t) + \widehat{\theta}_{a^\star}(t) - \left( \widehat{\mu}_a(t) + \widehat{\theta}_a(t) \right) - \frac{3}{2} \cdot 2^{-m_a} \\
&\geq \Delta_a - 2^{-m_a - 1} - \frac{3}{2} \cdot 2^{-m_a} \\
&> \Delta_a - 2 \times \frac{\Delta_a}{2} \\
&= 0,
\end{aligned}$$

where the first inequality follows from Lemma D.10 that $a^\star \in \mathcal{A}_{m_a}$ and $\Delta_a > 0$, and the second inequality holds due to $\mathcal{E}$ and Lemma 3.2.

According to the elimination rule (see elimination period in Algorithm 1), arm $a$ will not be in phases $m$ for all $m \geq m_a + 1$. □

**Lemma D.13.** *Let $\mathcal{A}(t)$ be the set of active arms at round $t$. Suppose event $\mathcal{E}$ occurs. For all $t \in [T]$, $a_t^\star \in \mathcal{A}(t)$ holds.*

*Proof.* We prove this by induction. The claim holds trivially at round $t = 1$. Suppose that the claim holds at round $t$ and then consider $t + 1$ round. We then use contradiction to show that $a_{t+1}^\star$ cannot be a bad arm. To this end, we assume $a_{t+1}^\star \notin \mathcal{A}(t+1)$ and then show a contradiction. Assume that round $t$ is in phase $m$ and phase $\tau$ is the last phase such that $a_{t+1}^\star \in \mathcal{A}_\tau$.

Then, we have

$$\begin{aligned}
&\theta_{a_t^\star} + \widehat{\mu}_{a_t^\star}(t+1) - \theta_{a_{t+1}^\star} - \widehat{\mu}_{a_{t+1}^\star}(t+1) \\
&\geq \theta_{a_t^\star} + \widehat{\mu}_{a_t^\star}(t) - \frac{1}{N_{a_t^\star}(t+1)} - \theta_{a_{t+1}^\star} - \widehat{\mu}_{a_{t+1}^\star}(t+1) \\
&\geq \theta_{a^\star} + \widehat{\mu}_{a^\star}(t) - \frac{1}{N_{a_t^\star}(t+1)} - \theta_{a_{t+1}^\star} - \widehat{\mu}_{a_{t+1}^\star}(t+1) \\
&\geq \theta_{a^\star} + \mu_{a^\star} - \sqrt{\frac{\log(4TK/\delta)}{2N_{a^\star}(t)}} - \frac{1}{N_{a_t^\star}(t+1)} - \theta_{a_{t+1}^\star} - \mu_{a_{t+1}^\star} - \sqrt{\frac{\log(4TK/\delta)}{2N_{a_{t+1}^\star}(t+1)}}
\end{aligned}$$

---

**Algorithm 4** Proposed algorithm for i.i.d. reward with offline elimination

---

**Input**: confidence $\delta \in (0,1)$, horizon $T$.

**Initialize**: active arm set $\mathcal{A}_1 = [K]$, bad arm set $\mathcal{B}_1 = \emptyset$, $T_0 = 1$.

1 **for** $m = 1, 2, \ldots$ **do**
2     Set $T_m$ according to Eq. (4).
3     **for** $a \in \mathcal{B}_m$ **do**                 ▷ Stabilize estimators for bad arms
4        Propose incentives $\pi^0(a; 1 + 1/T)$ for $Z_m$ rounds where $Z_m$ is given in Eq. (7).
5     **for** $a \in \mathcal{A}_m$ **do**
6        Invoke Algorithm 3 with input $(a, T)$ to get output $b_{m,a}$.        ▷ Search near-optimal incentive
7        Set $\bar{b}_{m,a}$ based on Eq. (6).
8        Propose incentives $\pi^0(a; \bar{b}_{m,a})$ for $T_m$ rounds.
9     **for** $a \in \mathcal{A}_m$ **do**                                  ▷ Offline elimination
10       Invoke Algorithm 3 again with input $(a, T)$ to get output $b'_{m,a}$ and denote the last round of search by $t_{m,a}$.
11       Let $\{\widehat{\theta}_a(t_{m,a})\}_{a \in [K]}$ be empirical means of all arms at round $t_{m,a}$.
12       Let us define

$$\epsilon_m = \frac{4}{T} + \frac{2 + \lceil \log_2 T \rceil}{T_m} + 2\sqrt{\frac{\mathcal{B}_m}{|\mathcal{A}_m| T_{m-1}}}. \tag{26}$$

13       **if** $\max_{z \in \mathcal{A}_m}\{\widehat{\theta}_z(t_{m,a}) - b'_{m,z}\} - (\widehat{\theta}_a(t_{m,a}) - b'_{m,a}) > \frac{3}{2} \cdot 2^{-m} + \epsilon_m$ **then**
14         Update $\mathcal{A}_{m+1} = \mathcal{A}_m - \{a\}$ and $\mathcal{B}_{m+1} = \mathcal{B}_m \cup \{a\}$.

---

$$
\begin{aligned}
&\geq \theta_{a^\star} + \mu_{a^\star} - (\theta_{a^\star_{t+1}} + \mu_{a^\star_{t+1}}) - \frac{1}{N_{a^\star_t}(t+1)} - 2^{-m-2} - 2^{-\tau-3} \\
&\geq 2^{-\tau} - \frac{1}{N_{a^\star_t}(t+1)} - 2^{-m-2} - 2^{-\tau-3} \\
&> 2^{-\tau} - \frac{2^{-2m}}{32} - 2^{-m-2} - 2^{-\tau-3} \\
&> 0,
\end{aligned}
$$

where the first inequality uses Eq. (18), the second inequality holds as $a^\star_t$ achieves the maximum and Lemma D.10 gives that $a^\star$ is always active, the third inequality follows from the definition of $\mathcal{E}$, the fourth inequality uses $N_{a^\star_{t+1}}(t+1) \geq T_\tau$ and $N_{a^\star}(t) \geq T_{m-1}$ by Lemma D.10, the fifth inequality uses Lemma D.15, the sixth inequality holds since the induction hypothesis gives $a^\star_t \in \mathcal{A}_t = \mathcal{A}_m$, and the last inequality holds since $\tau \leq m$.

The above result forms a contradiction as it does not satisfy the definition of $a^\star_{t+1}$. Thus, the proof is complete.     □

### D.8. Alternative Elimination Approach: Offline Elimination

In this subsection, we show an alternative way to conducting the elimination, and the regret bound of our algorithm maintains the same order. In other words, our algorithm can also be implemented by proposing a one-hot incentive (i.e., only one coordinate of $\pi(t)$ has positive value) similar to that of (Scheid et al., 2024b).

As the other two components remain the same, we then show the counterparts of online elimination lemmas in Appendix D.7 for offline elimination. Before that, we first notice that for $\epsilon_m$ given in Definition 26

$$\frac{4}{T} + \frac{\lceil \log_2 T \rceil}{N_a(t_{m,a})} + \frac{2}{\min_{i \in [K]} N_i(t_{m,a})} \leq \epsilon_m, \tag{27}$$

which can be easily verified by considering two cases $\mathcal{B}_m = \emptyset$ and $\mathcal{B}_m \neq \emptyset$ and using the lower bound on the number of plays.

**Lemma D.14.** *Suppose event $\mathcal{E}$ occurs. For all $m \in \mathbb{N}$, $a^\star \in \mathcal{A}_m$ holds.*

*Proof.* We prove the claim by the induction. For $m = 1$, the claim trivially holds. Suppose the claim holds for $m$ and consider for $m + 1$. As $\forall a \in \mathcal{A}_m : N_a(t_m) \geq T_m$, we have for each $a \in \mathcal{A}_m$:

$$
\begin{aligned}
0 &\leq \theta_{a^\star} + \mu_{a^\star} - (\theta_a + \mu_a) \\
&\leq \widehat{\theta}_{a^\star}(t_{m,a}) + \widehat{\mu}_{a^\star}(t_{m,a}) - (\widehat{\theta}_a(t_{m,a}) + \widehat{\mu}_a(t_{m,a})) + 2^{-m} \\
&= \widehat{\theta}_{a^\star}(t_{m,a}) - \pi^\star_{a^\star}(t_{m,a}) - (\widehat{\theta}_a(t_{m,a}) - \pi^\star_a(t_{m,a})) + 2^{-m} \\
&\leq \widehat{\theta}_{a^\star}(t_{m,a}) - b'_{m,a^\star} - (\widehat{\theta}_a(t_{m,a}) - b'_{m,a}) + 2^{-m} + \underbrace{\frac{4}{T} + \frac{\lceil \log_2 T \rceil}{N_{a^\star}(t_{m,a})} + \frac{2}{\min_{i \in [K]} N_i(t_{m,a})}}_{\leq \epsilon_m},
\end{aligned}
$$

where the second inequality holds by event $\mathcal{E}$, the induction hypothesis $a^\star \in \mathcal{A}_m$, and Lemma 3.2, the last equality adds and subtracts $\max_{b \in [K]} \widehat{\mu}_b(t_{m,a})$ on both sides, and the last inequality uses Lemma 3.1 with Eq. (27). This inequality implies $a^\star \in \mathcal{A}_{m+1}$. Once the induction done, the proof is complete. $\qquad\square$

**Lemma D.15.** *Suppose event $\mathcal{E}$ occurs. For each arm $a \in [K]$, if arm $a \in \mathcal{A}_m$ and $a \notin \mathcal{A}_{m+1}$, then $\Delta_a \geq 2^{-m}$.*

*Proof.* Notice that $a \in \mathcal{A}_m$ and $a \notin \mathcal{A}_{m+1}$ imply that

$$
\begin{aligned}
\frac{3}{2} \cdot 2^{-m} + \epsilon_m &< \max_{j \in \mathcal{A}_m} \left\{ \widehat{\mu}_j(t_{m,a}) - b'_{m,j} \right\} - \left( \widehat{\mu}_a(t_{m,a}) - b'_{m,a} \right) \\
&\leq \max_{j \in \mathcal{A}_m} \left\{ \widehat{\mu}_j(t_{m,a}) - \pi^\star_j(t_{m,a}) \right\} - \left( \widehat{\mu}_a(t_{m,a}) - \pi^\star_a(t_{m,a}) \right) + \epsilon_m \\
&= \max_{j \in \mathcal{A}_m} \left\{ \widehat{\mu}_j(t_{m,a}) + \widehat{\theta}_j(t_{m,a}) \right\} - \left( \widehat{\mu}_a(t_{m,a}) + \widehat{\theta}_a(t_{m,a}) \right) + \epsilon_m \\
&\leq \max_{j \in \mathcal{A}_m} \left\{ \mu_j + \theta_j + 2^{-m-2} \right\} - \left( \mu_a + \theta_a - 2^{-m-2} \right) \\
&\leq \Delta_a + 2^{-m-1} + \epsilon_m,
\end{aligned}
$$

where the second inequality uses Lemma 3.1 with Eq. (27), and the third inequality holds due to $\mathcal{E}$ together with Lemma 3.2 and the last inequality holds follows from Lemma D.10 that $a^\star \in \mathcal{A}_m$ for all $m$. Rearranging the above, we obtain the desired claim. $\qquad\square$

**Lemma D.16.** *Let $m_a$ be the smallest phase such that $\Delta_a > 2^{-m_a+1} + \epsilon_{m_a}$. Suppose that $\mathcal{E}$ occurs. For each arm $a$ with $\Delta_a > 0$, it will not be in $\mathcal{A}_m$ for all phases $m \geq m_a + 1$.*

*Proof.* Consider any arm $a$ with $\Delta_a > 0$. We only need to consider $a \in \mathcal{A}_{m_a}$ and otherwise, the claim naturally holds. One can show

$$
\begin{aligned}
\max_{z \in \mathcal{A}_{m_a}} &\left\{ \widehat{\theta}_z(t_{m_a,a}) - b'_{m_a,z} \right\} - \left( \widehat{\theta}_a(t_{m_a,a}) - b'_{m_a,a} \right) - \frac{3}{2} \cdot 2^{-m_a} \\
&\geq \max_{z \in \mathcal{A}_{m_a}} \left\{ \widehat{\theta}_z(t_{m_a,a}) - \pi^\star_z(t_{m_a,a}) \right\} - \left( \widehat{\theta}_a(t_{m_a,a}) - \pi^\star_a(t_{m_a,a}) \right) - \frac{3}{2} \cdot 2^{-m_a} - \epsilon_{m_a} \\
&= \max_{b \in \mathcal{A}_{m_a}} \left\{ \widehat{\mu}_b(t_{m_a,a}) + \widehat{\theta}_b(t_{m_a,a}) \right\} - \left( \widehat{\mu}_a(t_{m_a,a}) + \widehat{\theta}_a(t_{m_a,a}) \right) - \frac{3}{2} \cdot 2^{-m_a} - \epsilon_{m_a} \\
&\geq \widehat{\mu}_{a^\star}(t_{m_a,a}) + \widehat{\theta}_{a^\star}(t_{m_a,a}) - \left( \widehat{\mu}_a(t_{m_a,a}) + \widehat{\theta}_a(t_{m_a,a}) \right) - \frac{3}{2} \cdot 2^{-m_a} - \epsilon_{m_a} \\
&\geq \Delta_a - 2^{-m_a-1} - \frac{3}{2} \cdot 2^{-m_a} - \epsilon_{m_a} \\
&> 0,
\end{aligned}
$$

where the first inequality uses Lemma 3.1 with Eq. (27), the second inequality follows from Lemma D.10 that $a^\star \in \mathcal{A}_{m_a}$ and the third inequality holds due to $\mathcal{E}$ and Lemma 3.2.

According to the elimination rule, arm $a$ will not be in phases $m$ for all $m \geq m_a + 1$. $\qquad\square$

Now, we will show how these lemmas impact the analysis of Theorem 3.4. These changes mainly affect $\sum_{a \in \mathcal{A}_m} \sum_{t \in \mathcal{T}_{m,a}^E} \Delta_a$ in Eq. (16).

From Lemma D.12, if a suboptimal arm $a$ is active in phase $m$, then $m \le m_a$ where $m_a$ is the smallest phase such that $\Delta_a > 2^{-m_a+1} + \epsilon_{m_a}$. This implies that $\forall a \in \mathcal{A}_m$ with $\Delta_a > 0$ (recall that we focus on phase $m \ge 2$):

$$\frac{\Delta_a}{2} \le 2^{-(m_a-1)} + \epsilon_{m_a-1} \le 2^{-m+1} + \epsilon_{m-1} \le \mathcal{O}\left( \sqrt{\frac{\log(KT/\delta)}{T_m}} + \sqrt{\frac{\max\{1, |\mathcal{B}_m|\}}{T_m |\mathcal{A}_m|}} \right). \tag{28}$$

One can observe that the only difference is an extra $\sqrt{\frac{\max\{1, |\mathcal{B}_m|\}}{T_m |\mathcal{A}_m|}}$ term, which can be handled by the same way as we shown in Appendix D.4. Therefore, the regret bound maintains the same order.

## E. Omitted Proof of Self-interested Learning Agent with Exploration

### E.1. Omitted Pseudocode of Algorithm 5

The omitted pseudocode of our search algorithm can be found in Algorithm 5.

### E.2. Notations

We first introduce some notations used throughout the proof. Refer to Appendix A for some general notations.

- Let $\mathcal{T}^{(i)}(a; \mathcal{A}_m)$ be the set of rounds that the algorithm runs in line 12 for active arm $a \in \mathcal{A}_m$ in the $i$-th iteration of phase $m$.

- Let $\mathcal{T}(a; \mathcal{B}_m)$ be the set of rounds that the algorithm runs in line 4 for bad arm $a$ in phase $m$.

- Let $I_t$ be an indicator that the agent chooses to explore at round $t$.

- Let $j_{m,a}$ be the first index of $\{b_{m,a}^{(i)}\}_i$ such that the agent makes no exploration during Algorithm 3 runs to get $b_{m,a}^{(j_{m,a})}$.

- Let $\mathcal{T}_m$ be the set of all rounds in phase $m$.

- Let $i_{m,a}$ be the last iteration when the algorithm proposes incentive for arm $a$ in phase $m$.

Notice that there exist rounds in $\mathcal{T}^{(i)}(a; \mathcal{A}_m)$ and $\mathcal{T}(a; \mathcal{B}_m)$ such that the agent does not play target arm $a$ due to the agent exploration.

### E.3. Construction of nice event $\mathcal{E}$

Let us define event $\mathcal{E}_0$ as

$$\mathcal{E}_0 := \left\{ \forall (t, a) \in [T] \times [K] : |\widehat{\mu}_a(t) - \mu_a| \le \sqrt{\frac{\log(16TK/\delta)}{2N_a(t)}}, \left|\widehat{\theta}_a(t) - \theta_a\right| \le \sqrt{\frac{\log(16TK/\delta)}{2N_a(t)}} \right\}. \tag{30}$$

**Lemma E.1.** $\mathbb{P}(\mathcal{E}_0) \ge 1 - \delta/4$ *holds.*

*Proof.* By Hoeffding's inequality and invoking union bound, one can obtain the desired claim. $\square$

**Lemma E.2.** *With probability at least $1 - \delta/4$, for all phases $m$ and all bad arms $a \in \mathcal{B}_m$*

$$\sum_{t \in \mathcal{T}(a; \mathcal{B}_m)} I_t \le 2c_0 \sqrt{Z_m \log(2T)} + \sqrt{\frac{8 \log(8KT \log_2(T)\delta^{-1})}{Z_m}}.$$

*and for all phases $m$, all active arms $a \in \mathcal{A}_m$, and all iterations $i$,*

$$\sum_{t \in \mathcal{T}^{(i)}(a; \mathcal{A}_m)} I_t \le 2c_0 \sqrt{Y_{m,a}^{(i)} \log(2T)} + \sqrt{\frac{8 \log(\iota)}{Y_{m,a}^{(i)}}}.$$

---

**Algorithm 5** Proposed algorithm for self-interested learning agent with exploration

---

**Input**: confidences $\delta \in (0, 1)$, horizon $T$.

**Initialize**: active arm set $\mathcal{A}_1 = [K]$, bad arm set $\mathcal{B}_1 = \emptyset$, $T_0 = 1$.

1 **for** $m = 1, 2, \ldots$ **do**

2    Set $T_m$ based on Eq. (8).

3    **for** $a \in \mathcal{B}_m$ **do**                                  ▷ Stabilize estimators for bad arms

4       Propose incentives $\pi^0(a; 1 + T^{-1})$ for $Z_m = 2 \log^{\frac{1}{3}}(16KT/\delta) \left( \frac{|\mathcal{A}_m|}{\max\{1, |\mathcal{B}_m|\}} T_{m-1} \right)^{2/3}$ rounds.

5    **for** $a \in \mathcal{A}_m$ **do**                                     ▷ Search Incentives repeatedly

6       Invoke Algorithm 3 with input $(a, T)$ for $2 \log(4 \log_2 T/\delta)$ times and sort outputs such that

$$b_{m,a}^{(1)} \leq b_{m,a}^{(2)} \leq \cdots \leq b_{m,a}^{(2 \log(4 \log_2 T/\delta))}.$$

7       Set $\bar{b}_{m,a}^{(i)} = \min \left\{ 1 + T^{-1}, b_{m,a}^{(i)} + \left( \frac{\max\{1, |\mathcal{B}_m|\}}{T_{m-1}|\mathcal{A}_m|} \right)^{2/3} + \frac{1}{T_{m-1}} + 4\epsilon_m \right\}$ for all $i$ where $\epsilon_m$ is defined as

$$\epsilon_m = \left( \log(16KT/\delta) \frac{\max\{1, |\mathcal{B}_m|\}}{T_{m-1}|\mathcal{A}_m|} \right)^{1/3} + \sqrt{\frac{\log(16KT/\delta)}{T_{m-1}}}. \tag{29}$$

8    **for** $a \in \mathcal{A}_m$ **do**

9       **for** $i = 1, \ldots, 2 \log(4 \log_2 T/\delta)$ **do**                      ▷ Incentive testing

10          Set a counter $c_{m,a}^{(i)} = 0$ and $Y_{m,a}^{(i)} = 0$.

11          **repeat**

12             Propose incentives $\pi^0(a; \bar{b}_{m,a}^{(i)})$ and denote the current round by $t$.

13             Update $Y_{m,a}^{(i)} = Y_{m,a}^{(i)} + 1$, and if $A_t \neq a$, update $c_{m,a}^{(i)} = c_{m,a}^{(i)} + 1$.

         **until** $c_{m,a}^{(i)} > 2c_0 \sqrt{Y_{m,a}^{(i)} \log(2T)} + \sqrt{\frac{8 \log(\iota)}{Y_{m,a}^{(i)}}}$ **or** $Y_{m,a}^{(i)} = 2T_m$;

14          If $\sum_{j \leq i}(Y_{m,a}^{(j)} - c_{m,a}^{(j)}) \geq T_m$, then break the loop for $i$.

15    **for** $a \in \mathcal{A}_m$ **do**                                    ▷ Trustworthy online elimination

16       Set $\mathcal{L}_{m,a} = \emptyset$.

17       Let $t_{m,a}^0$ be the current round, and $\{\widehat{\theta}_a(t_{m,a}^0)\}_{a \in [K]}$ are empirical means at this round.

18       **for** $t = t_{m,a}^0, \ldots, t_{m,a}^0 + 8 \log(8K \log_2 T/\delta)$ **do**

19          Propose incentives $\pi(t)$ with $\pi_a(t) = 1 + \widehat{\theta}_a(t_{m,a}^0) + 5\sqrt{\frac{\log(16KT/\delta)}{2T_m}}$, $\pi_b(t) = 1 + \widehat{\theta}_b(t_{m,a}^0), \forall b \in \mathcal{A}_m - \{a\}$,

         and $\pi_i(t) = 0, \forall i \in \mathcal{B}_m$.

20          If $A_t \neq a$, then update $\mathcal{L}_{m,a} = \mathcal{L}_{m,a} \cup \{0\}$; else $\mathcal{L}_{m,a} = \mathcal{L}_{m,a} \cup \{1\}$.

21       Sort $\mathcal{L}_{m,a}$ in ascending order.

22       **if** $Median(\mathcal{L}_{m,a}) = 0$ **then**

23          Update $\mathcal{A}_{m+1} = \mathcal{A}_m - \{a\}$ and $\mathcal{B}_{m+1} = \mathcal{B}_m \cup \{a\}$.

---

*Proof.* We first fix a phase $m$ and a bad arm $a \in \mathcal{B}_m$. Then $|\mathcal{T}(a;\mathcal{B}_m)| = Z_m$ is also fixed. Notice that $\mathcal{T}(a;\mathcal{B}_m)$ is a set that contains $Z_m$ consecutive rounds, and thus there are total $T - Z_m + 1$ possible cases. Now, we consider a fixed $\mathcal{T}(a;\mathcal{B}_m)$. By Azuma–Hoeffding's inequality for martingale difference sequence, with probability at least $1 - \delta'$,

$$\sum_{t \in \mathcal{T}(a;\mathcal{B}_m)} I_t \leq \sum_{t \in \mathcal{T}(a;\mathcal{B}_m)} p_t + \sqrt{\frac{8 \log(1/\delta')}{|\mathcal{T}(a;\mathcal{B}_m)|}} \leq 2c_0 \sqrt{Z_m \log(2T)} + \sqrt{\frac{8 \log(1/\delta')}{Z_m}},$$

where the second inequality first bounds $\log(2t) \leq \log(2T)$ and then bounds $\sum_{t \in \mathcal{T}(a;\mathcal{B}_m)} t^{-1/2}$ for $p_t$. By choosing $\delta = \delta'/(8KT \log_2 T)$ and applying a union bound over $m, a$, and all possible sets $\mathcal{T}(a;\mathcal{B}_m)$, the above result holds with probability at least $1 - \delta/8$ for all $m$ and $a \in \mathcal{B}_m$.

Then, one can first fix $m, a, i$, and fixed $\mathcal{T}^{(i)}(a;\mathcal{A}_m)$. Once $\mathcal{T}^{(i)}(a;\mathcal{A}_m)$ fixed, $Y_{m,a}^{(i)} = |\mathcal{T}^{(i)}(a;\mathcal{A}_m)|$ is also fixed. Since $\mathcal{T}^{(i)}(a;\mathcal{A}_m)$ is a set contains consecutive rounds, the number of different $\mathcal{T}^{(i)}(a;\mathcal{A}_m)$ must be smaller than $T^2$. Following a similar reasoning gives with probability at least $1 - \delta/8$, for all $m, a, i, \mathcal{T}^{(i)}(a;\mathcal{A}_m)$:

$$c_{m,a}^{(i)} \leq 2c_0 \sqrt{Y_{m,a}^{(i)} \log(2T)} + \sqrt{\frac{8 \log(16KT^2 \log_2 T \log(4 \log_2 T/\delta)\delta^{-1})}{Y_{m,a}^{(i)}}}.$$

Finally, taking a union bound for the results of bad arms and active arms completes the proof. $\square$

**Lemma E.3.** *Let* $p \in (0,1), \delta' \in (0,1)$. *Consider a sequence of* $r = \log(1/\delta')/p$ *trials. Each trial $t$ has a success probability possibly depending on previous trial outcomes and is lower bounded $p$. Then, with probability at least $1 - \delta'$, there will be at least one success among all these trials.*

*Proof.* As the success probabilities of all trials are uniformly bounded below by a constant $p$, we have

$$\mathbb{P}\left(\text{no success in } r \text{ repetitions}\right) \leq (1-p)^r \leq \exp(-p \cdot r) \leq \exp(-\log(1/\delta')) = \delta',$$

where the second inequality uses $1 + x \leq e^x$ for all $x \in \mathbb{R}$. $\square$

**Lemma E.4** (Restatement of Lemma 4.1). *With probability at least $1 - \delta/4$, for all $m \geq 2$, among total $2 \log(4 \log_2 T/\delta)$ calls, there will be at least one call of Algorithm 3 such that the agent makes no exploration.*

*Proof.* We first consider a fixed phase $m \geq 2$. The probability that no exploration occurs in a single call in phase $m$ is (to save notation, the following product takes over all rounds in this single call)

$$\prod_t (1 - p_t) \geq \left(1 - c_0 \sqrt{\frac{\log(2T_{m-1})}{T_{m-1}}}\right)^{2\lceil \log_2 T \rceil} \geq 1 - 2\lceil \log_2 T \rceil \cdot c_0 \sqrt{\frac{\log(2T)}{T_{m-1}}} \geq \frac{1}{2},$$

where the first inequality lower bounds $p_t$ and uses Lemma E.9 that Algorithm 3 lasts at most $2\lceil \log_2 T \rceil$ rounds, the second inequality follows from the Bernoulli inequality, and the last inequality holds due to $T_{m-1} \geq T_1 \geq 16c_0^2 \log(2T)\lceil \log_2 T \rceil^2$.

By invoking Lemma E.3 with $p = \frac{1}{2}$, we obtain the claim for a fixed phase $m$. Invoking union bound for all $m$ and using the fact that there will be at most $\log_2 T$ phases completes the proof. $\square$

**Lemma E.5.** *With probability at least $1 - \delta/4$, for all phases $m$ and all $a \in \mathcal{A}_m$, if arm $a$ satisfies that for all rounds $t \in [t_{m,a}^0, t_{m,a}^0 + 8\log(8K\delta^{-1}\log_2 T)]$:*

$$\max_{b \in \mathcal{A}_m - \{a\}} (\pi_b(t) + \widehat{\mu}_b(t)) < (\pi_a(t) + \widehat{\mu}_a(t)), \tag{31}$$

*then $a \in \mathcal{A}_{m+1}$; if arm $a$ satisfies that for all rounds $t \in [t_{m,a}^0, t_{m,a}^0 + 8\log(8K\delta^{-1}\log_2 T)]$:*

$$\max_{b \in \mathcal{A}_m - \{a\}} (\pi_b(t) + \widehat{\mu}_b(t)) > (\pi_a(t) + \widehat{\mu}_a(t)), \tag{32}$$

*then $a \in \mathcal{B}_{m+1}$.*

*Proof.* Recall that $I_t$ is an indicator that the agent explores at round $t$. Let us consider a fixed phase $m$ and a fixed active arm $a \in \mathcal{A}_m$. If the arm $a$ satisfies Eq. (31) for all rounds $t \in [t^0_{m,a}, t^0_{m,a} + 8\log(8K\delta^{-1}\log_2 T)]$, then only when $\texttt{Median}(\mathcal{L}_{m,a}) = 0$ holds would mislead the algorithm to deem arm $a$ as bad arm. Let $\mathcal{T}^C_{m,a}$ be the set of rounds when the algorithm enters the elimination period in phase $m$ for arm $a$. Let $\mathbb{E}_t[\cdot] = \mathbb{E}[\cdot|\mathcal{F}_{t-1}]$ and let $L_{t,a} = \{A_t = a, \, a \text{ satisfies Eq. (31)}, t \in \mathcal{T}^C_{m,a}\}$ be an indicator function.

As the elimination period starts at the end of each phase $m$, we can upper bound $p_t \leq c_0\sqrt{\log(2T)/T_m} \leq \frac{1}{4}$ by using $T_m \geq T_1 \geq 16c_0^2\log(2T)$. Then, we have

$$\mathbb{E}_t[L_{t,a}] = 1 - p_t \geq \frac{3}{4}, \text{ which implies } \sum_{t \in \mathcal{T}^C_{m,a}} \mathbb{E}_t[L_{t,a}] \geq 6\log(8K\log_2 T/\delta). \tag{33}$$

Note that $\texttt{Median}(\mathcal{L}_{m,a}) = 0$ implies that $\sum_{t \in \mathcal{T}^C_{m,a}} L_{t,a} \leq 4\log(8K\log_2 T/\delta)$. Then

$$\mathbb{P}\left(\sum_{t \in \mathcal{T}^C_{m,a}} L_{t,a} \leq 4\log(8K\log_2 T/\delta)\right)$$
$$\leq \mathbb{P}\left(\left|\sum_{t \in \mathcal{T}^C_{m,a}} (L_{t,a} - \mathbb{E}_t[L_{t,a}])\right| \geq 2\log(8K\log_2 T/\delta)\right)$$
$$\leq \frac{\delta}{8K\log_2 T},$$

where the last inequality uses the Azuma–Hoeffding inequality for martingale difference sequence. By a union bound for all $a \in \mathcal{A}_m$ and $m$ (with the fact that the total number of phases is $\log_2 T$), with probability $1 - \delta/8$, the claim holds for all active arm in all phases.

Now, if the arm $a$ does not satisfy Eq. (31), then only when $\texttt{Median}(\mathcal{L}_{m,a}) = 1$ holds would mislead the algorithm to deem arm $a$ as an active arm. We reload the definition $L_{t,a} = \{A_t \neq a, \, a \text{ not satisfies Eq. (31)}, t \in \mathcal{T}^C_{m,a}\}$, and with this definition, $L_{t,a}$ satisfies Eq. (33). $\texttt{Median}(\mathcal{L}_{m,a}) = 1$ gives $\sum_{t \in \mathcal{T}^C_{m,a}} L_{t,a} \leq 4\log(8K\log_2 T/\delta)$. Again, we have $\mathbb{E}_t[L_{t,a}] = 1 - p_t \geq \frac{3}{4}$. Then, one can repeat a similar argument to get the claimed result.

Finally, using a union bound over two results completes the proof. $\qquad\square$

*Remark* E.6. It is noteworthy that Lemma E.5 does not provide any guarantee for the arm that may hold the equality in Eq. (31) for some rounds. In fact, since these arms sits on the boundary, it does not affect the analysis whether eliminate them or not.

**Definition E.7** (Define $\mathcal{E}$). Let $\mathcal{E}$ be the event that event $\mathcal{E}_0$ and inequalities in Lemma E.2, Lemma 4.1, and Lemma E.5 hold simultaneously.

Based on Definition E.7, one can easily see $\mathbb{P}(\mathcal{E}) \geq 1 - \delta$, by using a union bound.

### E.4. Proof of Theorem 4.2

The following analysis conditions on $\mathcal{E}$. Since the elimination lasts at most $\mathcal{O}(K\log(K\delta^{-1}\log(T)))$ rounds in each phase, the number of phases is at most $\mathcal{O}(\log T)$, and the per-round regret is bounded by a absolute constant, the total regret during all elimination periods is bounded by

$$\mathcal{O}\left(K\log(T)\log(K\delta^{-1}\log(T))\right).$$

The search algorithm last at most $\mathcal{O}(K\log(T)\log(\delta^{-1}\log T))$ rounds in a single phase and the per-round regret is bounded by an absolute constant. Hence, the total regret during the search is at most

$$\mathcal{O}(K\log^2(T)\log(\delta^{-1}\log T)).$$

The regret in the first phase can be bounded by $\mathcal{O}\left(K\log^3(KT/\delta)\right)$. Thus, we write

$$R_T \le \mathcal{O}\left(K\log^3(KT/\delta)\right) + \sum_{m\ge 2} R_m,$$

where $R_m$ is defined as: (see [Appendix E.2](#) for notations)

$$R_m = \sum_{a\in\mathcal{A}_m}\sum_{i\le i_{m,a}}\sum_{t\in\mathcal{T}^{(i)}(a;\mathcal{A}_m)}\left(\max_{b\in[K]}\{\theta_b - \pi_b^\star(t)\} - (\theta_{A_t} - \pi_{A_t}(t))\right)$$
$$+ \sum_{a\in\mathcal{B}_m}\sum_{t\in\mathcal{T}(a;\mathcal{B}_m)}\left(\max_{b\in[K]}\{\theta_b - \pi_b^\star(t)\} - (\theta_{A_t} - \pi_{A_t}(t))\right).$$

One can show for all $m \ge 2$

$$R_m \le \sum_{a\in\mathcal{A}_m}\sum_{i\le i_{m,a}}\sum_{t\in\mathcal{T}^{(i)}(a;\mathcal{A}_m)}\left(\max_{b\in[K]}\{\theta_b - \pi_b^\star(t)\} - (\theta_{A_t} - \pi_{A_t}(t))\right)$$
$$+ \mathcal{O}\left(\log^{\frac{1}{3}}(KT/\delta)|\mathcal{B}_m|^{\frac{1}{3}}\left(|\mathcal{A}_m|T_{m-1}\right)^{2/3}\right)$$
$$\le \sum_{a\in\mathcal{A}_m}\sum_{i\le i_{m,a}}\sum_{t\in\mathcal{T}^{(i)}(a;\mathcal{A}_m)}\left(\max_{b\in[K]}\{\theta_b - \pi_b^\star(t)\} - (\theta_a - \pi_a(t))\right)\mathbb{I}\{A_t = a\}$$
$$+ \mathcal{O}\left(\sum_{a\in\mathcal{A}_m}\sum_{i\le i_{m,a}}\left(\sqrt{Y_{m,a}^{(i)}\log(T)} + \sqrt{\frac{\log\iota}{Y_{m,a}^{(i)}}}\right)\right) + \mathcal{O}\left(\log^{\frac{1}{3}}(KT/\delta)|\mathcal{B}_m|^{\frac{1}{3}}\left(|\mathcal{A}_m|T_{m-1}\right)^{2/3}\right),$$

where the second inequality uses the repeat-condition. Further, one can bound

$$\sum_{a\in\mathcal{A}_m}\sum_{i\le i_{m,a}}\left(\sqrt{Y_{m,a}^{(i)}\log(T)} + \sqrt{\frac{\log\iota}{Y_{m,a}^{(i)}}}\right)$$
$$\le \mathcal{O}\left(\sum_{a\in\mathcal{A}_m}\sum_{i\le i_{m,a}}\left(\sqrt{T_m\log(T)} + \sqrt{\log\iota}\right)\right)$$
$$\le \mathcal{O}\left(|\mathcal{A}_m|\log(\delta^{-1}\log T)\left(\sqrt{T_m\log(T)} + \sqrt{\log\iota}\right)\right)$$
$$\le \mathcal{O}\left(\log(\delta^{-1}\log T)\left(\sqrt{|\mathcal{A}_m||\mathcal{T}_m|\log(T)} + |\mathcal{A}_m|\sqrt{\log\iota}\right)\right),\tag{34}$$

where the first inequality uses the fact that $Y_{m,a}^{(i)} \le 2T_m$, and the second inequality holds due to the number of iterations is at most $\mathcal{O}(\log(\delta^{-1}\log T))$, and the last inequality uses $|\mathcal{A}_m|T_m \le |\mathcal{T}_m|$.

We continue to bound the total regret of [Eq. (34)](#) as

$$\mathcal{O}\left(\sum_{m\ge 2}\log(\delta^{-1}\log T)\left(\sqrt{|\mathcal{A}_m||\mathcal{T}_m|\log(\iota)}\right)\right) \le \mathcal{O}\left(\log(\delta^{-1}\log T)\left(\sqrt{KT\log T\log(\iota)}\right)\right).$$

Then, for any $a \in \mathcal{A}_m$, $i \le i_{m,a}$ and $t \in \mathcal{T}^{(i)}(a;\mathcal{A}_m)$ with $A_t = a$, we can bound

$$\max_{b\in[K]}\{\theta_b - \pi_b^\star(t)\} - (\theta_a - \pi_a(t))$$
$$= \theta_{a_t^\star} + \widehat{\mu}_{a_t^\star}(t) - \max_{b\in[K]}\widehat{\mu}_b(t) - (\theta_a - \pi_a(t))$$
$$\le \theta_{a_t^\star} + \mu_{a_t^\star} + \sqrt{\frac{\log(16TK/\delta)}{2N_{a_t^\star}(t)}} - \max_{b\in[K]}\widehat{\mu}_b(t) - (\theta_a - \pi_a(t))$$

$$\leq \theta_{a_t^\star} + \mu_{a_t^\star} + \epsilon_m - \max_{b\in[K]} \widehat{\mu}_b(t) - (\theta_a - \pi_a(t))$$

$$= \theta_{a_t^\star} + \mu_{a_t^\star} + \epsilon_m - \max_{b\in[K]} \widehat{\mu}_b(t) - \left(\theta_a - \bar{b}_{m,a}^{(i)}\right)$$

$$\leq \theta_{a^\star} + \mu_{a^\star} + \epsilon_m - \max_{b\in[K]} \widehat{\mu}_b(t) - \left(\theta_a - \bar{b}_{m,a}^{(j_{m,a})}\right)$$

$$\leq \mathcal{O}\left(\Delta_a + \left(\frac{\max\{1, |\mathcal{B}_m|\}}{T_{m-1}|\mathcal{A}_m|}\right)^{2/3} + \epsilon_m\right), \tag{35}$$

where the first inequality holds due to $\mathcal{E}$, the second inequality uses Eq. (39), the third inequality uses the fact that $\bar{b}_{m,a}^{(i)} \leq \bar{b}_{m,a}^{(j_{m,a})}$ for all $i \leq j_{m,a}$ by Lemma E.11, and the last inequality uses Lemma E.12.

Note that line 14 of Algorithm 5 implies that $\sum_{i\leq i_{m,a}} \sum_{t\in\mathcal{T}^{(i)}(a;\mathcal{A}_m)} \mathbb{I}\{A_t = a\} \leq \mathcal{O}(T_m)$. Hence, using $T_{m+1} = \Theta(T_m)$ for all $m$, we have

$$\sum_{a\in\mathcal{A}_m} \sum_{i\leq i_{m,a}} \sum_{t\in\mathcal{T}^{(i)}(a;\mathcal{A}_m)} \left(\max_{b\in[K]} \{\theta_b - \pi_b^\star(t)\} - (\theta_a - \pi_a(t))\right) \mathbb{I}\{A_t = a\}$$

$$\leq \mathcal{O}\left(T_m \sum_{a\in\mathcal{A}_m} \Delta_a + \left(K \log\left(\frac{KT}{\delta}\right)\right)^{1/3} (T_m|\mathcal{A}_m|)^{2/3} + |\mathcal{A}_m|\sqrt{T_m \log\left(\frac{KT}{\delta}\right)}\right)$$

From Lemma E.15, if a suboptimal arm $a$ is active in phase $m$, then $m \leq m_a$ where $m_a$ is the smallest phase such that $\Delta_a > 9\sqrt{\frac{\log(16KT/\delta)}{2T_{m_a}}}$. This implies that

$$\forall a \in \mathcal{A}_m \text{ with } \Delta_a > 0: \quad \Delta_a \leq 9\sqrt{\frac{\log(16KT/\delta)}{2T_{m_a-1}}} \leq 9\sqrt{\frac{\log(16KT/\delta)}{2T_{m-1}}}. \tag{36}$$

Hence, we can bound

$$T_m \sum_{a\in\mathcal{A}_m} \Delta_a \leq \mathcal{O}\left(|\mathcal{A}_m|\sqrt{T_m \log(KT/\delta)}\right). \tag{37}$$

By bounding $T_m|\mathcal{A}_m| \leq |\mathcal{T}_m|$, we can further bound

$$\sum_{m\geq 2} \sum_{a\in\mathcal{A}_m} \sum_{i\leq j_{m,a}} \sum_{t\in\mathcal{T}^{(i)}(a;\mathcal{A}_m)} \left(\max_{b\in[K]} \{\theta_b - \pi_b^\star(t)\} - (\theta_a - \pi_a(t))\right) \mathbb{I}\{A_t = a\}$$

$$\leq \mathcal{O}\left(\sum_{m\geq 2} \left(\left(K \log\left(\frac{KT}{\delta}\right)\right)^{1/3} (|\mathcal{T}_m|)^{2/3} + \sqrt{|\mathcal{A}_m||\mathcal{T}_m| \log\left(\frac{KT}{\delta}\right)}\right)\right)$$

$$\leq \mathcal{O}\left(\left(K \log T \log\left(\frac{KT}{\delta}\right)\right)^{1/3} T^{2/3} + \sqrt{KT \log(T) \log\left(\frac{KT}{\delta}\right)}\right),$$

where the last inequality uses Hölder's inequality with the fact that the number of phases is at most $\mathcal{O}(\log T)$. Combining the above results, we have

$$R_T = \mathcal{O}\left(\left(K \log T \log\left(\frac{KT}{\delta}\right)\right)^{1/3} T^{2/3} + \log^2(KT/\delta)\sqrt{KT} + K \log^3(KT/\delta)\right).$$

By choosing $\delta = 1/T$, we get the claimed bound for $\mathbb{E}[R_T]$.

### E.5. Proof of Theorem 4.3

This proof idea is similar to that of Theorem 4.2. Let us first bound the regret caused by elimination. Notice that since the proposed incentive during trustworthy elimination is *not one-hot*, the regret in a single phase evaluated is bounded by the product of per-round regret upper bound $\mathcal{O}(K)$ and the upper bound of total number of rounds $\mathcal{O}(K \log(K\delta^{-1} \log(T)))$. As the number of phases is at most $\mathcal{O}(\log T)$, the total regret incurred by the elimination is bounded by

$$\mathcal{O}\left(K^2 \log T \log(K\delta^{-1} \log(T))\right).$$

Then, we bound the regret incurred by searching. Since the algorithm proposes one-hot incentive, and the search process lasts $\mathcal{O}(\log T)$ rounds for each active arm in each phase, the total regret incurred by searching is bounded by

$$\mathcal{O}(K \log^2(T) \log(\delta^{-1} \log T)).$$

Moreover, the total regret in phase $m = 1$ is bounded by

$$\mathcal{O}\left(K^2 \log^3(KT/\delta)\right).$$

Note that since the algorithm always proposes the one-hot incentive, except elimination period, we thus can write

$$\overline{R}_T = \mathcal{O}\left(K^2 \log^3(KT/\delta)\right) + \sum_{m \geq 2} \overline{R}_m,$$

where

$$\begin{aligned}
\overline{R}_m &= \sum_{a \in \mathcal{A}_m} \sum_{i \leq i_{m,a}} \sum_{t \in \mathcal{T}^{(i)}(a;\mathcal{A}_m)} \left( \max_{b \in [K]} \{\theta_b + \mu_b\} - \max_{z \in [K]} \mu_z - \left( \theta_{A_t} - \sum_{v \in [K]} \pi_v(t) \right) \right) \\
&\quad + \sum_{a \in \mathcal{B}_m} \sum_{t \in \mathcal{T}(a;\mathcal{B}_m)} \left( \max_{b \in [K]} \{\theta_b + \mu_b\} - \max_{z \in [K]} \mu_z - \left( \theta_{A_t} - \sum_{v \in [K]} \pi_v(t) \right) \right) \\
&= \sum_{a \in \mathcal{A}_m} \sum_{i \leq i_{m,a}} \sum_{t \in \mathcal{T}^{(i)}(a;\mathcal{A}_m)} \left( \max_{b \in [K]} \{\theta_b + \mu_b\} - \max_{z \in [K]} \mu_z - (\theta_{A_t} - \pi_{A_t}(t)) \right) \\
&\quad + \sum_{a \in \mathcal{B}_m} \sum_{t \in \mathcal{T}(a;\mathcal{B}_m)} \left( \max_{b \in [K]} \{\theta_b + \mu_b\} - \max_{z \in [K]} \mu_z - (\theta_{A_t} - \pi_{A_t}(t)) \right),
\end{aligned}$$

where the last equality follows from the fact that the proposed incentives are one-hot during $\mathcal{T}(a;\mathcal{B}_m)$ and $\mathcal{T}^{(i)}(a;\mathcal{A}_m)$.

The analysis on regret in phases $m \geq 2$ is a simplified version of the proof of Theorem 4.2. The only difference is to use Lemma E.13 to obtain Eq. (35). Thus, the regret bound for $\sum_{m \geq 2} R_m$ remains the same.

### E.6. Supporting Lemmas

**Lemma E.8.** *Suppose $\mathcal{E}$ holds. For each phase $m$ with $\mathcal{B}_m \neq \emptyset$, every bad arm $a \in \mathcal{B}_m$ will be played for at least* $\log^{\frac{1}{3}}(16KT/\delta) \left( \frac{|\mathcal{A}_m|}{\max\{1,|\mathcal{B}_m|\}} T_{m-1} \right)^{2/3}$ *times.*

*Proof.* Let $I_t$ be an indicator that the agent explores at round $t$. Consider a fixed phase $m$ with $\mathcal{B}_m \neq \emptyset$ (i.e., $m \geq 2$), and we fix a bad arm $a \in \mathcal{B}_m$. Lemma E.2 gives

$$\sum_{t \in \mathcal{T}(a;\mathcal{B}_m)} I_t \leq 2c_0 \sqrt{Z_m \log(2T)} + \sqrt{\frac{8 \log(8KT \log_2(T)\delta^{-1})}{Z_m}}.$$

Notice that if the agent does not explore at a round, then she must play the target arm $a$ based on the proposed incentive. Thus, it suffices to show

$$Z_m - \left( 2c_0 \sqrt{Z_m \log(2T)} + \sqrt{\frac{8\log(8KT\log_2(T)\delta^{-1})}{Z_m}} \right) \geq \frac{Z_m}{2}.$$

To this end, we let

$$a = \max\{2c_0\sqrt{\log(2T)}, \sqrt{8\log(8KT\log_2(T)/\delta)}\}.$$

Then, it suffices to show $Z_m - 2a(\sqrt{Z_m} + 1/\sqrt{Z_m}) \geq 0$. One can easily show that for $a \geq 1$, the function $f(x) = x - 2a(\sqrt{x} + 1/\sqrt{x})$ is monotonically increasing for $x \geq a^2$ and $f(16a^2) \geq 0$. Then, we verify $Z_m \geq 16a^2$ as:

$$\frac{Z_m}{2} = \log^{\frac{1}{3}}(16KT/\delta) \left( \frac{|\mathcal{A}_m|}{\max\{1, |\mathcal{B}_m|\}} T_{m-1} \right)^{2/3} \geq \left( \frac{T_1}{K} \right)^{2/3} \geq 8a^2.$$

We thus complete the proof. □

**Lemma E.9.** *Let* $t_{m,a}^{(j_{m,a})}$ *be the round that [Algorithm 3](#) ends the search for arm $a$ in phase $m$ and outputs $b_{m,a}^{(j_{m,a})}$. For each phase $m$, we have*

$$b_{m,a}^{(j_{m,a})} - \pi_a^\star(t_{m,a}^{(j_{m,a})}) \in \left( 0, \frac{4}{T} + \frac{\lceil \log_2 T \rceil}{N_a(t_{m,a}^{(j_{m,a})})} + \frac{2}{\min_{i \in [K]} N_i(t_{m,a}^{(j_{m,a})})} \right]. \tag{38}$$

*Proof.* Since in $j_{m,a}$-th iteration, the agents does not explore, the proof directly follows [Lemma 3.1](#). □

**Lemma E.10.** *Suppose that $\mathcal{E}$ occurs. For all $m \geq 2$ and $a \in \mathcal{A}_m$, if every active arm in $\mathcal{A}_{m-1}$ is played for $T_{m-1}$ times, then we have*

$$\forall t \in \bigcup_{i \leq j_{m,a}} \mathcal{T}_{m,a}^{(i)} : \quad \bar{b}_{m,a}^{(j_{m,a})} > \pi_a^\star(t).$$

*Proof.* Let us consider fixed phase $m \geq 2$ and arm $a \in \mathcal{A}_m \subseteq \mathcal{A}_{m-1}$ and let $t_{m,a}^{(j_{m,a})}$ be the round that [Algorithm 3](#) ends the search for arm $a$ at round $m$ and outputs $b_{m,a}^{(j_{m,a})}$.

$$\sqrt{\frac{\log(16KT/\delta)}{2\min_{i \in [K]} N_i(t)}} \leq \sqrt{\frac{\log(16KT/\delta)}{2\min\left\{ T_{m-1}, \log^{\frac{1}{3}}(16KT/\delta)\left( \frac{|\mathcal{A}_m|}{\max\{1,|\mathcal{B}_m|\}} T_{m-1} \right)^{2/3} \right\}}}$$

$$\leq \sqrt{\frac{\log(16KT/\delta)}{2T_{m-1}}} + \sqrt{\frac{\log(16KT/\delta)}{2T_{m-1}, \log^{\frac{1}{3}}(16KT/\delta)\left( \frac{|\mathcal{A}_m|}{\max\{1,|\mathcal{B}_m|\}} T_{m-1} \right)^{2/3}}}$$

$$\leq \epsilon_m \tag{39}$$

where the first inequality uses the assumption (i.e., the number of plays of active arms) and [Lemma E.8](#), and the last inequality holds due to the definition of $\epsilon_m$ given in [Eq. (29)](#).

By Hoeffding' inequality and the fact that the confidence interval is monotonically decreasing as samples increase, we have for all $b \in \mathcal{A}$ and $\forall t \in \bigcup_{i \leq j_{m,a}} \mathcal{T}_{m,a}^{(i)}$ (here $a$ is the fixed arm mentioned above)

$$\widehat{\mu}_b(t) \in \left[ \mu_b - \sqrt{\frac{\log(16KT/\delta)}{2N_b(t)}}, \mu_b + \sqrt{\frac{\log(16KT/\delta)}{2N_b(t)}} \right]$$

$$\subseteq \left[ \mu_b - \sqrt{\frac{\log(16KT/\delta)}{2\min_{i \in [K]} N_i(t)}}, \mu_b + \sqrt{\frac{\log(16KT/\delta)}{2\min_{i \in [K]} N_i(t)}} \right]$$

$$\subseteq [\mu_b - \epsilon_m, \mu_b + \epsilon_m], \tag{40}$$

where the last inequality uses Eq. (39). Let $b_t \in \operatorname{argmax}_{a \in \mathcal{A}} \widehat{\mu}_a(t)$. Note that Eq. (40) gives that

$$\forall t \in \bigcup_{i \leq j_{m,a}} \mathcal{T}_{m,a}^{(i)}: \quad \max_{b \in [K]} \widehat{\mu}_{b_t}(t) = \widehat{\mu}_{b_t}(t) \leq \widehat{\mu}_{b_t}(t_{m,a}^{(j_{m,a})} + 1) + 2\epsilon_m \leq \max_{b \in [K]} \widehat{\mu}_b(t_{m,a}^{(j_{m,a})} + 1) + 2\epsilon_m. \tag{41}$$

Hence, following a similar reasoning in Lemma D.7, one can show $\forall t \in \bigcup_{i \leq j_{m,a}} \mathcal{T}_{m,a}^{(i)}$:

$$\begin{aligned}
\bar{b}_{m,a}^{(j_{m,a})} &- \left( \max_{b \in [K]} \widehat{\mu}_b(t) - \widehat{\mu}_a(t) \right) \\
&= \bar{b}_{m,a}^{(j_{m,a})} - \min \left\{ 1, \max_{b \in [K]} \widehat{\mu}_b(t) - \widehat{\mu}_a(t) \right\} \\
&\geq \bar{b}_{m,a}^{(j_{m,a})} - \min \left\{ 1, \max_{b \in [K]} \widehat{\mu}_b(t_{m,a}^{(j_{m,a})} + 1) - \widehat{\mu}_a(t_{m,a}^{(j_{m,a})} + 1) + 4\epsilon_m \right\} \\
&\geq \bar{b}_{m,a}^{(j_{m,a})} - \min \left\{ 1, \max_{b \in [K]} \widehat{\mu}_b(t_{m,a}^{(j_{m,a})}) - \widehat{\mu}_a(t_{m,a}^{(j_{m,a})}) + \left( \frac{\max\{1, |\mathcal{B}_m|\}}{T_{m-1}|\mathcal{A}_m|} \right)^{2/3} + \frac{1}{T_{m-1}} + 4\epsilon_m \right\} \\
&> 0,
\end{aligned}$$

which implies that arm $a$ gets played at round $t + 1$. Once the induction is done, we get the desired claim for fixed $m, a$. Conditioning on $\mathcal{E}$, the claim holds for all $m, a$, which thus completes the proof. $\square$

**Lemma E.11.** *Suppose that $\mathcal{E}$ occurs. For each phase $m$, each active arm $a \in \mathcal{A}_m$ will be played for at least $T_m$ times before trustworthy online elimination starts. Moreover, for each phase $m$, active arm $a \in \mathcal{A}_m$, we have $i_{m,a} \leq j_{m,a}$ and $\bar{b}_{m,a}^{(i)} \leq \bar{b}_{m,a}^{(j_{m,a})}$ for all $i \leq j_{m,a}$.*

*Proof.* Before that, we first show for all $m \geq 1$

$$2T_m - \left( 2c_0 \sqrt{2T_m \log(2T)} + \sqrt{\frac{8 \log(\iota)}{T_m}} \right) \geq T_m. \tag{42}$$

For shorthand, we define

$$a = \max \left\{ 2c_0 \sqrt{2 \log(2T)}, \sqrt{8 \log \iota} \right\}.$$

To show Eq. (42), it suffices to show $T_m - a(\sqrt{T_m} + 1/\sqrt{T_m}) \geq 0$. One can easily show that for $a \geq 2$, the function $f(x) = x - a(\sqrt{x} + 1/\sqrt{x})$ is monotonically increasing for $x \geq a^2/4$ and $f(4a^2) \geq 0$. As we have $T_m \geq T_1 \geq 4a^2$, Eq. (42) holds for all $m \geq 1$.

Now, we start to prove the first claim by using induction on $m$. For the base case, one can see that the incentive on every arm is always $1 + T^{-1}$, which implies that if the agent does not explore, then she must play arm $a$. Thus, by Lemma E.2, in the first iteration, the repeat-loop only breaks when $Y_{1,a}^{(1)} = 2T_1$. By Eq. (42), for every arm $a$, the number of plays of each $a$ is at least $T_1$ times. Therefore, the base case holds.

Suppose that the claim holds for phase $m - 1$. Then, we aim to prove for phase $m$ and consider a fixed $a \in \mathcal{A}_m$. Assume that the number of plays for arm $a$ is less than $T_m$ before the $j_{m,a}$-th iteration (otherwise the claim holds true). We will show that the number of plays of arm $a$ in the $j_{m,a}$-th iteration is at least $T_m$ times. Based on Lemma E.10, we know that in the $j_{m,a}$-th iteration (the exploration does not occur when generate corresponding incentive), the agent will always play the target arm $a$. Thus, by Lemma E.2, in $j_{m,a}$-th iteration, the repeat-loop only breaks when $Y_{m,a}^{(j_{m,a})} = 2T_m$. By Eq. (42), for every arm $a$, the number of plays of each $a$ is at least $T_m$ times. Once the induction is done, we complete the proof for the first claim.

For the second claim, we consider fixed $m, a$. Since the above shows that the number of plays on $a$ is at least $T_m$ in the $j_{m,a}$-th iteration. According to line 14 of Algorithm 5, the iteration number cannot go beyond $j_{m,a}$. Moreover, $\{b_{m,a}^{(i)}\}_i$ is

assorted in ascending order, and thus $b_{m,a}^{(i)} \leq b_{m,a}^{(j_{m,a})}$ for all $i \leq j_{m,a}$. Conditioning on $\mathcal{E}$, we repeat this argument for all $m, a$ to complete the proof. $\qquad \square$

**Lemma E.12.** *Suppose that $\mathcal{E}$ occurs. For each phase $m \geq 2$ and each arm $a \in \mathcal{A}_m$, we have*

$$\forall t \in \bigcup_{i \leq j_{m,a}} \mathcal{T}^{(i)}(a; \mathcal{A}_m) : \overline{b}_{m,a}^{(j_{m,a})} \leq \max_{b \in [K]} \widehat{\mu}_b(t) - \mu_a + \frac{4}{T} + \frac{\lceil \log_2 T \rceil}{T_{m-1}} + 4 \left( \frac{\max\{1, |\mathcal{B}_m|\}}{T_{m-1}|\mathcal{A}_m|} \right)^{2/3} + \frac{4}{T_{m-1}} + 5\epsilon_m.$$

*Proof.* Consider fixed $m, a$. Let $t_{m,a}^{(j_{m,a})}$ be the round that Algorithm 3 ends the search for arm $a$ at round $m$, which outputs $b_{m,a}^{(j_{m,a})}$. For all $t \in \cup_{i \leq j_{m,a}} \mathcal{T}^{(i)}(a; \mathcal{A}_m)$, we use a similar reasoning of Lemma D.9 with counterparts Lemma E.11, Lemma E.8 and Eq. (40) to show

$$
\begin{aligned}
\overline{b}_{m,a}^{(j_{m,a})} &\leq b_{m,a}^{(j_{m,a})} + \left( \frac{\max\{1, |\mathcal{B}_m|\}}{T_{m-1}|\mathcal{A}_m|} \right)^{2/3} + \frac{1}{T_{m-1}} + 4\epsilon_m \\
&\leq \pi_a^\star(t_{m,a}^{(j_{m,a})}) + \frac{4}{T} + \frac{\lceil \log_2 T \rceil}{T_{m-1}} + 3 \left( \frac{\max\{1, |\mathcal{B}_m|\}}{T_{m-1}|\mathcal{A}_m|} \right)^{2/3} + \frac{3}{T_{m-1}} + 4\epsilon_m \\
&= \max_{b \in [K]} \widehat{\mu}_b(t_{m,a}^{(j_{m,a})}) - \widehat{\mu}_a(t_{m,a}^{(j_{m,a})}) + \frac{4}{T} + \frac{\lceil \log_2 T \rceil}{T_{m-1}} + 3 \left( \frac{\max\{1, |\mathcal{B}_m|\}}{T_{m-1}|\mathcal{A}_m|} \right)^{2/3} + \frac{3}{T_{m-1}} + 4\epsilon_m \\
&\leq \max_{b \in [K]} \widehat{\mu}_b(t_{m,a}^{(j_{m,a})} + 1) - \widehat{\mu}_a(t_{m,a}^{(j_{m,a})} + 1) + \frac{4}{T} + \frac{\lceil \log_2 T \rceil}{T_{m-1}} + 4 \left( \frac{\max\{1, |\mathcal{B}_m|\}}{T_{m-1}|\mathcal{A}_m|} \right)^{2/3} + \frac{4}{T_{m-1}} + 4\epsilon_m \\
&\leq \max_{b \in [K]} \widehat{\mu}_b(t) - \mu_a + \frac{4}{T} + \frac{\lceil \log_2 T \rceil}{T_{m-1}} + 4 \left( \frac{\max\{1, |\mathcal{B}_m|\}}{T_{m-1}|\mathcal{A}_m|} \right)^{2/3} + \frac{4}{T_{m-1}} + 5\epsilon_m.
\end{aligned}
$$

Conditioning on $\mathcal{E}$, the argument holds for each $m \geq 2$ and $a \in \mathcal{A}_m$, and thus the proof is complete. $\qquad \square$

**Lemma E.13.** *Suppose that $\mathcal{E}$ occurs. For each phase $m \geq 2$ and each arm $a \in \mathcal{A}_m$, we have*

$$\forall t \in \bigcup_{i \leq j_{m,a}} \mathcal{T}^{(i)}(a; \mathcal{A}_m) : \overline{b}_{m,a}^{(j_{m,a})} \leq \max_{b \in [K]} \mu_b - \mu_a + \frac{4}{T} + \frac{\lceil \log_2 T \rceil}{T_{m-1}} + 4 \left( \frac{\max\{1, |\mathcal{B}_m|\}}{T_{m-1}|\mathcal{A}_m|} \right)^{2/3} + \frac{4}{T_{m-1}} + 6\epsilon_m.$$

*Proof.* Consider fixed $m, a$. Let $b_t \in \text{argmax}_{a \in [K]} \widehat{\mu}_a(t)$. For any $t \in \bigcup_{i \leq j_{m,a}} \mathcal{T}^{(i)}(a; \mathcal{A}_m)$, we can use Eq. (40) to bound

$$\widehat{\mu}_{b_t}(t) \leq \mu_{b_t} + \epsilon_m \leq \max_{a \in [K]} \mu_a + \epsilon_m.$$

Pluggin the above into Lemma E.12, we show the claim for the fixed $m, a$. Conditioning on $\mathcal{E}$, the argument holds for each $m \geq 2$ and $a \in \mathcal{A}_m$, and thus the proof is complete. $\qquad \square$

**Lemma E.14.** *Suppose event $\mathcal{E}$ occurs. For all $m \in \mathbb{N}$, $a^\star \in \mathcal{A}_m$ holds.*

*Proof.* We prove the claim by the induction. For $m = 1$, the claim trivially holds. Suppose the claim holds for $m$, and then we consider phase $m + 1$. For each round $t \in [t_{m,a^\star}^0, t_{m,a^\star}^0 + 8 \log(8K\delta^{-1} \log_2 T)]$ and each $a \in \mathcal{A}_m - \{a^\star\}$, we have

$$
\begin{aligned}
0 &\leq \theta_{a^\star} + \mu_{a^\star} - (\theta_a + \mu_a) \\
&\leq \widehat{\theta}_{a^\star}(t_{m,a^\star}^0) + \widehat{\mu}_{a^\star}(t) - (\widehat{\theta}_a(t_{m,a^\star}^0) + \widehat{\mu}_a(t)) + 4\sqrt{\frac{\log(16KT/\delta)}{2T_m}} \\
&= \pi_{a^\star}(t) + \widehat{\mu}_{a^\star}(t) - (\widehat{\mu}_a(t) + \pi_a(t)) - \sqrt{\frac{\log(16KT/\delta)}{2T_m}},
\end{aligned}
$$

where the second inequality holds since Lemma E.11 gives that in each phase $m$, every active arm is played for $T_m$ times before the elimination starts, and $a^\star$ is active in phase $m$ by the hypothesis induction, and the equality holds since when

testing arm $a^\star$, $\pi_{a^\star}(t) = 1 + \widehat{\theta}_{a^\star}(t^0_{m,a^\star}) + 5\sqrt{\frac{\log(16KT/\delta)}{2T_m}}$, and the incentives of all other active arms $a \in \mathcal{A}_m - \{a^\star\}$ are all equal to $1 + \widehat{\theta}_a(t^0_{m,a^\star})$. Since the above holds for all $a \in \mathcal{A}_m$, we rearrange it to get

$$\pi_{a^\star}(t) + \widehat{\mu}_{a^\star}(t) \geq \max_{a \in \mathcal{A}_m - \{a^\star\}} \{\widehat{\mu}_a(t) + \pi_a(t)\} + \sqrt{\frac{\log(16KT/\delta)}{2T_m}} > \max_{a \in \mathcal{A}_m - \{a^\star\}} \{\widehat{\mu}_a(t) + \pi_a(t)\}.$$

Conditioning on $\mathcal{E}$, we use Lemma E.5 to get $a^\star \in \mathcal{A}_{m+1}$. Once the induction done, the proof is complete. $\square$

**Lemma E.15.** *Let $m_a$ be the smallest phase such that $\Delta_a > 9\sqrt{\frac{\log(8KT/\delta)}{2T_{m_a}}}$. Suppose that $\mathcal{E}$ occurs. For each arm $a$ with $\Delta_a > 0$, it will not be in $\mathcal{A}_m$ for all phases $m \geq m_a + 1$.*

*Proof.* Consider any arm $a$ with $\Delta_a > 0$. We only need to consider $a \in \mathcal{A}_{m_a}$ and otherwise, the claim naturally holds. For any round $t \in [t^0_{m,a}, t^0_{m,a} + 8\log(8K\delta^{-1}\log_2 T)]$, we have

$$\max_{b \in \mathcal{A}_{m_a} - \{a\}} \{\widehat{\mu}_b(t) + \pi_b(t)\} - (\widehat{\mu}_a(t) + \pi_a(t))$$
$$\geq (\widehat{\mu}_{a^\star}(t) + \pi_{a^\star}(t)) - (\widehat{\mu}_a(t) + \pi_a(t))$$
$$= \widehat{\mu}_{a^\star}(t) + \widehat{\theta}_{a^\star}(t^0_{m,a}) - \left(\widehat{\mu}_a(t) + \widehat{\theta}_a(t^0_{m,a})\right) - 5\sqrt{\frac{\log(16KT/\delta)}{2T_{m_a}}}$$
$$\geq \Delta_a - 9\sqrt{\frac{\log(16KT/\delta)}{2T_{m_a}}}$$
$$> 0,$$

where the first inequality uses $a \neq a^\star$ and Lemma E.14 that $a^\star \in \mathcal{A}_{m_a}$, the equality holds since when testing arm $a$, $\pi_a(t) = 1 + \widehat{\theta}_a(t^0_{m,a}) + 5\sqrt{\frac{\log(16KT/\delta)}{2T_m}}$, and the incentives of all other (non-target) active arms $b \in \mathcal{A}_m$ are all equal to $1 + \widehat{\theta}_b(t^0_{m,a})$, and the second inequality holds since Lemma E.11 gives that in each phase $m$, every active arm is played for $T_m$ times before the elimination starts.

According to Lemma E.5, arm $a$ will not be in phases $m$ for all $m \geq m_a + 1$. $\square$

# F. Omitted Proof of Self-interested Oracle-Agent with Exploration

---

**Algorithm 6** Proposed algorithm for self-interested oracle-agent with exploration

---

**Input**: confidences $\delta \in (0, 1)$, horizon $T$.
**Initialize**: active arm set $\mathcal{A}_1 = [K]$, bad arm set $\mathcal{B}_1 = \emptyset$, $T_0 = 1$.

1   **for** $m = 1, 2, \ldots$ **do**
2     Set $T_m$ based on Eq. (8).
3     **for** $a \in \mathcal{A}_m$ **do**                                                ▷ Search Incentives repeatedly
4       Use binary search for $2 \log(4 \log_2 T/\delta)$ times and sort outputs such that

$$b_{m,a}^{(1)} \leq b_{m,a}^{(2)} \leq \cdots \leq b_{m,a}^{(2 \log(4 \log_2 T/\delta))}.$$

5       Set $\overline{b}_{m,a}^{(i)} = b_{m,a}^{(i)} + T^{-1}$ for all $i$.
6     **for** $a \in \mathcal{A}_m$ **do**
7       **for** $i = 1, \ldots, 2 \log(4 \log_2 T/\delta)$ **do**                         ▷ Incentive testing
8         Set a counter $c_{m,a}^{(i)} = 0$ and $Y_{m,a}^{(i)} = 0$.
9         **repeat**
10           Propose incentives $\pi^0(a; \overline{b}_{m,a}^{(i)})$ and denote the current round by $t$.
11           Update $Y_{m,a}^{(i)} = Y_{m,a}^{(i)} + 1$, and if $A_t \neq a$, update $c_{m,a}^{(i)} = c_{m,a}^{(i)} + 1$.
         **until** $c_{m,a}^{(i)} > 2c_0 \sqrt{Y_{m,a}^{(i)} \log(2T)} + \sqrt{\frac{8 \log(\iota)}{Y_{m,a}^{(i)}}}$ **or** $Y_{m,a}^{(i)} = 2T_m$
12         If $\sum_{j \leq i}(Y_{m,a}^{(j)} - c_{m,a}^{(j)}) \geq T_m$, then break the loop for $i$.
13     **for** $a \in \mathcal{A}_m$ **do**                                        ▷ Trustworthy online elimination
14       Set $\mathcal{L}_{m,a} = \emptyset$.
15       Let $t_{m,a}^0$ be the current round, and $\{\widehat{\theta}_a(t_{m,a}^0)\}_{a \in [K]}$ are empirical means at this round.
16       **for** $t = t_{m,a}^0, \ldots, t_{m,a}^0 + 8 \log(8K \log_2 T/\delta)$ **do**
17         Propose incentives $\pi(t)$ with $\pi_a(t) = 1 + \widehat{\theta}_a(t_{m,a}^0) + 3\sqrt{\frac{\log(8KT/\delta)}{2T_m}}$, $\pi_b(t) = 1 + \widehat{\theta}_b(t_{m,a}^0)$, $\forall b \in \mathcal{A}_m - \{a\}$,
         and $\pi_i(t) = 0$, $\forall i \in \mathcal{B}_m$.
18         If $A_t \neq a$, then update $\mathcal{L}_{m,a} = \mathcal{L}_{m,a} \cup \{0\}$; else $\mathcal{L}_{m,a} = \mathcal{L}_{m,a} \cup \{1\}$.
19       Sort $\mathcal{L}_{m,a}$ in ascending order.
20       **if** $Median(\mathcal{L}_{m,a}) = 0$ **then**
21         Update $\mathcal{A}_{m+1} = \mathcal{A}_m - \{a\}$ and $\mathcal{B}_{m+1} = \mathcal{B}_m \cup \{a\}$.

---

## F.1. Notations

Throughout the proof in this section, we follow the exactly same notations used in Appendix E.2.

## F.2. Construction of Nice Event $\mathcal{E}$

All lemmas in this section follows exactly the same argument used in Appendix E.3.

Let us define event $\mathcal{E}_0$ as

$$\mathcal{E}_0 := \left\{ \forall (t, a) \in [T] \times [K] : \left| \widehat{\theta}_a(t) - \theta_a \right| \leq \sqrt{\frac{\log(8TK/\delta)}{2N_a(t)}} \right\}. \tag{43}$$

**Lemma F.1.** $\mathbb{P}(\mathcal{E}_0) \geq 1 - \delta/4$ *holds*.

*Proof.* By Hoeffding's inequality and invoking union bound, one can obtain the desired claim. $\square$

**Lemma F.2.** *With probability at least $1 - \delta/4$, for all phases $m$, all active arms $a \in \mathcal{A}_m$, and all iterations $i$,*

$$\sum_{t \in \mathcal{T}^{(i)}(a;\mathcal{A}_m)} I_t \leq 2c_0 \sqrt{Y_{m,a}^{(i)} \log(2T)} + \sqrt{\frac{8 \log(\iota)}{Y_{m,a}^{(i)}}}.$$

**Lemma F.3.** *With probability at least $1 - \delta/4$, for all $m \geq 2$, among total $2 \log(4 \log_2 T/\delta)$ calls, there will be at least one call of Algorithm 3 such that the agent makes no exploration.*

**Lemma F.4.** *With probability at least $1 - \delta/4$, for all phases $m$ and all $a \in \mathcal{A}_m$, if arm $a$ satisfies that for all rounds $t \in [t_{m,a}^0, t_{m,a}^0 + 8 \log(8K\delta^{-1} \log_2 T)]$:*

$$\max_{b \in \mathcal{A}_m - \{a\}} (\pi_b(t) + \mu_b) < (\pi_a(t) + \mu_a),$$

*then $a \in \mathcal{A}_{m+1}$; if arm $a$ satisfies that for all rounds $t \in [t_{m,a}^0, t_{m,a}^0 + 8 \log(8K\delta^{-1} \log_2 T)]$:*

$$\max_{b \in \mathcal{A}_m - \{a\}} (\pi_b(t) + \mu_b) > (\pi_a(t) + \mu_a),$$

*then $a \in \mathcal{B}_{m+1}$.*

**Definition F.5** (Define $\mathcal{E}$)**.** Let $\mathcal{E}$ be the event that $\mathcal{E}_0$ and inequalities in Lemma F.2, Lemma 4.1, and Lemma F.4 hold simultaneously.

Based on Definition F.5, one can easily see $\mathbb{P}(\mathcal{E}) \geq 1 - \delta$, by using a union bound.

## F.3. Supporting Lemmas

**Lemma F.6.** *Suppose that $\mathcal{E}$ occurs. For all phases $m$ and all active arms $a$, we have*

$$\overline{b}_{m,a}^{(j_{m,a})} \in \left( \pi_a^\star, \pi_a^\star + \frac{2}{T} \right].$$

*Proof.* As $\mathcal{E}$ holds, Lemma F.3 ensures the existence of $j_{m,a}$. Then, (Scheid et al., 2024b, Lemma 8) gives the desired result. $\square$

**Lemma F.7.** *Suppose that $\mathcal{E}$ occurs. For each phase $m$, each active arm $a \in \mathcal{A}_m$ will be played for at least $T_m$ times before the elimination starts. Moreover, for each phase $m$ and active arm $a \in \mathcal{A}_m$, $i_{m,a} \leq j_{m,a}$ and $\overline{b}_{m,a}^{(i)} \leq \overline{b}_{m,a}^{(j_{m,a})}$ for all $i \leq j_{m,a}$.*

*Proof.* Since we assume $\mathcal{E}$ holds, Lemma F.6 implies that if the algorithm tests $\overline{b}_{m,a}^{(j_{m,a})}$ for target active arm $a$, then the agent will always play arm $a$, except exploration occurs. Thus, the desired claim is immediate via the exact same argument in Lemma E.11. $\square$

**Lemma F.8.** *Suppose that $\mathcal{E}$ holds. For all $m \in \mathbb{N}$, $a^\star \in \mathcal{A}_m$.*

*Proof.* This proof is similar to Lemma E.14. We prove the claim by the induction. For $m = 1$, the claim trivially holds. Suppose the claim holds for $m$ and consider for $m + 1$. For each round $t \in [t_{m,a^\star}^0, t_{m,a^\star}^0 + 8 \log(8K\delta^{-1} \log_2 T)]$ and each $a \in \mathcal{A}_m - \{a^\star\}$, we have

$$0 \leq \theta_{a^\star} + \mu_{a^\star} - (\theta_a + \mu_a)$$

$$\leq \widehat{\theta}_{a^\star}(t_{m,a}^0) + \mu_{a^\star} - (\widehat{\theta}_a(t_{m,a}^0) + \mu_a) + 2\sqrt{\frac{\log(8KT/\delta)}{2T_m}}$$

$$= \pi_{a^\star}(t) + \mu_{a^\star} - (\mu_a + \pi_a(t)) - \sqrt{\frac{\log(16KT/\delta)}{2T_m}},$$

where the second inequality holds since Lemma F.7 gives that in each phase $m$, every active arm is played for $T_m$ times before the elimination starts, and $a^\star$ is active in phase $m$ by the hypothesis induction, and the equality holds since when testing arm $a^\star$, $\pi_{a^\star}(t) = 1 + \widehat{\theta}_{a^\star}(t^0_{m,a^\star}) + 3\sqrt{\frac{\log(16KT/\delta)}{2T_m}}$, and the incentives of all other active arms $a \in \mathcal{A}_m - \{a^\star\}$ are all equal to $1 + \widehat{\theta}_a(t^0_{m,a^\star})$. Since the above holds for all $a \in \mathcal{A}_m$, we rearrange it to get

$$\pi_{a^\star}(t) + \mu_{a^\star} \geq \max_{a \in \mathcal{A}_m - \{a^\star\}} \{\mu_a + \pi_a(t)\} + \sqrt{\frac{\log(16KT/\delta)}{2T_m}} > \max_{a \in \mathcal{A}_m - \{a^\star\}} \{\mu_a(t) + \pi_a(t)\}.$$

Conditioning on $\mathcal{E}$, we use Lemma F.4 to get $a^\star \in \mathcal{A}_{m+1}$. Once the induction done, the proof is complete. □

**Lemma F.9.** *Let $m_a$ be the smallest phase such that $\Delta_a > 5\sqrt{\frac{\log(8KT/\delta)}{2T_{m_a}}}$. Suppose that $\mathcal{E}$ occurs. For each arm $a$ with $\Delta_a > 0$, it will not be in $\mathcal{A}_m$ for all phases $m \geq m_a + 1$.*

*Proof.* This proof follows a similar to Lemma E.15. Consider any arm $a$ with $\Delta_a > 0$. We only need to consider $a \in \mathcal{A}_{m_a}$ and otherwise, the claim naturally holds. For any round $t \in [t^0_{m,a}, t^0_{m,a} + 8\log(8K\delta^{-1}\log_2 T)]$, we have

$$\max_{b \in \mathcal{A}_{m_a} - \{a\}} \{\mu_b + \pi_b(t)\} - (\mu_a + \pi_a(t))$$
$$\geq (\mu_{a^\star} + \pi_{a^\star}(t)) - (\mu_a + \pi_a(t))$$
$$= \mu_{a^\star} + \widehat{\theta}_{a^\star}(t^0_{m,a}) - \left(\mu_a + \widehat{\theta}_a(t^0_{m,a})\right) - 3\sqrt{\frac{\log(16KT/\delta)}{2T_{m_a}}}$$
$$\geq \Delta_a - 5\sqrt{\frac{\log(16KT/\delta)}{2T_{m_a}}}$$
$$> 0,$$

where the first inequality uses $a \neq a^\star$ and Lemma F.8 that $a^\star \in \mathcal{A}_{m_a}$, the equality holds since when testing arm $a$, $\pi_a(t) = 1 + \widehat{\theta}_a(t^0_{m,a}) + 3\sqrt{\frac{\log(16KT/\delta)}{2T_m}}$, and the incentives of all other (non-target) active arms $b \in \mathcal{A}_m$ are all equal to $1 + \widehat{\theta}_b(t^0_{m,a})$, and the second inequality holds since Lemma F.7 gives that in each phase $m$, every active arm is played for $T_m$ times before the elimination starts.

According to Lemma F.4, arm $a$ will not be in phases $m$ for all $m \geq m_a + 1$. □

### F.4. Comparison with (Dogan et al., 2023a) under Same Regret Metric

Dogan et al. (2023a) use a different regret definition $\mathbb{E}[\overline{R}_T]$ where

$$\overline{R}_T = \sum_{t=1}^{T} \left( \max_{b \in [K]} \{\theta_b + \mu_b\} - \max_{z \in [K]} \mu_z - \left( \theta_{A_t} - \sum_{a \in \mathcal{A}} \pi_a(t) \right) \right). \tag{44}$$

To fairly compare the regret bound, we evaluate the regret bound by $\mathbb{E}[\overline{R}_T]$.

**Theorem F.10.** *Suppose $T = \Omega(K)$. By choosing $\delta = 1/T$, Algorithm 5 ensures*

$$\mathbb{E}[\overline{R}_T] = \mathcal{O}\left( \log^{\frac{5}{3}}(KT)K^{\frac{1}{3}}T^{\frac{2}{3}} + \log^2(KT)\sqrt{KT} + K^2\log^3(KT) \right).$$

*Proof.* We first follow the same reasoning in Appendix E.5 to bound the regret in binary search, elimination period, and the first phase by $\mathcal{O}\left(K^2\log^3(KT/\delta)\right)$. Then, $\overline{R}_T$ can be written as:

$$\overline{R}_T = \mathcal{O}\left(K^2\log^3(KT/\delta)\right) + \sum_{m \geq 2} \overline{R}_m,$$

where (recall that Algorithm 6 does not play bad arms for stabilization)

$$\overline{R}_m = \sum_{a \in \mathcal{A}_m} \sum_{i \leq i_{m,a}} \sum_{t \in \mathcal{T}^{(i)}(a;\mathcal{A}_m)} \left( \max_{b \in [K]} \{\theta_b + \mu_b\} - \max_{z \in [K]} \mu_z - \left( \theta_{A_t} - \sum_{v \in \mathcal{A}} \pi_v(t) \right) \right)$$

$$= \sum_{a \in \mathcal{A}_m} \sum_{i \leq i_{m,a}} \sum_{t \in \mathcal{T}^{(i)}(a;\mathcal{A}_m)} \left( \max_{b \in [K]} \{\theta_b + \mu_b\} - \max_{z \in [K]} \mu_z - (\theta_{A_t} - \pi_a(t)) \right),$$

where the second equality follows from the fact that the proposed incentive is one-hot (arm $a$ has the only positive value) for all rounds in $\mathcal{T}^{(i)}(a;\mathcal{A}_m)$.

According to the incentive testing, the algorithm proposes one-hot incentives, and thus we have

$$\overline{R}_m = \sum_{a \in \mathcal{A}_m} \sum_{i \leq i_{m,a}} \sum_{t \in \mathcal{T}^{(i)}(a;\mathcal{A}_m)} \left( \max_{b \in [K]} \{\theta_b + \mu_b\} - \max_{z \in [K]} \mu_z - (\theta_{A_t} - \pi_a(t)) \right)$$

$$= \sum_{a \in \mathcal{A}_m} \sum_{i \leq i_{m,a}} \sum_{t \in \mathcal{T}^{(i)}(a;\mathcal{A}_m)} \left( \max_{b \in [K]} \{\theta_b + \mu_b\} - \max_{z \in [K]} \mu_z - \left( \theta_{A_t} - \overline{b}_{m,a}^{(i)} \right) \right)$$

$$\leq \sum_{a \in \mathcal{A}_m} \sum_{i \leq i_{m,a}} \sum_{t \in \mathcal{T}^{(i)}(a;\mathcal{A}_m)} \left( \max_{b \in [K]} \{\theta_b + \mu_b\} - \max_{z \in [K]} \mu_z - \theta_{A_t} + \pi_a^\star + \frac{2}{T} \right)$$

$$= \sum_{a \in \mathcal{A}_m} \sum_{i \leq i_{m,a}} \sum_{t \in \mathcal{T}^{(i)}(a;\mathcal{A}_m)} \left( \Delta_a + \frac{2}{T} \right) \mathbb{I}\{A_t = a\}$$

$$+ \sum_{a \in \mathcal{A}_m} \sum_{i \leq i_{m,a}} \sum_{t \in \mathcal{T}^{(i)}(a;\mathcal{A}_m)} \left( \max_{b \in [K]} \{\theta_b + \mu_b\} - \max_{z \in [K]} \mu_z - \theta_{A_t} + \pi_a^\star + \frac{2}{T} \right) \mathbb{I}\{A_t \neq a\},$$

where the inequality uses Lemma F.7 and Lemma F.6 to bound for all $i \leq j_{m,a}$:

$$\overline{b}_{m,a}^{(i)} \leq \overline{b}_{m,a}^{(j_{m,a})} \leq \pi_a^\star + \frac{2}{T}.$$

Then, we use Eq. (34) to bound

$$\sum_{a \in \mathcal{A}_m} \sum_{i \leq i_{m,a}} \sum_{t \in \mathcal{T}^{(i)}(a;\mathcal{A}_m)} \left( \max_{b \in [K]} \{\theta_b + \mu_b\} - \max_{z \in [K]} \mu_z - \theta_{A_t} + \pi_a^\star + \frac{2}{T} \right) \mathbb{I}\{A_t \neq a\}$$

$$\leq \mathcal{O} \left( \log(\delta^{-1} \log T) \left( \sqrt{K |\mathcal{T}_m| \log(T)} + |\mathcal{A}_m| \sqrt{\log \iota} \right) \right).$$

Note that line 12 of Algorithm 6 implies that $\sum_{i \leq i_{m,a}} \sum_{t \in \mathcal{T}^{(i)}(a;\mathcal{A}_m)} \mathbb{I}\{A_t = a\} \leq \mathcal{O}(T_m)$. Hence, using $T_{m+1} = \Theta(T_m)$ for all $m$, we have

$$\sum_{a \in \mathcal{A}_m} \sum_{i \leq i_{m,a}} \sum_{t \in \mathcal{T}^{(i)}(a;\mathcal{A}_m)} \Delta_a \mathbb{I}\{A_t = a\} \leq \mathcal{O} \left( T_m \sum_{a \in \mathcal{A}_m} \Delta_a \right)$$

$$\leq \mathcal{O} \left( |\mathcal{A}_m| \sqrt{T_m \log(KT/\delta)} \right)$$

$$\leq \mathcal{O} \left( \sqrt{|\mathcal{A}_m| |\mathcal{T}_m| \log(KT/\delta)} \right).$$

where the second inequality uses the same approach in Eq. (37) with Lemma F.9, and the last inequality holds due to $|\mathcal{A}_m| T_m \leq |\mathcal{T}_m|$. Therefore,

$$\sum_{m \geq 2} \overline{R}_m \leq \mathcal{O} \left( \sum_{m \geq 2} \left( \sqrt{|\mathcal{A}_m| |\mathcal{T}_m| \log(KT/\delta)} + \log(\delta^{-1} \log T) \left( \sqrt{K |\mathcal{T}_m| \log(T)} + |\mathcal{A}_m| \sqrt{\log \iota} \right) \right) \right)$$

---

**Algorithm 7** Proposed algorithm for linear reward model

---

**Input**: confidence $\delta \in (0, 1)$, horizon $T$.

**Initialize**: active arm set $\mathcal{A}_1 = \mathcal{A}$, bad arm set $\mathcal{B}_1 = \emptyset$, $\epsilon_0 = 1$.

1 **for** $m = 1, 2, \ldots$ **do**

2     Set $T_m = 2^{m+4} d \log(4KT\delta^{-1})$ and $\epsilon_m = 4\sqrt{\frac{d\log(4KT\delta^{-1})}{\min\{T_m, (d\log(4KT\delta^{-1}))^{1/3}T_m^{2/3}\}}}$.

3     **if** $\mathcal{B}_m \neq \emptyset$ **then**

4        Find a design $\omega_m$ for $\mathcal{Z} = \mathcal{B}_m$ and $C = 2$ in Definition G.1.

5        **for** $a \in \mathcal{B}_m$ **do**                                       ▷ Stabilize estimators for bad arms

6           Propose incentives $\pi^0(a; 2d + T^{-1})$ for $U_m(a) = \lceil \omega_m(a)(d\log(4TK/\delta))^{1/3} \cdot T_m^{2/3}\rceil$ rounds.

7     Find a design find a design $\rho_m$ for $\mathcal{Z} = \mathcal{A}_m$ and $C = 2$ in Definition G.1.

8     Invoke Algorithm 8 with input $(\epsilon_{m-1}, T^{-1})$ to get output $c_m \in \mathbb{R}^d$.           ▷ Search parameter for $s^\star$

9     **for** $a \in \mathcal{A}_m$ **do**

10        Set $\bar{b}_{m,a} = \min\left\{2d + T^{-1}, \max_{b \in \mathcal{A}}\langle c_m, b - a\rangle + (1 + 32d)\epsilon_{m-1} + \frac{1}{T}\right\}$.

11        For the following $\lceil \rho_m(a)T_m\rceil$ rounds, propose incentive $\pi^0(a; \bar{b}_{m,a})$.

12     Update estimates

$$\widehat{\nu}_m = V_m^{-1}\sum_{t \in \mathcal{T}_m} A_t X_{A_t}(t), \quad \text{where} \quad V_m = \sum_{a \in \mathcal{A}_m}\lceil \rho_m(a)T_m\rceil aa^\top + \sum_{b \in \mathcal{B}_m} U_m(a)bb^\top, \quad (45)$$

    where $\mathcal{T}_m$ is a set contains all rounds when interaction occurs in line 11 and line 6 in phase $m$.

13     Invoke Algorithm 8 again with input $(\epsilon_m, T^{-1})$ to get output $c'_m \in \mathbb{R}^d$.

14     **for** $a \in \mathcal{A}_m$ **do**                                                    ▷ Offline Elimination

15        **if** $\max_{b \in \mathcal{A}_m}\langle \widehat{\nu}_m + c'_m, b - a\rangle > (7 + 32d)\epsilon_m$ **then**

          Update $\mathcal{A}_{m+1} = \mathcal{A}_m - \{a\}$ and $\mathcal{B}_{m+1} = \mathcal{B}_m \cup \{a\}$

---

$$\leq \mathcal{O}\left(\log^2(KT/\delta)\sqrt{KT}\right).$$

Combining all the above, we have

$$\overline{R}_T = \mathcal{O}\left(\log^2(KT/\delta)\sqrt{KT} + K^2\log^3(KT/\delta)\right).$$

By choosing $\delta = T^{-1}$, we complete the proof.                                                     $\square$

# G. Omitted Proof for Linear Reward in Section 5

## G.1. Omitted Pseudocode of Algorithm 7 and Algorithm 8

We present the following definition.

**Definition G.1.** Let $\mathcal{Z} \subseteq \mathbb{R}^d$ be a finite and compact set. A distribution $\pi : \mathcal{Z} \to [0, 1]$ is a $C$-approximate design with an approximation factor $C \geq 1$, if $\sup_{z \in \mathcal{Z}} \|z\|_{G(\pi;\mathcal{Z})^{-1}}^2 \leq C \cdot d$ where $G(\pi; \mathcal{Z}) = \sum_{z \in \mathcal{Z}} \pi(z)zz^\top$.

Since the exactly optimal design (i.e., $C = 1$) is typically hard to compute, we consider the 2-approximately optimal design, which can be computed efficiently (Todd, 2016).

The omitted pseudocode can be found in Algorithm 7 and Algorithm 8.

## G.2. Notations

We introduce some notations that will be used throughout the proof. Refer to Appendix A for some general notations.

- Let $m(t)$ be the phase that round $t$ lies in.

---

**Algorithm 8** Multiscale steiner potential with conservative cut

---

**Input**: error $\epsilon > 0, \xi > 0$.

**Initialize:** let current round be $t_0$, $S_{t_0} = B(0, 1)$, $\overline{\mathcal{A}} = \{z : z = x - y \text{ s.t. } x \neq y \in \mathcal{A}\}$, $z_i = 2^{-i}/(8d)$, $\forall i \in \mathbb{N}$.

1 **for** $t = t_0, \dots$ **do**
2      Pick $x_t = (a_t^1 - a_t^2)/ \left\| a_t^1 - a_t^2 \right\|$ where $a_t^1 - a_t^2 \in \operatorname{argmax}_{u \in \overline{\mathcal{A}}} \texttt{width}(S_t, u)$ and $a_t^1 \neq a_t^2 \in \mathcal{A}$
3      Find largest index $i$ such that $\texttt{width}(S_t, x_t) \leq 2^{-i}$.
4      **if** $z_i < 4 \left\| a_t^1 - a_t^2 \right\|^{-1} \epsilon$ **then**
5          Break and randomly pick a vector from $S_t$ to return.
6      **else**
7          Query $y_t$ such that

$$\operatorname{Vol} \left( v \in S_t + z_i B(0, 1) : \langle v, x_t \rangle \geq \left\| a_t^1 - a_t^2 \right\|^{-1} (y_t - \epsilon) \right) = \frac{1}{2} \operatorname{Vol}(S_t + z_i B(0, 1)).$$

8          Propose incentive $\pi_{a_t^1}(t) = d + \xi$, $\pi_{a_t^2}(t) = d + \xi + y_t$, and $\pi_b(t) = 0$ for all $b \neq a_t^1, a_t^2$.
9          **if** $A_t = a_t^1$ **then**
10              Update $S_{t+1} = \left\{ v \in S_t : \langle v, x_t \rangle \geq \left\| a_t^1 - a_t^2 \right\|^{-1} (y_t - \epsilon) \right\}$.
11          **else**
12              Update $S_{t+1} = \left\{ v \in S_t : \langle v, x_t \rangle \leq \left\| a_t^1 - a_t^2 \right\|^{-1} (y_t + \epsilon) \right\}$.

---

- Let $\mathcal{T}_m$ be the set of rounds that in phase $m$, *excluding* those rounds for running Algorithm 8. In other words, $\mathcal{T}_m$ is a set contains all rounds when interaction occurs in line 11 and line 6 in phase $m$.

- Let $\mathcal{T}_{m,a} = \{t \in \mathcal{T}_m : A_t = a\}$.

- Let $\mathcal{T}_{m,a}^E$ be a set of all rounds when Algorithm 7 runs line 11 or line 6 for target arm $a$ in phase $m$. In fact, Lemma G.6 implies that conditioning on $\mathcal{E}$, $\mathcal{T}_{m,a}^E = \mathcal{T}_{m,a}$.

- Let $U_t = \sum_{s=1}^t A_s A_s^\top$ and let $U_t^\dagger$ be its pseudo-inverse. With the definition of $U_t$, $\widehat{s}_t$ can be written as:

$$\widehat{s}_t = U_t^\dagger \sum_{s=1}^t R_{A_s}(s) A_s.$$

- Let $\mathcal{T}_{>1}$ be the set of all rounds that not in phase 1 and let us define event

$$\mathcal{E} = \mathcal{E}^{\texttt{Principle}} \cap \mathcal{E}^{\texttt{Agent}},$$

where

$$\mathcal{E}^{\texttt{Agent}} = \left\{ \forall (a, t) \in \mathcal{A} \times \mathcal{T}_{>1} : |\langle \widehat{s}_t - s^\star, a \rangle| \leq \sqrt{2 \left\| a \right\|_{U_t^{-1}} \log \left( \frac{4KT}{\delta} \right)} \right\},$$

and

$$\mathcal{E}^{\texttt{Principle}} = \left\{ \forall (a, m) \in \mathcal{A} \times \mathbb{N} : |\langle \widehat{\nu}_m - \nu^\star, a \rangle| \leq \sqrt{2 \left\| a \right\|_{V_m^{-1}} \log \left( \frac{4KT}{\delta} \right)} \right\}.$$

Notice that in phase 1, every arm will be played deterministically according to (approximately) G-optimal design. Therefore, once phase 1 ends, for all $t \in \mathcal{T}_{>1}$, $U_t$ is invertible which implies that $U_t^\dagger = U_t^{-1}$.

## G.3. Proof of Theorem 5.1

The following analysis conditions on $\mathcal{E}$, which occurs with probability at least $1 - \delta$ by a standard analysis of linear bandits (Lattimore & Szepesvári, 2020, Section 20). As the algorithm runs in phases, we bound the regret in phase $m = 1$ can be bounded by $\left( d^2 \log(KT/\delta) \right)$ since per-round regret is $\mathcal{O}(d)$ and $T_1 = \mathcal{O}(d \log(KT\delta^{-1}))$. Then we bound the regret in each phase $m \geq 2$. Lemma G.5 shows Algorithm 8 lasts at most $\mathcal{O}(d \log^2(d\epsilon_{m-1})) \leq \mathcal{O}(d \log^2(dKT/\delta))$ rounds in phase $m \geq 2$. As the number of phases is at most $\mathcal{O}(\log T)$, we have

$$R_T \leq \mathcal{O}\left( d^2 \log^2(dKT/\delta) \cdot \log(T) \right) + \sum_{m \geq 2} R_m,$$

where

$$R_m = \sum_{t \in \mathcal{T}_m} \left( \max_{a \in \mathcal{A}} \{ \langle \nu^\star, a \rangle - \pi_a(t) \} - (\langle \nu^\star, A_t \rangle - \pi_{A_t}(t)) \right).$$

It remains to bound $R_m$ for $m \geq 2$. Then, let consider a fixed $m \geq 2$ and bound

$$
\begin{aligned}
R_m &= \sum_{t \in \mathcal{T}_m} \left( \langle \nu^\star, a_t^\star \rangle + \langle \widehat{s}_t, a_t^\star \rangle - \max_{b \in \mathcal{A}} \langle \widehat{s}_t, b \rangle - (\langle \nu^\star, A_t \rangle - \pi_{A_t}(t)) \right) \\
&= \sum_{a \in \mathcal{A}_m} \sum_{t \in \mathcal{T}_{m,a}} \left( \langle \nu^\star, a_t^\star \rangle + \langle \widehat{s}_t, a_t^\star \rangle - \max_{b \in \mathcal{A}} \langle \widehat{s}_t, b \rangle - (\langle \nu^\star, A_t \rangle - \pi_{A_t}(t)) \right) \\
&\quad + \sum_{a \in \mathcal{B}_m} \sum_{t \in \mathcal{T}_{m,a}} \left( \langle \nu^\star, a_t^\star \rangle + \langle \widehat{s}_t, a_t^\star \rangle - \max_{b \in \mathcal{A}} \langle \widehat{s}_t, b \rangle - (\langle \nu^\star, A_t \rangle - \pi_{A_t}(t)) \right) \\
&= \sum_{a \in \mathcal{A}_m} \sum_{t \in \mathcal{T}_{m,a}} \left( \langle \nu^\star, a_t^\star \rangle + \langle \widehat{s}_t, a_t^\star \rangle - \max_{b \in \mathcal{A}} \langle \widehat{s}_t, b \rangle - (\langle \nu^\star, a \rangle - \pi_a(t)) \right) \\
&\quad + \sum_{a \in \mathcal{B}_m} \sum_{t \in \mathcal{T}_{m,a}} \left( \langle \nu^\star, a_t^\star \rangle + \langle \widehat{s}_t, a_t^\star \rangle - \max_{b \in \mathcal{A}} \langle \widehat{s}_t, b \rangle - (\langle \nu^\star, a \rangle - \pi_a(t)) \right) \\
&\leq \sum_{a \in \mathcal{A}_m} \sum_{t \in \mathcal{T}_{m,a}} \left( \langle \nu^\star, a_t^\star \rangle + \langle \widehat{s}_t, a_t^\star \rangle - \max_{b \in \mathcal{A}} \langle \widehat{s}_t, b \rangle - (\langle \nu^\star, a \rangle - \pi_a(t)) \right) \\
&\quad + \mathcal{O}\left( d^{\frac{4}{3}} T_m^{\frac{2}{3}} \log^{\frac{1}{3}}(TK/\delta) \right),
\end{aligned}
$$

where the first equality holds due to the definition of $a_t^\star$, the third equality follows from Lemma G.6, and the fact that the proposed incentives in these rounds are one-hot, and the last inequality bounds the regret on all bad arms by multiplying the number of rounds by $\mathcal{O}(d)$ (the upper bound of per-round regret).

Then, for each active arm $a \in \mathcal{A}_m$ and $t \in \mathcal{T}_{m,a}$, we turn to bound

$$
\begin{aligned}
&\langle \nu^\star, a_t^\star \rangle + \langle \widehat{s}_t, a_t^\star \rangle - \max_{b \in \mathcal{A}} \langle \widehat{s}_t, b \rangle - (\langle \nu^\star, a \rangle - \pi_a(t)) \\
&\leq \langle \nu^\star, a_t^\star \rangle + \langle s, a_t^\star \rangle + \epsilon_{m-1} - \max_{b \in \mathcal{A}} \langle \widehat{s}_t, b \rangle - (\langle \nu^\star, a \rangle - \pi_a(t)) \\
&\leq \langle \nu^\star + s^\star, a^\star \rangle + \epsilon_{m-1} - \max_{b \in \mathcal{A}} \langle \widehat{s}_t, b \rangle - (\langle \nu^\star, a \rangle - \pi_a(t)) \\
&\leq \mathcal{O}\left( \Delta_a + d\epsilon_{m-1} + \frac{1}{T} \right),
\end{aligned}
$$

where the first inequality holds due to Lemma G.7, and the last inequality uses Lemma G.8.

By Lemma G.6, each active arm $a \in \mathcal{A}_m$ will be played for $\lceil \rho_m(a) T_m \rceil$ times in $\mathcal{T}_{m,a} = \mathcal{T}_{m,a}^E$. As $|\mathcal{T}_{m,a}| = \lceil \rho_m(a) T_m \rceil$ for any active arm $a$, $T_{m+1} = \Theta(T_m)$ for all $m$, and $|\mathrm{supp}(\nu_m)| \leq \mathcal{O}(d \log \log d)$, we have

$$\sum_{a \in \mathcal{A}_m} \sum_{t \in \mathcal{T}_{m,a}} \left( \langle \nu^\star, a_t^\star \rangle + \langle \widehat{s}_t, a_t^\star \rangle - \max_{b \in \mathcal{A}} \langle \widehat{s}_t, b \rangle - (\langle \nu^\star, a \rangle - \pi_a(t)) \right)$$

$$\leq \mathcal{O}\left(T_m \sum_{a \in \mathcal{A}_m} \rho_m(a)\Delta_a + \frac{T_m}{T} + T_m d\epsilon_{m-1} + d \log \log d\right)$$

$$\leq \mathcal{O}\left(T_m \sum_{a \in \mathcal{A}_m} \rho_m(a)\Delta_a + d^{\frac{3}{2}}\sqrt{T_m \log(KT/\delta)} + \frac{T_m}{T} + d^{\frac{4}{3}}T_m^{\frac{2}{3}}\log^{\frac{1}{3}}(TK/\delta) + d \log \log d\right),$$

where the last inequality bounds

$$\epsilon_{m-1} = 4\sqrt{\frac{d \log\left(4KT\delta^{-1}\right)}{\min\{T_{m-1}, (d\log(4KT\delta^{-1}))^{1/3}T_{m-1}^{2/3}\}}}$$

$$\leq 4\sqrt{\frac{d \log\left(4KT\delta^{-1}\right)}{T_{m-1}}} + 2\sqrt{\frac{d \log\left(4KT\delta^{-1}\right)}{(d\log(4KT\delta^{-1}))^{1/3}T_{m-1}^{2/3}}}.$$

From Lemma G.10, if a suboptimal arm $a$ is active in phase $m$, then $m \leq m_a$ where $m_a$ is the smallest phase such that $\Delta_a > (9 + 64d)\epsilon_{m_a}$. This implies that $\forall a \in \mathcal{A}_m$:

$$\Delta_a \leq (9 + 64d)\epsilon_{m_a-1} \leq (9 + 64d)\epsilon_{m-1} \leq \mathcal{O}\left(d^{\frac{3}{2}}\sqrt{\frac{\log(KT/\delta)}{T_{m-1}}} + d^{\frac{4}{3}}(T_{m-1})^{-\frac{1}{3}}\log^{\frac{1}{3}}(TK/\delta)\right), \tag{46}$$

where the second inequality uses $m \leq m_a$.

By again using $T_{m+1} = \Theta(T_m)$ for all $m$ and $\sum_{a \in \mathcal{A}_m} \rho_m(a) = 1$, we have

$$R_T \leq \mathcal{O}\left(d^{\frac{3}{2}}\sqrt{\log(KT/\delta)}\sum_m \sqrt{T_m} + d^{\frac{4}{3}}\log^{\frac{1}{3}}(KT/\delta)\sum_m T_m^{\frac{2}{3}} + d^2 \log^2(dKT/\delta)\log T\right)$$

$$\leq \mathcal{O}\left(d^{\frac{3}{2}}\log(KT/\delta)\sqrt{T} + d^{\frac{4}{3}}T^{\frac{2}{3}}\log^{\frac{2}{3}}(KT/\delta) + d^2 \log^2(dKT/\delta)\log T\right),$$

where the last inequality uses Hölder's inequality together with the fact that the number of phases is at most $\mathcal{O}(\log T) \leq \mathcal{O}(\log(TK/\delta))$.

### G.4. Technical Lemmas for Algorithm 8

**Lemma G.2.** *If index $i$ is selected at round $t$ and the algorithm does not break this round, then*

$$\mathrm{Vol}(S_{t+1} + z_i B) \leq \frac{7}{8}\mathrm{Vol}(S_t + z_i B).$$

*Proof.* According to the incentive proposed during running Algorithm 8, the agent picks either $a_t^1$ or $a_t^2$ at any round $t$. For the case $A_t = a_t^1$, a similar argument of (Liu et al., 2021) gives that $\mathrm{Vol}(S_{t+1} + z_i B) \leq \frac{3}{4}\mathrm{Vol}(S_t + z_i B)$. For the case $A_t = a_t^2$, we have $S_{t+1} = \left\{v \in S_t : \langle v, x_t\rangle \leq \left\|a_t^1 - a_t^2\right\|^{-1}(y_t + \epsilon)\right\}$. Notice that

$$\left\{v \in S_{t+1} + z_i B : \langle v, x_t\rangle \leq \left\|a_t^1 - a_t^2\right\|^{-1}(y_t - \epsilon)\right\}$$

$$= \left\{v \in S_t + z_i B : \langle v, x_t\rangle \leq \left\|a_t^1 - a_t^2\right\|^{-1}(y_t - \epsilon)\right\}.$$

Due to the half-cut (line 7 in Algorithm 8), we have

$$\mathrm{Vol}\left(\left\{v \in S_{t+1} + z_i B : \langle v, x_t\rangle \leq \left\|a_t^1 - a_t^2\right\|^{-1}(y_t - \epsilon)\right\}\right) = \frac{1}{2}\mathrm{Vol}(S_t + z_i B).$$

Then, we bound the volume of $S_{t+1} + z_i B$ with $\langle v, x_t\rangle \geq \left\|a_t^1 - a_t^2\right\|^{-1}(y_t - \epsilon)$. Let $C$ be the largest volume of a section of $S_t + z_i B$ along the direction $x_t$. By the analysis of (Liu et al., 2021, Lemma 2.1), we have

$$\mathrm{Vol}(S_t + z_i B) \geq 4z_i C. \tag{47}$$

Since $\left\{v \in S_{t+1} + z_i B : \langle v, x_t \rangle \geq \left\| a_t^1 - a_t^2 \right\|^{-1} (y_t - \epsilon) \right\}$ has cross-section with volume at most $C$, the width is $z_i + 2 \left\| a_t^1 - a_t^2 \right\|^{-1} \epsilon \leq \frac{3}{2} z_i$ where the inequality uses the non-break condition (recall line 4), we have

$$\text{Vol}\left( \left\{ v \in S_{t+1} + z_i B : \langle v, x_t \rangle \geq \left\| a_t^1 - a_t^2 \right\|^{-1} (y_t - \epsilon) \right\} \right) \leq \frac{3}{2} z_i C \leq \frac{3}{8} \text{Vol}(S_t + z_i B),$$

where the last inequality holds due to Eq. (47). Combining both volumes, the proof is complete. $\square$

**Lemma G.3.** *For each round $t$ that Algorithm 8 runs in phase $m$, if $|\langle \widehat{s}_t - s^\star, a \rangle| \leq \epsilon/2$ for all $a \in \mathcal{A}$, then $s^\star \in S_t$ for all those $t$.*

*Proof.* For any round $t$, we assume $A_t = a_t^1$ (the other case $A_t = a_t^2$ is analogous). The agent selects $a_t^1$ indicates that

$$\langle \widehat{s}_t, a_t^1 \rangle + d + \xi \geq \langle \widehat{s}_t, a_t^2 \rangle + d + \xi + y_t.$$

By rearranging the above, we have $y_t \leq \langle \widehat{s}_t, a_t^1 - a_t^2 \rangle$. We use the assumption to get

$$y_t \leq \langle \widehat{s}_t, a_t^1 - a_t^2 \rangle \leq \langle s^\star, a_t^1 - a_t^2 \rangle + \epsilon,$$

Dividing the above by $\left\| a_t^1 - a_t^2 \right\|$ on both sides, we have

$$\frac{y_t}{\left\| a_t^1 - a_t^2 \right\|} \leq \left\langle s^\star, \frac{a_t^1 - a_t^2}{\left\| a_t^1 - a_t^2 \right\|} \right\rangle + \frac{\epsilon}{\left\| a_t^1 - a_t^2 \right\|} = \langle s^\star, x_t \rangle + \frac{\epsilon}{\left\| a_t^1 - a_t^2 \right\|}.$$

Based on the update rule, we have $s \in S_{t+1}$. As this holds for each $t$, the proof is complete. $\square$

**Lemma G.4.** *If Algorithm 8 breaks at round $t$ and the condition in Lemma G.3 holds, then*

$$\max_{u \in \overline{\mathcal{A}}} \texttt{width}(S_t, u) \leq 32 d\epsilon \quad \text{where} \quad \overline{\mathcal{A}} = \{z : z = x - y \text{ such that } x \neq y \in \mathcal{A}\}.$$

*Proof.* At each round $t$, if index $i$ is selected, then $\texttt{width}(S_t, x_t) \leq 2^{-i}$. We note that Lemma G.3 gives that $s^\star \in S_t$ and hence the index $i$ is well-defined. When the algorithm breaks, we have

$$z_i = \frac{2^{-i}}{8d} \leq 4 \left\| a_t^1 - a_t^2 \right\|^{-1} \epsilon.$$

which immediately leads to

$$4 \left\| a_t^1 - a_t^2 \right\|^{-1} \epsilon \geq \frac{1}{8d} \texttt{width}(S_t, x_t).$$

As $x_t = \left\| a_t^1 - a_t^2 \right\|^{-1} (a_t^1 - a_t^2)$, multiplying $\left\| a_t^1 - a_t^2 \right\|$ on both sides gives $\texttt{width}(S_t, a_t^1 - a_t^2) \leq 32 d\epsilon$. Recall the definition of $\overline{\mathcal{A}}$ and the way to picking $a_t^1, a_t^2$ from Algorithm 8, and thus the proof is complete. $\square$

**Lemma G.5.** *Suppose $\mathcal{E}$ holds. If Algorithm 8 runs in phase $m$ and the input $\epsilon$ satisfies $|\langle \widehat{s}_t - s^\star, a \rangle| \leq \epsilon/2$ for all $a \in \mathcal{A}$ and all $t$ that the algorithm runs in this phase, then it lasts at most $\mathcal{O}\left( d \log^2(d\epsilon) \right)$ rounds.*

*Proof.* From Lemma G.3, $s \in S_t$ for all $t$, and thus $\text{Vol}(S_t + z_i B) \geq \text{Vol}(z_i B) = z_i^d \text{Vol}(B)$. Whenever index $i$ is chosen, $\text{Vol}(S_t + z_i B)$ decreases by a constant factor, which implies that an index $i$ can be picked at most $\mathcal{O}\left( d \log(1/z_i) \right)$ times.

Then, we claim that any index $i$ that does not incur a break, must satisfy $z_i^{-1} \leq (2\epsilon)^{-1}$ and $i \leq \lceil \log_2(4d/\epsilon) \rceil$. We prove this by contradiction. Assume the algorithm picks index $i$ with $z_i^{-1} > 1/(2\epsilon)$ and does not break. In this case, we have $z_i < 2\epsilon = \frac{4\epsilon}{2} \leq \frac{4\epsilon}{\left\| a_t^1 - a_t^2 \right\|}$, which forms a contradiction.

Rearranging $z_i^{-1} \leq (2\epsilon)^{-1}$ yields $i \leq \lceil \log_2(4d/\epsilon) \rceil$. Therefore, the total number of round that the algorithm will last is at most

$$\mathcal{O}\left( \sum_{i \leq \lceil \log_2(4d/\epsilon) \rceil} d \log(z_i^{-1}) \right) \leq \mathcal{O}\left( \sum_{i \leq \lceil \log_2(4d/\epsilon) \rceil} d \log(1/\epsilon) \right) \leq \mathcal{O}\left( d \log^2(d/\epsilon) \right),$$

which completes the proof. $\square$

**Lemma G.6.** *Suppose that $\mathcal{E}$ occurs. For all phase $m$, all $a \in \mathcal{A}$ and all $t \in \mathcal{T}_{m,a}^E$,*

$$\pi_a(t) > \pi_a^\star(t).$$

*Proof.* Notice that since every target bad arm $a$ will be assigned with incentive $\pi_a(t) = 2d + T^{-1}$, and $\pi_a^\star(t) = \max_{b \in \mathcal{A}} \langle \widehat{s}_t, b - a \rangle \leq \max_{b \in \mathcal{A}} \|\widehat{s}_t\| \|b - a\| \leq 2d$, we have $\pi_a(t) > \pi_a^\star(t)$ for all $a \in \mathcal{T}_{m,a}^E$.

We then prove the claim for all active arms by using induction on $m$. For the base case $m = 1$. In this case, for all $a \in \mathcal{A}_m$ and all $t \in \mathcal{T}_{m,a}^E$, $\pi_a(t) = 2d + T^{-1}$. From the same analysis for bad arms, the claim holds for $m = 1$.

Assume that the claim also holds for phase $m - 1 \geq 1$, and then we prove the claim for phase $m$. In what follows, we focus on those rounds $t \in \mathcal{T}_{m,a}^E$. For shorthand, let

$$z_t \in \operatorname*{argmax}_{a \in \mathcal{A}} \langle \widehat{s}_t, a \rangle.$$

If $\pi_a(t) = 2d + T^{-1}$, then, we have $\pi_a(t) > \pi_a^\star(t)$ by the same argument used to prove for bad arms. Thus, we assume that $\pi_a(t) \neq 2d + T^{-1}$.

By the definition of $\mathcal{E}$, for any round $t$ in phase $m \geq 2$:

$$
\begin{aligned}
\max_{a \in \mathcal{A}} |\langle \widehat{s}_t - s^\star, a \rangle| &\leq \max_{a \in \mathcal{A}} \sqrt{2 \|a\|_{U_t^{-1}} \log\left(\frac{4KT}{\delta}\right)} \\
&\leq \max_{a \in \mathcal{A}} \sqrt{2 \|a\|_{V_{m(t)-1}^{-1}} \log\left(\frac{4KT}{\delta}\right)} \\
&\leq 2 \underbrace{\sqrt{\frac{d \log\left(4KT\delta^{-1}\right)}{\min\{T_{m(t)-1}, (d \log(4KT\delta^{-1}))^{1/3} T_{m(t)-1}^{2/3}\}}}}_{=\frac{\epsilon_{m(t)-1}}{2}},
\end{aligned}
\tag{48}
$$

where the first inequality uses the definition of $\mathcal{E}$, the second inequality holds due to $U_t \succeq V_{m(t)-1}$, and the last inequality follows from the induction hypothesis that each $a \in \mathcal{A}$ is played for $t \in \mathcal{T}_{m(t)-1,a}$. We further show that for all rounds $t \in \mathcal{T}_{m,a}^E$:

$$
\begin{aligned}
&\pi_a(t) - \pi_a^\star(t) \\
&= \max_{b \in \mathcal{A}} \langle c_{m(t)}, b - a \rangle + (1 + 32d)\epsilon_{m(t)-1} + \frac{1}{T} - \langle \widehat{s}_t, z_t - a \rangle \\
&\geq \langle c_{m(t)}, z_t - a \rangle + (1 + 32d)\epsilon_{m(t)-1} + \frac{1}{T} - \langle \widehat{s}_t, z_t - a \rangle \\
&\geq \langle c_{m(t)}, z_t - a \rangle + (1 + 32d)\epsilon_{m(t)-1} + \frac{1}{T} - \left(\langle s^\star, z_t - a \rangle + \epsilon_{m(t)-1}\right) \\
&= \langle c_{m(t)} - s^\star, z_t - a \rangle + 32d\epsilon_{m(t)-1} + \frac{1}{T} \\
&\geq -32d\epsilon_{m(t)-1} + 32d\epsilon_{m(t)-1} + \frac{1}{T} - \epsilon_{m(t)-1} \\
&= \frac{1}{T},
\end{aligned}
$$

where the first inequality uses the definition of $z_t$, the second inequality uses Eq. (48), and the last inequality uses Lemma G.4 with $\epsilon = \epsilon_{m(t)-1}$ (the condition to invoke this lemma is checked by Eq. (48)). Once the induction is done, we complete the proof. $\square$

**Lemma G.7.** *Suppose $\mathcal{E}$ holds. For all $t \in [T]$ with $m(t) \geq 2$,*

$$\max_{a \in \mathcal{A}} |\langle \widehat{s}_t - s^\star, a \rangle| \leq \epsilon_{m(t)-1}.$$

*Proof.* With Lemma G.6 in hand, we repeat Eq. (48) for all $m(t) \geq 2$ to complete the proof. $\qquad\square$

**Lemma G.8.** *For all $m \geq 2$, all $a \in \mathcal{A}$, and all $t \in \mathcal{T}_{m,a}^E$*

$$\pi_a(t) - \left( \max_{b \in \mathcal{A}} \langle \widehat{s}_t, b \rangle - \langle s^\star, a \rangle \right) \leq (2 + 64d) \, \epsilon_{m(t)-1} + \frac{1}{T}.$$

*Proof.* Let $b_t \in \mathrm{argmax}_{a \in \mathcal{A}} \langle c_{m(t)}, b \rangle$, $z_t \in \mathrm{argmax}_{a \in \mathcal{A}} \langle \widehat{s}_t, a \rangle$, and $m(t)$ be the phase that round $t$ lies in. One can show

$$\begin{aligned}
&\pi_a(t) - \left( \max_{b \in \mathcal{A}} \langle \widehat{s}_t, b \rangle - \langle s^\star, a \rangle \right) \\
&= \max_{b \in \mathcal{A}} \langle c_{m(t)}, b - a \rangle + (1 + 32d)\epsilon_{m(t)-1} + \frac{1}{T} - \langle \widehat{s}_t, z_t \rangle + \langle s^\star, a \rangle \\
&\leq \langle c_{m(t)}, b_t - a \rangle + (1 + 32d)\epsilon_{m(t)-1} + \frac{1}{T} - \langle \widehat{s}_t, b_t \rangle + \langle s^\star, a \rangle \\
&\leq \langle c_{m(t)} - s^\star, b_t - a \rangle + (2 + 32d) \, \epsilon_{m(t)-1} + \frac{1}{T} \\
&\leq (2 + 64d) \, \epsilon_{m(t)-1} + \frac{1}{T},
\end{aligned}$$

where the first inequality uses the definition of $z_t$, the second inequality holds due to $\mathcal{E}$ and the last inequality uses Lemma G.4 with $\epsilon = \epsilon_{m(t)-1}$. $\qquad\square$

### G.5. Lemmas for Elimination

**Lemma G.9.** *Suppose that $\mathcal{E}$ occurs. For every phase $m$, $a^\star \in \mathcal{A}_m$.*

*Proof.* We prove this by induction on $m$. Obviously, the base case holds. Suppose that the claim holds for phase $m$ and we consider phase $m + 1$. By the definition of $\mathcal{E}$, for any phase $m$

$$\forall a \in \mathcal{A}_m : \; |\langle \widehat{\nu}_m - \nu^\star, a \rangle| \leq \sqrt{2 \|a\|_{V_m^{-1}} \log \left( \frac{4KT}{\delta} \right)} \leq 2\sqrt{\frac{d \log(4KT\delta^{-1})}{T_m}} \leq \epsilon_m, \qquad (49)$$

where the second inequality uses Lemma G.6 which gives that each active arm $a$ will be played for $\lceil \rho_m(a) T_m \rceil$ times. We have for each $a \in \mathcal{A}_m$

$$\begin{aligned}
0 &\leq \langle \nu^\star + s^\star, a^\star - a \rangle \\
&\leq \langle \widehat{\nu}_m + s^\star, a^\star - a \rangle + 2\epsilon_m \\
&= \langle \widehat{\nu}_m + c_m', a^\star - a \rangle + \langle s^\star - c_m', a^\star - a \rangle + 2\epsilon_m \\
&\leq \langle \widehat{\nu}_m + c_m', a^\star - a \rangle + 32d\epsilon_m + 2\epsilon_m,
\end{aligned}$$

where the second inequality follows from Eq. (49) as well as the induction hypothesis $a^\star \in \mathcal{A}_m$, and the last inequality uses Lemma G.4 with $\epsilon = \epsilon_m$. To use Lemma G.4, we needs to verify the condition. By the definition of $\mathcal{E}$, for any round $t$ in elimination period:

$$\begin{aligned}
\max_{a \in \mathcal{A}} |\langle \widehat{s}_t - s^\star, a \rangle| &\leq \max_{a \in \mathcal{A}} \sqrt{2 \|a\|_{U_t^{-1}} \log \left( \frac{4KT}{\delta} \right)} \\
&\leq \max_{a \in \mathcal{A}} \sqrt{2 \|a\|_{V_{m(t)}^{-1}} \log \left( \frac{4KT}{\delta} \right)} \\
&\leq \underbrace{2\sqrt{\frac{d \log(4KT\delta^{-1})}{\min\{T_{m(t)}, (d\log(4KT\delta^{-1}))^{1/3} T_{m(t)}^{2/3}\}}}}_{= \frac{\epsilon_{m(t)}}{2}}, \qquad (50)
\end{aligned}$$

where the first inequality follows from the definition of $\mathcal{E}$, the second inequality uses Lemma G.6, and the last inequality holds due to $V_{m(t)} \succeq \sum_{a \in \mathcal{A}_m} \lceil \rho_m(a) T_m \rceil aa^\top$ and $V_{m(t)} \succeq \sum_{b \in \mathcal{B}_m} U_m(a) bb^\top$ where $U_m(a) = \lceil \omega_m(a)(d\log(4TK/\delta))^{1/3} \cdot T_m^{2/3} \rceil$.

This inequality implies $a^\star \in \mathcal{A}_{m+1}$. Once the induction done, the proof is complete. $\square$

**Lemma G.10.** *Let $m_a$ be the smallest phase such that $\Delta_a > (9 + 64d)\epsilon_{m_a}$. Suppose that $\mathcal{E}$ occurs. For each arm $a$ with $\Delta_a > 0$, it will not be in $\mathcal{A}_m$ for all phases $m \geq m_a + 1$.*

*Proof.* Consider any arm $a$ with $\Delta_a > 0$. We only need to consider $a \in \mathcal{A}_{m_a}$ and otherwise, the claim naturally holds. One can show

$$
\begin{aligned}
&\max_{b \in \mathcal{A}_{m_a}} \left\langle c'_{m_a} + \widehat{\nu}_{m_a}, b \right\rangle - \left\langle c'_{m_a} + \widehat{\nu}_{m_a}, a \right\rangle - (7 + 32d)\epsilon_{m_a} \\
&\geq \left\langle c'_{m_a} + \widehat{\nu}_{m_a}, a^\star \right\rangle - \left\langle c'_{m_a} + \widehat{\nu}_{m_a}, a \right\rangle - (7 + 32d)\epsilon_{m_a} \\
&\geq \left\langle c'_{m_a} + \nu^\star, a^\star \right\rangle - \left\langle c'_{m_a} + \nu^\star, a \right\rangle - (9 + 32d)\epsilon_{m_a} \\
&= \left\langle s^\star + \nu^\star, a^\star - a \right\rangle - \left\langle s^\star - c'_{m_a}, a^\star - a \right\rangle - (9 + 32d)\epsilon_{m_a} \\
&\geq \Delta_a - (9 + 64d)\epsilon_{m_a} \\
&> 0,
\end{aligned}
$$

where the first inequality follows from Lemma G.9 that $a^\star \in \mathcal{A}_{m_a}$, the second inequality holds due to Eq. (49), and the third inequality uses Lemma G.4 with $\epsilon = \epsilon_m$ (the condition to use Lemma G.4 is checked in Eq. (50)). According to the elimination rule (see elimination period in Algorithm 7), arm $a$ will not be in phases $m$ for all $m \geq m_a + 1$. $\square$

