# OpenReview forum: "Principal-Agent Bandit Games with Self-Interested and Exploratory Learning Agents"
_ICML.cc/2025/Conference — ICML 2025 poster_

### Official Review · Reviewer_YgEB · 2025-03-12

**Overall Recommendation:** 3

**Summary:**

The paper studies a principal-agent bandit game where the principal first provides an incentive, and then the agent selects the arm based on estimation and the provided incentive. The authors propose a novel elimination algorithm for the i.i.d. setting and the linear bandit setting. The corresponding regret upper bounds are $\sqrt{T}$ and $T^{2/3}$. The algorithm's regret outperforms that of Dogan et al 2023.

**Claims And Evidence:**

The paper proposes elimination algorithms with detailed analysis on the regret upper bounds. From my level of understanding, I do not find big issues in the analysis.

**Essential References Not Discussed:**

I'm unfamiliar with principal-agent bandit games so I'm unsure whether essential references are missed.

**Experimental Designs Or Analyses:**

The paper does not have experiments.

**Methods And Evaluation Criteria:**

Table 1 provides detailed regret comparison for different algorithms along with underlying assumptions, which makes a convincing point that the proposed algorithms outperform previous algorithms.

**Other Comments Or Suggestions:**

I do not have other comments.

**Other Strengths And Weaknesses:**

Strengths:
The proposed elimination framework with searching is novel and balances a good tradeoff between exploration and exploitation in the problem. Compared with Dogan et al. 2023, the algorithm outperforms in terms of regret upper bound.

Weakness:
I understand that the paper is of theoretical nature, but I believe that the paper can benefit from experiments on simulated data and real-world data.

**Questions For Authors:**

I'm confused of the annotation in Table 1 "worst-case bound......enjoys a gap-dependent bound". What are the gap-dependent bounds? Are they logarithmic of $T$?

**Relation To Broader Scientific Literature:**

The principal-agent bandit game is well motivated in the introduction with a good example. Previous papers on this topic usually assume that agents have access to true means, while this paper along with Dogan et al. 2023 bridge the gap without the assumption.

**Theoretical Claims:**

I have not checked correctness of all proofs.

---

> ### Author Rebuttal · Authors · 2025-03-31
>
> We thank the reviewer for the valuable feedback. Below we address your concerns.
>
> ---
> **W:** I understand that the paper is of theoretical nature, but I believe that the paper can benefit from experiments on simulated data and real-world data.
>
> **A:** Thank you for the suggestion. We do agree that experiments could provide additional value. Please see our response to Reviewer RBF5 (W3 and A3)
>
> ---
> **Q:** I'm confused of the annotation in Table 1 "worst-case bound......enjoys a gap-dependent bound". What are the gap-dependent bounds? Are they logarithmic of $T$ ?
>
> **A:** The gap-dependent bound we refer to is of the form $O ( \sum_{a \in [K]: \Delta_a>0}\frac{\log T}{\Delta_a})$ where $\Delta_a := \max_{b \in [K]} \\{\theta_b + \mu_b\\} - (\theta_a + \mu_a)$ measures the suboptimality gap of arm $a$ in the oracle-agent setup.
> As noted in the footnote of Table 1, Scheid et al. (2024b) obtain a similar gap-dependent regret bound, but only under the oracle-agent model. In contrast, for the more general learning agent setting, whether such a bound can be achieved remains an open question (see the discussion below Theorem 3.4). Intuitively, we are pessimistic about this possibility, as it appears necessary for the algorithm to continue sampling suboptimal arms in order to stabilize the agent’s internal estimates and reduce the estimation error of the optimal incentive (see Section 3.3 for a more detailed discussion).

---

### Official Review · Reviewer_YpW5 · 2025-03-12

**Overall Recommendation:** 3

**Summary:**

This paper studies the bandit principal-agent problem, where a principal tries to incentivize an agent playing a bandit instance so as to maximize their own cumulative reward. It extends the previous works of Scheid 2023 and Dogan 2023 by considering an agent who selects the arm based on *empirical* reward means. It covers both the i.i.d reward case and the linear reward case, with agents either self-interested (that is, greedy) or exploratory.

## update after rebuttal
I thank the authors for having clarified the validity of definition 2.1. I think the revised version of the article should feature the general version of definition 2.1 with the constants (a,b) (as explained by the authors in their reply) rather than just the special case (a,b) = (1/2, 1/2); as well as the discussion about the algorithms satisfying def 2.1.
I keep my score unchanged because I still think def 2.1 is slightly restrictive, but I'm convinced by the overall quality of the work.

**Claims And Evidence:**

The claims in the paper are appropriate, clear and supported by strong theoretical guarantees. The rates in the regret bound are consistent with existing literature, while derived under a less stringent assumption regarding the agent's behavior.

**Essential References Not Discussed:**

I do not think of any essential reference not discussed by the authors.

**Experimental Designs Or Analyses:**

NA

**Methods And Evaluation Criteria:**

There is not experiment supporting the results. This is not a problem to me, as the theoretical results seem strong enough.

**Other Comments Or Suggestions:**

- The definition of pi^0 should be introduced earlier in the paper, as it appears in the pseudo code of algorithm 1 without having been defined beforehand.
- The pseudo code of algorithm 1 could be clearer. For instance, in each phase m, it is not clear that the first Z_m rounds are dedicated to stabilizing estimators, then next T_m ones to determining optimal incentivizes: there could be an ambiguity about whether these rounds overlap.

**Other Strengths And Weaknesses:**

STRENGHTS:

- The results are interesting. The authors manage to match the existing bounds on regret (or with a slightly degraded rate, such as in the contextual case) while working under a less restrictive assumption for the agent.

- The paper is precise and well written. I particularly appreciated the effort to intuitively explain the steps of the algorithms, as well as the additional discussions in the appendix.

WEAKNESSES:

- While considering  an agent that compares empirical means rather than theoretical means is an improvement over Dogan (2023) and Scheid (2023), it seems that this paper employs many technics from these papers without bringing a lot  novelties from a mathematical point of view (except using online elimination and MSP).

- The pseucode of algorithm 1 is a bit hard to parse.

**Questions For Authors:**

- Is assumption in definition 2.1 satisfied by classic bandit routines, such as UCB or Thompson sampling?

- It is not clear to me why considering a vector of incentives rather than just a scalar incentive on one arm at each round (just as in Scheid 2023) is better. What are the advantages of this approach, given that the agent only pulls one arm at a time?

-

**Relation To Broader Scientific Literature:**

I find the discussion about the existing scientific literature very satisfactory. While the number of references is relatively low (I counted 32), this is understandable given that that principal-agent bandit games are a fairly new topic of research. The authors made a clear efffort to compare their results and assumptions with the two closest papers, namely Scheid 2023 and Dogan 2023. I particularly appreciated table 1, which gives a clear overview of these differences.

**Theoretical Claims:**

I did went the correctness of the proofs in the appendix.

---

> ### Author Rebuttal · Authors · 2025-03-31
>
> We thank the reviewer for the valuable feedback. Below we address your concerns.
>
> ---
> **W1:** It seems that this paper employs many technics from these papers (Dogan et al., 2023a) and (Scheid et al., 2024b) without bringing a lot novelties from a mathematical point of view (except using online elimination and MSP).
>
> **A1:** Here, we clarify that our techniques and analysis are **significantly different** from those in Scheid et al. (2024b), and **entirely distinct** from those in Dogan et al. (2023a).
>
> - **Comparison with Scheid et al. (2024b).** While Scheid et al. (2024b) use a standard binary search in an oracle-agent setting, we propose a noise-robust variant (Algorithm 3) specifically designed for the more challenging learning-agent setting. This is a non-trivial extension, as optimal incentives in our setting are time-varying—they depend on the agent’s evolving internal estimates. The search procedure must therefore be robust to fluctuations introduced by the agent’s learning process. Algorithm 3 is carefully designed to address these issues and is supported by a detailed analysis (see Lemma 3.1).
> We also introduce a novel elimination framework tailored to learning agents. Unlike traditional methods (e.g., Even-Dar et al., 2006), our algorithm does not permanently eliminate suboptimal arms but continues to play them moderately to stabilize the agent’s estimates. This is essential to avoid linear regret, as discussed in Section 3.3. In contrast, Scheid et al. (2024b) do not face these challenges, as agent responses in the oracle model are fixed for a given incentive.
>
> - **Comparison with Dogan et al. (2023a).** Our algorithms (Algorithms 5 and 6 in Appendices F and G) are fundamentally different from those of Dogan et al. (2023a). Their approach relies on an $\epsilon$-greedy strategy and explicitly estimates the agent’s model by solving an optimization problem at every round.
> In contrast, our approach is built around a **newly developed, novel** online elimination framework that **does not require** estimating the agent’s model. As noted in our response to Reviewer RBF5 (see W3 and A3), such estimation would incur significant computational and memory costs. A key innovation is our probabilistic amplification technique (see Section 4.2, also Algorithms 5 and 6), which ensures robustness to the randomness induced by the agent’s exploratory behavior. This technique supports two central components of our algorithm: **incentive testing** and **trustworthy online elimination**. Specifically, incentive testing refers to the process of identifying reliable incentives in the presence of agent learning, while trustworthy elimination ensures that, with high probability, the elimination process is not disrupted by exploratory actions of the agent.
>
> We believe that the techniques developed around probabilistic amplification offer a robust and flexible foundation for principal-agent interactions with learning agents, and may inspire further exploration in related dynamic settings.
>
> ---
> **W2:** Presentation of Algorithm 1 and introducing $\pi^0$ earlier.
>
> **A2:** Thanks for your suggestions. We will make sure to polish up our presentation according to your suggestions in the next version.
>
> ---
> **Q1:** Is assumption in definition 2.1 satisfied by classic bandit routines, such as UCB or Thompson sampling?
>
> **A:** Thank you for the insightful question. UCB does not satisfy Definition 2.1, as it continues to explore suboptimal arms indefinitely rather than decaying exploration over time. Whether Thompson Sampling satisfies the definition is unclear and may warrant further study. Since our focus is on designing principal algorithms for agents satisfying Definition 2.1, characterizing which other bandit algorithms meet this condition is beyond the scope of this work.
>
> ---
> **Q2:** Unclear why considering a vector of incentives rather than just a scalar incentive on one arm at each round (just as in Scheid 2024b) is better. What are the advantages of this approach, given that the agent only pulls one arm at a time?
>
> **A:** Incentivizing multiple arms is critical for our online elimination process as detailed in Section 3.4 (and its robust version for exploratory agent). Specifically, when conducting elimination at round $t$, the principal needs to compare $\hat{\theta}_a(t)+\hat{\mu}_a(t)+\epsilon$ against the empirical maximum $\max_b \hat{\theta}_b(t)+\hat{\mu}_b(t)$ over all active arm $a$, where $\epsilon>0$ is an error specified by the algorithm, $\hat{\theta}_a(t)$ is the principal's estimate and $\hat{\mu}_a(t)$ is the agent's estimate. Since the agent's estimates are unknown to the principal, the principal is unaware of which active arm achieves empirical maximum. To address this, the algorithm incentivizes every active arm $a$ by $\hat{\theta}_a(t)$ at round $t$. By doing this, the algorithm indirectly compares $\hat{\theta}_a(t)+\hat{\mu}_a(t)+\epsilon$ against empirical maximum by observing the played arm.

---

> > ### Comment · Reviewer_YpW5 · 2025-04-03
> >
> > I thank the authors for their detailed response and appreciate the clarifications regarding the technical novelties. However, I find it quite concerning that the authors are unable to present a classic bandit algorithm that satisfies Definition 2.1. This represents a significant limitation of their study from a practical perspective. Given the overall quality of the work, I am maintaining my recommendation; however, this issue should be explicitly addressed, or at least discussed, in the revised version.

---

> > > ### Author Response · Authors · 2025-04-05
> > >
> > > **Q:**  However, I find it quite concerning that the authors are unable to present a classic bandit algorithm that satisfies Definition 2.1. This represents a significant limitation of their study from a practical perspective.
> > >
> > >
> > > **A:** Thank you for the thoughtful comment. We would like to clarify two key points in the following response.
> > >
> > >
> > > **$\epsilon$-greedy type algorithms satisfy definition 2.1.**
> > > In the following, we show $\epsilon$-greedy type algorithms satisfy definition 2.1. For clarity, let's restate def 2.1 as follows.
> > >
> > >
> > > **Definition 2.1.** Let the probability that agent explores at
> > > round t be $p_t=\mathbb{P}(A_t\not\in \arg\max_{a\in \mathcal{A}}\\{\hat{\mu}_a(t)+\pi_a(t)\\})$. There exists a absolute constant $c_0 \geq 0$ such that $p_t \leq c_0 \sqrt{\log(2t)/t}$ at any time step $t \in [\tau,T]$ where $\tau \geq 2$ is the minimum integer satisfying $c_0 \sqrt{\log(2\tau)}<\sqrt{\tau}$.
> > >
> > >
> > > We point out that $\epsilon$-greedy type algorithms with decaying exploration rates $\epsilon_t=O(t^{-1/2})$ (e.g., Algorithm 2 of Dogan et al., (2023a) uses $\epsilon_t =O(t^{-1/2})$ for learning agent) satisfy this condition directly.
> > > While some variants (e.g., Alg 1.2, (Sliv19) choose $\epsilon_t =O( t^{-1/3})$) do not meet this exactly, they fall under a mild generalization of def 2.1 by assuming $p_t \leq c_0 \log^b(2t) t^{-a}$ where $a,b \in (0,1)$ are two absolute constant. This results in an additional regret in the order of $O( \log^b(T) T^{1-a})$.
> > >
> > >
> > >
> > > **Practical Perspective from Human Behavior.**  Definition 2.1 generalizes agent behavior models studied in prior principal-agent bandit work (e.g., Dogan et al., 2023a; Scheid et al., 2024b) by allowing agents to both learn their preferences and occasionally explore. In behavioral and cognitive science (e.g., [1–6]), the $\epsilon$-greedy model has been widely used to capture how humans make sequential decisions under uncertainty. While $\epsilon$-greedy is not always the most accurate model of human behavior, its simplicity and interpretability make it a valuable reference point. This further motivates our use of Definition 2.1 as a meaningful step toward modeling realistic learning agents.
> > >
> > >
> > >
> > >
> > > We will include this clarification and broader motivation in the revision.
> > >
> > >
> > >
> > > References:
> > >
> > >
> > > [Sliv19] Slivkins, Introduction to Multi-Armed Bandits, 2019.
> > >
> > > [1] Barron, G., Erev, I. Small feedback-based decisions and their limited correspondence to description-based decisions, Journal of behavioral decision making, 2003.
> > >
> > > [2] Kalidindi, K., Bowman, H. Using $\epsilon$-greedy reinforcement learning methods to further understand ventromedial prefrontal patients’ deficits on the Iowa Gambling Task, Neural Networks, 2007
> > >
> > > [3] Gershman, S., Deconstructing the human algorithms for exploration, Cognition, 2018
> > >
> > >
> > > [4] Lee, Michael D., et al. "Psychological models of human and optimal performance in bandit problems." Cognitive systems research 12.2 (2011): 164-174.
> > >
> > > [5] Daw, Nathaniel D., et al. "Cortical substrates for exploratory decisions in humans." Nature 441.7095 (2006): 876-879.
> > >
> > >
> > > [6] Cohen, Jonathan D., Samuel M. McClure, and Angela J. Yu. "Should I stay or should I go? How the human brain manages the trade-off between exploitation and exploration." Philosophical Transactions of the Royal Society B: Biological Sciences 362.1481 (2007): 933-942.

---

### Official Review · Reviewer_RBF5 · 2025-03-13

**Overall Recommendation:** 3

**Summary:**

This paper studies the problem of principal-agent interactions with self-interested agents. Different from previous studies like Dogan et al.(2023a, 2023b) and Scheid et al. (2024b), this paper assumes an empirical mean maximizer agent behavior model rather than true mean maximizer. The authors’ elimination framework and search algorithms are novel, efficiently handling uncertainty from greedy agent learning with iid rewards while achieving strong regret bounds. The authors also extend their algorithm to exploratory agents and linear rewards and show strong regret bounds under those settings as well.

**Claims And Evidence:**

Yes.

**Essential References Not Discussed:**

I did not notice any essential reference not being discussed.

**Experimental Designs Or Analyses:**

n/a.

**Methods And Evaluation Criteria:**

n/a.

**Other Comments Or Suggestions:**

n/a.

**Other Strengths And Weaknesses:**

On the positive side, this paper considers a learning approach to learning the near-optimal/no-regret principal policy in principal-agent bandit games. I found the problem presented by the paper interesting, and relevant. In general, this paper proposed an innovative research topic.

On the other hand, I think the outperformance of soft-O(T^{11/12}) is not a fair comparison, since their regret is defined with respect to different behavior models, i.e., true mean versus empirical mean. If we compare regret under different models, then it seems the proposed Algorithm 7 performs worse than C-IPA Scheid et al. (2024b) under the linear reward model.

What is more, besides the unfair comparison to prior work (e.g., Dogan et al., 2023a) due to different agent behavior model, the "significantly improve" claim remains unsubstantiated in practice without empirical comparisons. This paper is missing comparable algorithms to validate its performance, I wonder maybe if some naive heuristic strategies can perform well in practice.  Even simple synthetic experiments would strengthen its claims, and provide more insights for the algorithm's performance improvement.

**Questions For Authors:**

n/a.

**Relation To Broader Scientific Literature:**

This paper is related to the literature of principal-agent bandit games.

**Theoretical Claims:**

I did not check the correctness of proofs.

---

> ### Author Rebuttal · Authors · 2025-03-31
>
> We thank the reviewer for the valuable feedback. Below we address your concerns.
>
> ---
> **W1:** I think the outperformance of soft $O(T^{11/12})$ is not a fair comparison, since their regret is defined with respect to different behavior models, i.e., true mean versus empirical mean.
>
> **A1:** We would like to clarify that the comparison with the $O(T^{11/12})$ regret bound (Dogan et al., 2023a), as well as others in Table 1, **is indeed fair** because they can all be evaluated under the same behavior model and regret definition. Specifically, our behavior model generalizes theirs: by letting the agent always receive the constant reward equal to the true mean (as noted in Remark 2.2), our model reduces to theirs. In this case, the empirical mean becomes the true mean, ensuring that the regret definitions and behavior models align. Thus, our comparisons are fair.
>
> ---
> **W2:** If we compare regret under different models, then it seems the proposed Algorithm 7 performs worse than C-IPA Scheid et al. (2024b) under the linear reward model.
>
> **A2:** As mentioned in A1, regret bounds can be fairly compared when the models and regret definitions align, particularly when one model strictly generalizes another. Our setting is strictly more general than that of Scheid et al. (2024b), as their oracle-agent model is a special case of our learning agent framework. While our regret bound is indeed worse when restricted to this special case, our algorithm is designed to work in the broader and more challenging learning agent setting which is not addressed in Scheid et al. (2024b).
>
> ---
> **W3:** The "significantly improve" claim remains unsubstantiated in practice without empirical comparisons.
>
> **A3:** We thank the reviewer for this suggestion. Our primary focus in this work is on theoretical improvements in the order of the regret bound (improving from $O(T^{11/12})$ to $O(\sqrt{T})$ in exploratory oracle-agent setup). As such, this version does not include experiments. We do agree that empirical evaluation is a valuable direction and plan to pursue it in future work.
>
> Nonetheless, we would like to highlight several practical advantages of our algorithm compared to Dogan et al. (2023a), particularly in terms of implementation:
>
> - **Computational efficiency.** Their algorithm solves a linear program at each round, with the number of constraints growing linearly in $t$, leading to increasing runtime. In contrast, our Algorithms 5 and 6 only require simple arithmetic operations per round, resulting in much greater efficiency.
>
> - **Memory usage.** Their method stores all past incentives and actions, resulting in memory usage that scales with $t$. Our algorithms avoid this by maintaining only a few counters or estimates, ensuring constant or sublinear memory usage.
>
> - **Hyperparameter tuning.** Their algorithm requires two hyperparameters that must be carefully tuned. Our algorithms, by contrast, only require a standard confidence parameter $\delta \in (0,1)$, which typically does not require fine-tuning in practice.

---

### Decision · Program_Chairs · 2025-05-01

**Decision:**

Accept (poster)

**Comment:**

All the 4 reviewers appreciated the novel setup and developed approaches about the paper, which studies principal-agent framework for online learning. There is some concern during the discussion about agent's behavioral modeling and lack of experiments, but overall the reviewers are all positive about the contributions of this paper. Hope the authors could take into account various useful comments from the review process to improve the paper in its next version, but overall this paper is a great addition to ICML!